# A new instrument for stable isotope measurements of $^{13}$C and $^{18}$O in CO$_2$ - Instrument performance and ecological application of the Delta Ray IRIS analyzer

Jelka Braden-Behrens[1], Yuan Yan[1], and Alexander Knohl[1,2]

[1]University of Goettingen, Bioclimatology, Faculty of Forest Sciences and Forest Ecology, Germany
[2]University of Goettingen, Centre of Biodiversity and Sustainable Land Use (CBL), Germany

*Correspondence to:* Jelka Braden-Behrens (jbraden1@gwdg.de)

**Abstract.**

We used the recently developed commercially available Delta Ray Isotope Ratio Infrared Spectrometer (IRIS) to continuously measure the CO$_2$ concentration $c$ and its isotopic composition $\delta^{13}$C and $\delta^{18}$O in a managed beech forest in Central Germany. Our objectives are (a) to characterize the Delta Ray IRIS and evaluate its internal calibration procedure and (b) to quantify the seasonal variability of $c$, $\delta^{13}$C, $\delta^{18}$O and the isotopic composition of nighttime net ecosystem CO$_2$ exchange (respiration) $R_{eco}^{13}$C and $R_{eco}^{18}$O derived from Keeling-Plot intercepts. The analyzer's minimal Allan deviation (as a measure of precision) was below 0.01 ppm for the CO$_2$ concentration and below 0.03‰ for both $\delta$ values. The potential accuracy (defined as the $1\sigma$ deviation from the respective linear regression that was used for calibration) was approximately 0.45 ppm for c, 0.24‰ for $^{13}$C and 0.3‰ for $^{18}$O. For repeated measurements of a target gas in the field, the long-term standard deviation from the mean was 0.3 ppm for $c$ and below 0.3‰ for both $\delta$ values. We used measurements of nine different inlet heights, to evaluate the isotopic compositions of nighttime net ecosystem CO$_2$ exchange $R_{eco}^{13}$C and $R_{eco}^{18}$O in a three months measurement campaign in a beech forest in autumn 2015. During this period, an early snow and frost event occurred, coinciding with a change in the observed characteristics of both $R_{eco}^{13}$C and $R_{eco}^{18}$O. Before the first snow, $R_{eco}^{13}$C correlated significantly (p$< 10^{-4}$) with time-lagged net radiation $R_n$, a driver of photosynthesis and photosynthetic discrimination against $^{13}$C . This correlation became insignificant (p$> 0.1$) for the period after the first snow, indicating a decoupling of $\delta^{13}$C of respiration from recent assimilates. For $^{18}$O, we measured a decrease of 30‰ within 10 days in $R_{eco}^{18}$O after the snow event, potentially reflecting the influence of $^{18}$O depleted snow on soil moisture. This decrease was ten times larger than the corresponding decrease in $\delta^{18}$O in ambient CO$_2$ (below 3‰) and took three times longer to recover (three weeks vs. one week). In summary, we conclude that 1) the new Delta Ray IRIS with its internal calibration procedure provides an opportunity to precisely and accurately measure $c$, $\delta^{13}$C and $\delta^{18}$O at field sites and 2) even short snow or frost events might have strong effects on the isotopic composition (in particular $^{18}$O) of CO$_2$ exchange at ecosystem scale.

# 1 Introduction

The stable isotopic compositions of $CO_2$ and water vapor have been intensely used to study ecosystem gas exchange (Yakir and Sternberg, 2000). In particular, measurements of the $\delta^{13}C$ and $\delta^{18}O$ isotopic composition of $CO_2$ have provided important insights into the carbon cycle over a large variety of spatial and temporal scales (Flanagan and Ehleringer, 1998; Affek and
Yakir, 2014). There are many examples for the utility of the stable isotopic composition of $CO_2$ to study biosphere-atmosphere exchange processes on ecosystem scale, such as the partitioning of net ecosystem $CO_2$ exchange into respiration and photosynthesis. Different partitioning methods include the combination of gradient approaches with stable isotope measurements (Yakir and Wang, 1996), direct isotope gradient approaches (Zhang et al., 2006), the combination of eddy covariance measurements with isotope flask measurements (Bowling et al., 2001; Ogée et al., 2003; Knohl and Buchmann, 2005), and direct
isotope eddy covariance measurements (Wehr et al., 2016; Oikawa et al., 2017). Other field applications of stable $CO_2$ isotopes measurements investigate the temporal variability of the isotopic composition of a particular flux component. The temporal variability of the isotopic composition of respiration for example has been studied on timescales ranging from sub-diurnal (Barbour et al., 2011) to seasonal (Ekblad and Högberg, 2001; Bowling et al., 2002; Knohl et al., 2005). Further, the isotopic composition in $CO_2$ profiles has been studied on several sites over multiple years for $^{13}C$ (e.g. Bowling et al., 2002; Wehr
et al., 2016) as well as for $^{18}O$ (e.g. Bowling et al., 2003b; Shim et al., 2013). The $^{13}C$ composition of ecosystem respiration $R_{eco}^{13}C$ on the one hand, has been used to assess the time lag between assimilation and respiration (Ekblad and Högberg, 2001; Bowling et al., 2002; Knohl et al., 2005) and to evaluate biosphere models on global scale (Ballantyne et al., 2011). The $^{18}O$ composition of ecosystem $CO_2$ exchange $R_{eco}^{18}O$ on the other hand is particularly interesting to study the coupled $CO_2$ and water cycle (see e.g. Yakir and Wang, 1996).

A long established and broadly used technique to measure stable isotopic compositions is Isotope Ratio Mass Spectrometry (IRMS) (Griffis, 2013), a technique that is based on the fact that moving ions with different mass-to-charge ratio can be separated by (orthogonal) magnetic fields (Thomson, 1908). For measurements of the isotopic composition of $CO_2$, IRMS has typical precisions of approximately 0.02 to 0.1‰ for $^{13}C$ and 0.05 to 0.2‰ for $^{18}O$. IRMS has been widely used for isotope studies in environmental sciences, though it shows limited applicability for *in situ* measurements (Griffis, 2013) , but
see also the field applicable continuous flow IRMS described by Schnyder et al. (2004). Disadvantages of flask-sampling based IRMS techniques include high sample preparation effort and costs (Griffis, 2013), low temporal resolution and discontinuous measurements. Additionally, there are potential problems during sample storage and transport, see Knohl et al. (2004) for minimizing such storage effects in case of $^{13}C$. For $^{18}O$ storage effects can be related to oxygen exchange between water and $CO_2$ (Gemery et al., 1996; Tuzson et al., 2008). Optical based techniques can compete with or complement IRMS measurements
and progress in optical based techniques over the last decade enhanced the potential of measurements of isotopic compositions (Werner et al., 2012). These developments have a particular impact on micrometeorological studies, as they increased the accessibility of field-deployable optical instruments and thus enabled a number of micrometeorological applications of stable isotope techniques, as reviewed by Griffis (2013). Optical instruments to study the isotopic composition of trace gases use the absorption of infrared photons by exciting a molecule's rotational and vibrational energy states. These rotational and

vibrational transitions are characteristically different for isotopologues, defined e.g. by Coplen (2011) as 'molecular species that differ only in isotopic composition', (see e.g. Esler et al., 2000; Kerstel and Gianfrani, 2008)[1]. The isotopologue-specific absorption lines are related to the concentration of the respective isotopologue via Beer's law and thus the isotopic composition of a certain molecule (Werle, 2004). Available optical instruments that are capable of measuring isotopic compositions at trace
gas concentrations show different implementations of this principle by using different light sources (broadband light sources, mid or near infrared lasers) (see e.g. Griffis, 2013; Kerstel and Gianfrani, 2008) and/or different absorption cells (multi-path or resonant) (Werle, 2004). Minimal Allan deviations $\sigma_A$ and the corresponding averaging times $\tau_{\min}$ for different optical instruments are shown in Table 2, but see also Table 2 in the review of Griffis (2013) for more detailed information, including instrument stability and an overview of applications, for most of these instruments.

Here we present a new laser based direct absorption spectrometer in the mid infrared, the Isotope Ratio Infrared Spectrometer (IRIS) Delta Ray *Thermo Scientific Inc., Waltham, USA*. This spectrometer uses two tunable near infrared diode lasers in combination with a nonlinear crystal to produce a laser beam in the mid infrared (Thermo Fisher Scientific, 2014). The instrument scans a spectral region from 4.3293 $\mu$m to 4.3275 $\mu$m, containing four $CO_2$ absorption lines: at 4.3277 $\mu$m and 4.3280 $\mu$m (both for $^{16}O^{12}C^{16}O$), 4.3283 $\mu$m (for $^{16}O^{13}C^{16}O$), and 4.3286 $\mu$m (for $^{16}O^{12}C^{18}O$) (Geldern et al., 2014). A measured
and a fitted spectrum is shown in Fig. 1. The fitting procedure is based on a Voigt-Profile fit, that relates the isotopologue-specific absorption lines to their respective concentrations (information from the manufacturer, Thermo Fisher Scientific). The instrument has a flow rate of 0.08 slpm, a cell pressure of approximately 100 mbar, an optical path length of approximately 5 m and an internal calibration procedure that automatically includes two point calibrations for concentration $c$ and both $\delta$ values as well as corrections for the concentration dependency of the measured $\delta$-values (Thermo Fisher Scientific, 2014). The
objectives of our study are (a) to characterize the Delta Ray IRIS and its performance under field conditions as well as (b) to quantify the seasonal variability of $\delta^{13}C$, $\delta^{18}O$ and the isotopic composition of $CO_2$ exchange for both $\delta^{13}C$ and $\delta^{18}O$ derived from Keeling-Plot intercepts.

## 2  Material and methods

### 2.1  Field site

This study was conducted at a meteorological tower in a managed beech forest (*Fagus sylvatica L.*) in Thuringia (Central Germany) at 51°19'41,58" N; 10°22'04,08" E at 450 meters above sea level. The forest in the dominant wind direction of the tower has an average canopy height of approximately 34 m with approximately 120 year old trees, a top-weighted canopy and a homogeneous stand structure, surrounded by trees of three age classes (approximately 30–40, 80 and 160 years), (Anthoni et al., 2004). The field site is described in detail by Anthoni et al. (2004), and soil characteristics of this site were analyzed by
Mund (2004).

---

[1]In general this is also true for isotopemers, defined e.g. by Coplen (2011) as 'molecular species having the same number of each isotopic atom [...] but differing in their positions.', (e.g. Mohn et al., 2008).

## 2.2 Campaign design

We measured the $CO_2$ concentration $c$ and its isotopic composition $\delta^{13}C$ and $\delta^{18}O$ in ambient air from $21^{st}$ August 2015 to $16^{th}$ November 2015. We measured these quantities with the field deployable Isotope Ratio Infrared Spectrometer (IRIS) Delta Ray (*Thermo Scientific, Waltham, USA*) at nine inlet heights ranging from 0.1 to 45 m in an automatic measurement setup. After the tubing was purged for 60 s, each inlet was measured for 80 s, (consisting of four measurements each averaged for 20 s - thus the averaging time is longer than the instrument internal cell response time $\tau_{10\%}$ c.f. section 3.1.4). A full measurement cycle took 30 minutes and consisted of measurements of all nine inlet heights and a target standard with known $CO_2$ concentration and isotopic composition ($CO_2$ in synthetic air, tank 'SA-$CO_2$-5 in Table 3), supplemented by an internal calibration measurement, called 'referencing' (c.f. Sect. 2.8). In less detail, the experimental setup is also described in (Braden-Behrens et al., 2017).

We used the nighttime measurements of $c$, $\delta^{13}C$ and $\delta^{18}O$ of the different inlet heights in a Keeling-Plot approach (Keeling, 1958) to calculate the nighttime Keeling-Plot intercept that can be used to estimate the isotopic composition of nighttime net ecosystem $CO_2$ exchange (respiration) $\delta R_{eco}$ for both measured $\delta$ values: $^{13}C$ and $^{18}O$. Additionally, we used the half hourly measurements of the target standard to track the repeatability of the Delta Ray analyzer and performed additional (manual) measurements characterize the analyzer.

## 2.3 Spectrometer setup

We set up the spectrometer to use the absorption lines at 4.3277 $\mu$m (for $^{16}O^{12}C^{16}O$), 4.3283 $\mu$m (for $^{16}O^{13}C^{16}O$), and 4.3286 $\mu$m (for $^{18}O^{12}C^{16}O$). Thus, only three of the four absorption lines in the instrument's measured spectra (Fig. 1), were used for the spectral fit. In particular, for $^{16}O^{12}C^{16}O$, we did not use the strong absorption line at 4.3280 $\mu$m. The corresponding mode of operation is called 'high concentration mode' in the instrument's operational software QTEGRA. Additionally, the sample was dried before it entered the measurement cell with the (instrument's internal) Nafion drier.

## 2.4 Application of the Keeling-Plot approach

The Keeling-Plot approach (Keeling, 1958) is based on a simple two-component mixing model that describes how air from a source with effectively constant isotopic composition $\delta_s$ mixes with a background (with constant $c_{bg}$ and $\delta_{bg}$). For this simple two-component mixing model, one can derive a linear relationship between the measured isotopic composition $\delta_{meas}$ and the reciprocal concentration $1/c_{meas}$ by applying conservation of mass for the total concentration as well as for each isotopologue separately, for derivation (see e.g. Pataki et al., 2003).

$$\delta_{meas} = \underbrace{(\delta_{bg} - \delta_s)\, c_{bg}}_{m_{KP}}\ \frac{1}{c_{meas}} + \underbrace{\delta_s}_{\delta_{KP}} \tag{1}$$

This linear relationship with slope $m_{KP}$ and intercept $\delta_{KP}$ can be derived for each isotopic species independently, so in our case for both $\delta^{13}C$ or $\delta^{18}O$. The applicability of the Keeling-Plot approach to a certain experimental setup essentially depends on the question if $c_{bg}$, $\delta_{bg}$ and $\delta_s$ are constant over the spatial and temporal distribution of all measurements that are taken into account for the linear regression. In this study we apply a Keeling-Plot approach to a forest ecosystem, aiming at measuring

the isotopic composition of ecosystem integrated $CO_2$ exchange. The source of $CO_2$ is thus composed of different individual source components $i$ (e.g. stem, leaf and soil respiration), each accounting for the individual components with their isotopic compositions $\delta_{s,i}$ . The corresponding isotopic composition of the integrated source $\delta_s$ can be expressed by defining $\alpha_i$, as the relative contributions of the individual source components to the integrated source.

$$\delta_s = \sum_i \delta_{s,i}\, \alpha_i \qquad \text{with:} \qquad \sum_i \alpha_i = 1 \tag{2}$$

If the relative distributions among the different source components $\alpha_i$ produce significant changes in $\delta_s$ over the spatial and temporal distribution of measurements, the basic two component assumption of stable $\delta_s$ is violated. During daytime the application of a Keeling-Plot approach on ecosystem scale in a forest is in general problematic, as photosynthesis and respiration are two separately controlled and spatially separated processes - so we generally can not assume spatiotemporally constant $\alpha_i$. But for nighttime, when there is only respiration, the nighttime Keeling-Plot intercept $\delta_{\text{KP}}$ can be interpreted as the isotopic composition of nighttime net ecosystem $CO_2$ exchange (respiration) $\delta^{13}C\, R_{\text{eco}}$ or $\delta^{18}O\, R_{\text{eco}}$. Measures to assure and test the applicability of this two component approach and to improve the quality of the calculated Keeling-Plot intercepts are discussed and evaluated in appendix A. In brief, they include the minimization of the sampling time for each Keeling-Plot, an inclusion of all inlet heights into each Keeling-Plot analysis to increase the $CO_2$ concentration range, data filtering and weighted averaging of Keeling-Plots on smaller timescales.

## 2.5   Material and technical specifications

Technical specifications of the setup including plumbing and the automatic switching unit are shown schematically in Fig. 2. The automatic switching unit consisted of ten electromagnetic 3/2-way valves (Fig. 2) and was operated by a PC using a software for measuring technology (ProfiLabExpert 4.0, Abacom, Germany). The operating software controlled the valve positions using two USB relay boards (Abacom, Germany). When switching the valves to a new position, the operating software additionally sent a 1 s long rectangular trigger pulse with 5 V DC to one of the Delta Ray analyzer's two different analogue input channels. One of these channels was used when a target gas measurement had to be started, while a trigger pulse at the other input channel initialized the height measurements. After the Delta Ray analyzer received one of the trigger pulses, the tubes and the measurement cell were purged for 60 s before the analyzer took measurements for 80 s. This purging time was used to ensure that the first measurement after switching contained less than 0.1 % of the previously measured sample (c.f. Sect. 3.1.4).

We used poly ethylene (PE) tubes with 6 mm outer diameter and 4 mm inner diameter (Landefeld GmbH, Kassel, Germany) for the plumbing in the switching unit as well as for the nine height inlets. These inlets were additionally equipped with biweekly replaced 1.2 $\mu$m PTFE membrane filters (Rettberg GmbH, Göttingen, Germany). The tubes for the nine height inlets (c.f. Fig. 2) were all equally long (50 m) - except for the highest inlet that had to be extended to 52 m for practical reasons. The equal (or similar) length of the inlet tubes lead to similar flow rates in the tubing system and similar inlet pressures for the analyzer regardless of the valve position. This decreased pressure jumps when switching from one height position to another. We purged the main tube to reduce the time the air masses spend in the tubing. To avoid condensation, we heated the valve box

(at which we expect a pressure drop) and the adjacent tubing. For heating we used self-regulating heating wires (Horst GmbH, Lorsch, Germany) which produce a constant temperature of $65°C$. The flow rate in the height inlet tubes was approximately 1.5 slpm for all heights all the time and the major part of the gas flow was directed into the purging pump. In case of the target standard, the tubing was only purged when the target standard was measured. In this case, an overblow opened to enable gas release at approximately 1 slpm (Fig. 2). For the target measurements as well as for the height measurements the analyzer took a sub-sample of the corresponding inlet line with a flow rate of approximately 0.08 slpm. The flow in all tubing was laminar with Reynolds numbers below 100.

For measurements as well as calibration, we used gas tanks in 50 l steel containers at 150 to 200 bar pressure containing synthetic air, synthetic air with different $CO_2$ concentrations and pressurized air (Westfalen AG, Gleichen, Germany). Additionally, we used three 1 l gas tanks at 10 bar pressure with pure $CO_2$ at different (known) $\delta$ values that were shipped with the Delta Ray analyzer (Air Liquide, Düsseldorf, Germany). All used $CO_2$ containing gas tanks were measured high precisely for their $CO_2$ concentration and isotopic composition in $^{13}C$ and $^{18}O$ at the Max Planck Institute for Biogeochemistry in Jena. There, the $CO_2$ concentrations were measured with a Picarro CRDS G1301 and the isotopic composition was measured with IRMS linked to VPDB (VPDP-$CO_2$) by using the multi point scale anchor JRA-S06 (Wendeberg et al., 2013). The pure $CO_2$ tanks that were used for $\delta$ calibration were additionally measured for their $^{13}C$ composition with IRMS at Geoscience Center in Göttingen (Isotope Geology Division, Göttingen University). All known $\delta$-values and concentrations for the gas tanks used in this application can be found in Table 3 with their corresponding uncertainties.

## 2.6 Instrument characterization measurements

We carried out additional measurement in the field and in the lab to quantify precision, evaluate the calibration strategy and quantify the instrument's response time and repeatability. These measurements involved changes in the analyzers plumbing. For all measurements that required connecting different gas tanks to the analyzer, they were either connected directly to the analyzer's internal ports ('Ref1', 'Ref2', 'CRef1' and 'CRef2') or the plumbing was equivalent to the plumbing of the target gas (Fig.2).

1. Lab measurements to quantify precision and evaluate the calibration strategy

    – We measured the Allan deviation by connecting pressurized air at atmospheric $\delta$ values to the analyzer and took measurements at the analyzer's maximum data acquisition rate of 1 Hz for two hours.

    – We diluted pure $CO_2$ with synthetic air over a $CO_2$ concentration range of 200 to 1500 ppm to measure the concentration dependency of the measured (raw) $\delta$ values. This dilution experiment was carried out for three different tanks with pure $CO_2$ at different $\delta$ values. Each gas tank was measured twice. (Used gas tanks: 'ambient', 'bio1' and 'bio2', c.f. Table 3).

    – We measured the concentration $c$ and the isotopic compositions $\delta^{13}C$ and $\delta^{18}O$ of gases with concentrations ranging from (350 to 500 ppm) and isotopic compositions ranging from -37 to -9.7 ‰ for $\delta^{13}C$ and from -35 to -5 ‰ for

$\delta^{18}O$ . Each of these measurements was performed three times. (Used gas tanks: 'ambient', 'bio1','bio2', 'PA-tank', SA-CO$_2$-1,SA-CO$_2$-4, SA-CO$_2$-6, c.f. Table 3).

- We carried out repeated measurements of two pure CO$_2$ gas tanks at different $\delta$ values (diluted to different concentrations between 200 and 3000 ppm) as well as measurements of two gas tanks at different CO$_2$ concentrations (350 and 500 ppm). These measurements were repeated every six hours for a period of nine days. (Used gas tanks: 'ambient', 'bio', ('SA-CO$_2$-1 and 'SA-CO$_2$-6, c.f. Table 3.)'

2. Field measurements to quantify the setup's response time and repeatability

   - The response time of the tubing and the analyzer was measured by using the automatic switching unit (Fig. 2) to switch from ambient air (height9) to the target standard. We superimposed the measurements of four switching events to observe the adjacent temporal response processes.

   - The analyzer's repeatability under field conditions was quantified by the half hourly target measurements described in Sect. 2.5

## 2.7 Meteorological measurements

Supplementary to the measurements with the Delta Ray analyzer, the meteorological tower at the field site is equipped with an Eddy Covariance system to measure CO$_2$ and H$_2$O$_v$ fluxes as well as latent and sensible heat fluxes. Additional standard meteorological measurements include continuous measurements of short wave and long wave radiation, wind speed and direction, precipitation, air and soil temperature and air and soil humidity (Anthoni et al., 2004).

## 2.8 Calibration

### 2.8.1 Instrument internal calibration

The Delta Ray analyzer is equipped with three different internal calibration routines (Thermo Fisher Scientific, 2014). We performed these routines at the field site (*in situ*) each time the analyzer had to be restarted e.g. after power supply failures, instrument issues or when we manually turned off the analyzer for other reasons. All three instrument internal calibration procedures were usually done one day after restarting the analyzer, thus the instrument was in thermal equilibrium during calibration. The three different instrument internal calibration procedures are described below:

- **'Correction of concentration dependency' (called 'linearity calibration' in the instrument's documentation and operational software)**

  This calibration routine evaluates the concentration dependency of $\delta$ value measurements (Thermo Fisher Scientific, 2014). Mathematically, an experimentally derived correction factor $f_{\text{correct}}(c_{\text{raw}})$ is multiplied with the raw isotopic ratio R (information from the manufacturer, Thermo Fisher Scientific):

$$R_{\text{c-corrected}} = f_{\text{correct}}(c_{\text{raw}}) \times R_{\text{raw}} \tag{3}$$

This factor as a function of concentration is determined via a natural spline fit of measurements of a gas tank with constant $\delta$ value at different concentrations (information from the manufacturer, Thermo Fisher Scientific). This is implemented by mixing pure $CO_2$ with $CO_2$-free air, yielding concentrations between 200 to 3500 ppm. In our setup we used the pure $CO_2$ with near to ambient $\delta$ values (tank 'ambient', c.f. Table 3) and synthetic air for this calibration.

– **'Delta scale calibration'**

This calibration routine is based on a two-point-calibration of $\delta$ values using two tanks of pure $CO_2$ with different $\delta$ values, that are diluted with synthetic air. For this calibration, we used the pure $CO_2$ tanks 'ambient' and 'bio', c.f. Table 3.

    – **'Concentration calibration'**

This calibration routine performs a two-point-calibration for $CO_2$ concentration using two gas tanks with different $CO_2$ concentrations. We performed this measurement simultaneously to the other two calibration routines in the field, but for one particular calibration on $15^{th}$ of October, we had to replace it by a post-calibration, described in Sect. 2.8.2.

The instrument's internal calibration procedure is based on the measurement of these calibration curves after the instrument is started in combination with repeated measurements of a known gas, so called 'referencing' (see below). As the different

calibrations are only performed once after the instrument is restarted, the accuracy and repeatability of measurements is further based on the assumption that, these relationships remain sufficiently constant, and temporal changes are corrected by 'referencing'.

    – **'Referencing'**

This procedure applies an offset correction of the calibrated $\delta$ values using a gas with known $\delta$ values that is measured at a

freely selectable concentration in regular intervals (information from the manufacturer, Thermo Fisher Scientific). In our experimental setup, referencing is carried out every 30 minutes for 80 s after the tubes have been purged for 60 s using the pure $CO_2$ standard ('ambient', c.f. Table 3) diluted with synthetic air. We chose the reference concentration to be the same as the concentration at the highest inlet in the adjacent measurement cycle, because most of the measurement inlets had concentrations close to those at the highest inlet and the temporal variability of the measured concentrations generally

decreased with height. Thus, we performed the 'referencing' as close as possible to as many height measurements as possible by using these settings.

Thus, the calibration procedure for $\delta$ values can be expressed with the following formula with the correction factor $f_{\text{correct}}(c_{\text{raw}})$ as determined from the concentration dependency correction, and the slope $m_{\delta\text{scale}}$ derived from the $\delta$ scale calibration (information from the manufacturer, Thermo Fisher Scientific).

$$\delta_{\text{calibrated}}(R_{\text{raw}}; c_{\text{raw}}; t) = m_{\delta\text{scale}} \times \underbrace{\left( \frac{f_{\text{correct}}(c_{\text{raw}})R_{\text{raw}}}{R_{\text{std}}} - 1 \right)}_{\delta_{\text{c-corrected}}(c_{\text{raw}})} + \delta_{\text{Offset}}(t) \tag{4}$$

### 2.8.2 Post processing for concentration calibration

For the time period from the $15^{th}$ of October to $15^{th}$ of November, we replaced the instrument's internal concentration calibration by a manual linear calibration, based on manual measurements with six different gas tanks in the field. This was necessary, because measurements with these different gas tanks (including the target standard) showed a consistent linear re-
lationship between raw and known concentrations, that deviated from the linear relationship that was used in the instrument's internal calibration. Thus, we conclude that during this period there was a problem with the instrument's internal concentration calibration which might be related to gas flow or a leak during this particular concentration calibration.

## 2.9 Multilayer modeling

To test if the measured variability of the $^{13}$C composition of respiration can be partly explained by the variability of the $^{13}$C
composition of recent assimilates, we used the multilayer model CANVEG to simulate the isotopic composition of assimilated material during our measurement campaign. In particular, we analyzed the correlation of modeled $^{13}$C$_{\text{Ass}}$ to net radiation $R_n$, a driver of photosynthesis and photosynthetic discrimination, during our measurement period in autumn 2015. We further compared the resulting relationship between $R_n$ and $^{13}$C$_{\text{Ass}}$ to the observed (time lagged) relationship between $R_n$ and the $^{13}$C composition of ecosystem respiration R$^{13}_{\text{eco}}$C, derived from the measured Keeling-Plots, c.f. section 3.2.2. This analysis was per-
formed to test the hypotheses of a link between $\delta$ values in assimilated material and respiration. We used the multilayer model CANVEG to calculate the isotopic composition of assimilated material $\delta^{13}$C$_{\text{Ass}}$. CANVEG is a biophysical one-dimensional multilayer canopy model, (see e.g. Baldocchi, 1997; Baldocchi and Wilson, 2001). This multilayer model uses hourly meteorological inputs (among others temperature, radiation, vapor pressure deficit, wind velocity and $CO_2$ concentration) as main drivers, as well as site specific parameters (leaf area index, leaf clumping status, canopy height et. al.). Based on these input
variables, CANVEG iteratively computes the biosphere-atmosphere exchange of water, carbon dioxide and energy as well as the microclimate within and above the canopy at hourly time steps. The carbon, water and energy modules have been validated for various environmental conditions and forest types, (see e.g. Baldocchi et al., 1997, 1999, 2002). In particular, the model has also been applied to an unmanaged beech dominated forest field site in approximately $30\,\text{km}$ air-line distance to the measurement site of this study (Knohl and Baldocchi, 2008). The isotope enabled version of this model additionally calculates
$\delta^{13}$C$_{ij}$, the $^{13}$C composition of $CO_2$ for each canopy layer $i$ and each hourly timestep $j$ and the corresponding $^{13}$C composition of assimilated material $\delta^{13}$C$_{\text{Ass},ij}$ (Baldocchi and Bowling, 2003). In our application, we set up the model to use 40 equally thick layers $i$ and we used our meteorological measurements at the field site, described in Sect. 2.7, as input variables. We validated the model with Eddy Covariance measurements (Table 4) and used the model to calculate the isotopic composition of assimilated material $\delta^{13}$C$_{\text{Ass},ij}$ for each of the 40 canopy layers $i$ and for each hourly time step $j$. The $^{13}$C composition of
assimilated material $\delta^{13}C_{\text{Ass}}$ on daily timescale was calculated as an assimilation weighted sum over all layers and time steps, with the modeled assimilation rate $A_{ij}$ as a weighting factor:

$$\delta^{13}\text{C}_{\text{Ass}} = \frac{\sum_{i=1}^{40} \sum_{j=1}^{24} A_{ij} \cdot \delta^{13}\text{C}_{\text{Ass},ij}}{\sum_{i=1}^{40} \sum_{j=1}^{24} A_{ij}} \tag{5}$$

We included only hours $j$ and layers $i$ during photosynthesis (with positive assimilation rates).

## 3   Results and discussion

### 3.1   Instrument characteristics

#### 3.1.1   Precision

We use the Allan deviation $\sigma_A$ at different averaging times $\tau$ (Table 5) to characterize the Delta Ray IRIS analyzer's precision. Starting at an averaging time of 1 s, that corresponds to the analyzer's maximum data acquisition frequency, the Allan deviation $\sigma_A$ decreased with $\tau^{-1/2}$ (Fig. 3). This matches the expected behavior of a system that is dominated by white frequency noise. The measured Allan deviation $\sigma_A$ followed this slope up to averaging times for approximately 300 s for $\delta$ value measurements and approximately 200 s for concentration measurements. At these timescales the analyzer showed its maximum precision of 0.02 ‰ VPDB for $\delta^{13}$C, 0.03 ‰ VPDB-CO$_2$ for $\delta^{18}$O and 0.007 ppm for CO$_2$ concentration. For averaging times above 200-300 s other error sources (such as instrument drift) became significant. For $\delta^{13}$C, the precision of an earlier version of the instrument has also been measured by Geldern et al. (2014), reporting a minimum of $\sigma_A$ at around 0.04 ‰ for an averaging time of $\tau \approx 550$ s. At this averaging time, we measured a comparable (slightly better) Allan Deviation below 0.03 ‰ (c.f. Table 5). Two other averaging times are particularly interesting for our application: Firstly, the averaging period of 20 s yields Allan variances below 0.1 ‰ for both $\delta$ values and 0.02 ppm for CO$_2$ concentration. Secondly we set the IRIS analyzer's internal referencing procedure (described in Sect. 2.8) to 1800 s which corresponds to an Allan variance of 0.03 ‰ for $\delta^{13}$C and 0.08 ‰ for $\delta^{18}$O values and 0.01 ppm for CO$_2$ concentration.

#### 3.1.2   Evaluation of the calibration strategy

The instrument's internal calibration strategy (described in section 2.8) is based on:

 - A nonlinear relationship between raw $\delta$ values and concentrations (Fig. 4).

 - A linear relationship between calibrated $\delta$ value (measured with IRMS) and the concentration-corrected $\delta$ value - $\delta_{c-\mathrm{corrected}}$ in Eq. 4 (Fig. 5, middle and right panel).

 - A linear relationship between measured (raw) and real concentrations (Fig. 5, left panel).

 - The repeatability of the calibration curves – for $\delta$ values modulo the Offset correction, that is applied by the instrument's internal 'referencing' (Fig.6 and Table 7).

Raw $\delta$ values show a nonlinear dependency from raw concentrations (Fig. 4). This nonlinear relationship deviates from the concentration-dependency correction applied by the instrument, $\delta_{c-\mathrm{corrected}}(c_{\mathrm{raw}})$ in Eq. 4, as shown in Fig. 4. Here, the instrument internal concentration-dependency correction is shown for the used gas tank 'ambient' after an Offset correction

at a concentration of 400 ppm, which is similar to the instrument's internal 'referencing'. Thus, the meam deviations of the measured $\delta$ values from the concentration-dependency correction (top panel of Fig.4) give an estimate of the uncertainty of measurements that is related to the deviation from the reference concentration. For referencing at 400 ppm, these deviations were approximately below 0.2 ‰ for $^{13}$C and 0.4 ‰ for $^{18}$O.

The measured linear relationships for concentration and $\delta$ scale calibration (Fig. 5) have $R^2$ values of above 0.9999 for concentrations, above 0.999 for $\delta$ $^{13}$C, and above 0.998 for $\delta$ $^{18}$O. The linearity and potential accuracy, as defined by Tuzson et al. (2008), can be quantified as the $1\sigma$ standard deviation from the linear fits. The so defined potential accuracy of the instrument internal linear calibrations is 0.45 ppm for $CO_2$ concentration, 0.24 ‰ for $\delta^{13}$C and 0.3 ‰ for $\delta^{18}$O. For both $\delta$ values, this is comparable to the uncertainty related to the nonlinear concentration calibration that varies with $\delta$ and $c$ as
discussed above.

The repeatability of the calibration curves is discussed here based on measurements of the nonlinear concentration dependency (Fig. 4), and repeated measurements of gas tanks with two different $c$ and $\delta$ values to evaluate temporal changes in the respective linear relationships (Fig. 5). These measurements were taken every six hours for a period of nine days. For these repeated measurements the standard deviation of the calibrated values was below 0.2 ppm for concentrations and (if delta
values were measured at 400 ppm and referenced at 380 ppm) below 0.05 and 0.1 ‰ for $^{13}$C and $^{18}$O respectively. Thus, the uncertainty related to the repeatability of the linear calibrations is smaller than the potential accuracy discussed above. For $\delta$ values, these values are comparable to the repeatability reported by several authors measured with other laser spectrometers (e.g. Sturm et al., 2012, 2013; Vogel et al., 2013). For concentrations on the other hand, Sturm et al. (2013) reported a much smaller value of 0.03 ppm, based on more frequent calibration. In our setup, the concentration calibration is only performed
once after the instrument is restarted, thus there might be a potential for better repeatability in concentration measurements with more frequent concentration calibration. For $\delta$ values measured at concentrations that deviate further from the reference concentration (here 380 ppm), also the repeatability depends on concentration (Table 7). Repeated measurements of these deviations have standard deviations of below 0.15 ‰ for both $\delta$ values for concentrations between 200 and 1600 ppm.

For concentration measurements, the uncertainty related to the linear calibration dominates the overall uncertainty, whereas
the uncertainty of $\delta$-values measurements depends on the setup, in particular on the 'referencing'. If measurements were carried out at the concentration used during 'referencing', the accuracy is limited by the linear calibrations and the corresponding repeatability (c.f Table 6). If measurements are carried out at concentrations that deviate from the 'referencing' concentration, the accuracy is limited by the actual concentration dependency that deviates from the instrument internal correction of concentration dependency (c.f. Fig. 4 and Table 6). In this case, the accuracy could be further improved by applying a correction of
the concentration dependency based on more points.

### 3.1.3   Repeatability during the field campaign

We analyzed the repeatability of the Delta Ray analyzer under field conditions by evaluating half-hourly measurements of the same gas tank (SA-$CO_2$-5) during the whole measurement period. We use the standard deviations of measured concentrations and delta values to quantify the repeatability of our set up in the field including our calibration strategy. The standard deviations

of these long-term measurements were below 0.3 ppm for $CO_2$ concentration, below 0.2‰ for $\delta^{13}C$ and below 0.25‰ for $\delta^{18}O$ (frequency distributions and time series of the long term measurements are shown with color-coded metadata in Fig. 7.[2] For concentrations, the measured repeatability of approximately 0.3 ppm is slightly larger than the repeatability of the concentration calibration discussed above, but still below the potential accuracy discussed in section 3.1.2. In the case of $\delta$

values, the obtained repeatability of approximately 0.2‰ for $^{13}C$ and 0.25‰ for $^{18}O$ is larger than the repeatability of the linear calibration parameters obtained during lab measurements (0.05‰ for $^{13}C$ and 0.1‰ for $^{18}O$). The measured repeatability during the field campaign also exceeds the repeatability of the measurements of the concentration dependency (below 0.15‰ for both $\delta$ values over a large concentration range) c.f. section 3.1.2. This could be related to the fact, that the $\delta$ values of our target standard were out of the calibration range, leading to an enhancement of fluctuations in the calibration parameters.

### 3.1.4  Response time

We measured the response time of our system (tubing and measurement cell of the Delta Ray analyzer) by using the valve system shown in Fig. 2 to switch from ambient air with $\delta^{13}C \approx$-9‰ and $\delta^{18}O \approx$1‰ to tank air with $\delta^{13}C \approx$-38‰ and $\delta^{18}O \approx$-36‰. The time series of the measured $\delta$-values after the change of the valve position (Fig. 8) consisted of three different phases that can be related to different physical processes: Within a first phase, the measured $\delta$-values remained

constant for $\tau_1 \approx$14 s. This is the setup specific time it took for the gas to flush the tubes and valves before entering the cell. As a second phase, we observed a quadratic decay of the measured $\delta$-values, which we relate to mixing of gas within the tubes (before it enters the cell). This phase dominated the temporal response of our system for $\tau_2 \approx$4.5 s. The third phase of temporal response is the exponential decay with a characteristic decay time (defined here using the 10 %-threshold) $\tau_{10\%} \approx 10$ s for $\delta^{13}C$ and $\tau_{10\%} \approx 11$ s for $\delta^{18}O$. This exponential behavior can be derived for an idealized situation that includes perfect mixing in a

volume $V_{\mathrm{mix}}$ yielding:

$$\tau_{10\%} = \frac{log(10) \cdot p_{\mathrm{cell}} \cdot V_{\mathrm{mix}}}{\Phi}$$

With flow rate $\Phi$, cell pressure $p_{\mathrm{cell}}$ and effective mixing volume $V_{\mathrm{mix}}$. Using the volume of the measurement cell as an upper threshold for the effective mixing volume within the cell: $\max(V_{\mathrm{mix}}) = V_{\mathrm{cell}} = 80$ ml, we can calculate an upper threshold for $\tau_{10\%}$. With the instruments flow rate of $\Phi = 0.08$ slpm and the cell pressure of $p_{\mathrm{cell}} \approx 100$ mbar we get $\tau_{10\%,\mathrm{max}} \approx 13.6$ s. Thus

the measured value of $\tau_{10\%}$ is slightly below this value, indicating $V_{\mathrm{mix}} < V_{\mathrm{cell}}$. We define the total response time $\tau_{\mathrm{tot}}$ as the time-span it took until the step change between the two inlets reached 0.1 % of the corresponding difference in $\delta$ values, with $\tau_3 = \tau_{0.1\%} = 3 \cdot \tau_{10\%}$. The three different phases of instrument response (tube transport $\tau_1$, tube mixing dominated change $\tau_2$ and cell mixing dominated change $\tau_3$) summed up to a net response time $\tau_{\mathrm{tot}} = \tau_1 + \tau_2 + \tau_3 < 60$ s. Thus, the cell flushing time of our application (60 s) is appropriate to produce independent measurements of two different inlets.

---

[2]In the case of $^{13}C$ , we excluded the target measurements between $23^{rd}$ of September till $29^{th}$ of September, because we obtained a problem with the $^{13}C$ calibration that lead to a large jump in the $\delta$ $^{13}C$ value of the (very depleted) target standard. This jump did not occur in the height measurements, probably because they were much closer to the reference $\delta$ value.

### 3.1.5 Utilization rate, power consumption and maintenance effort

We define the utilization rate as the number of successfully recorded measurement cycles divided by the number of measurement cycles that were theoretically possible during the field campaign (approximately 4200). This can be calculated separately for a) profile measurements and b) target gas measurements, because some data gaps were specific for target measurements. The utilization rate for was approximately 80 % for measurements of the height profile and approximately 70 % for target gas measurements. Two major reasons for data gaps reduced the utilization rate for both, profile and target measurement by 8.6 % (a laser alignment problem that was resolved after 7 days) and 6 % (three data acquisition problems, the longest lasting three days). Additionally, four external power supply problems at the field site lead to a further reduction of the utilization rate by 3.3 %. These data gaps, as well as smaller datagaps, that reduced the utilization rate are listed in Table 8. In case of target measurements, the main reason for data gaps (accounting for a reduction of utilization rate of more than 9 %) were plumbing issues that lead to a contamination of the target gas by ambient air. Thus a more stable target plumbing would be a promising approach to increase the utilization rate, as well as a more stable power supply and more frequent field trips.

Maintenance effort and power consumption of the whole setup were moderate: The analyzer's power consumption of approximately 220 W was slightly smaller than the power consumption of the basic infrastructure of the setup that included the pump to purge the nine inlet tubes and the heated valve box (330 W). To maintain and to control the setup, we went to the field site weekly or biweekly and used remote access to the instrument via a satellite connection.

## 3.2 Ecological Application

### 3.2.1 Time series of measured quantities

The measured $CO_2$ concentrations in 45 m height at our field site in a managed beech forest in Central Germany ranged from 385 to 450 ppm with corresponding $\delta$ values between -11 to -7‰ for $^{13}C$ and between -6 and 2‰ for $^{18}O$ over a three-month period in autumn 2015 (Fig. 9). As the lower heights commonly contain larger amounts of respired $CO_2$ with a typically lighter carbon and oxygen composition, the lower inlets show larger $CO_2$ concentrations $c$ with smaller $\delta$ values. We calculated a three-month time series of nighttime Keeling-Plot intercepts $\delta^{13}C_{KP}$ and $\delta^{18}O_{KP}$, that can be interpreted as the respective isotopic composition of nighttime net ecosystem $CO_2$ exchange (respiration) $R_{eco}^{13}C$ and $R_{eco}^{18}O$ (shown with temperature and precipitation data in Fig. 10). A particular feature of the measurement period is an early snow and frost event with negative temperatures during four nights between 11. and 15. October 2015 (Fig. 10). The corresponding snow event on $13^{th}$ of October was visible on a canopy picture, taken at midday on 13. October 2015. The time of the snow and frost event coincided with changes in the characteristics of $\delta^{18}O$, $R_{eco}^{18}O$ and $R_{eco}^{13}C$: For $\delta^{18}O$ and $R_{eco}^{18}O$ a strong decrease was obtained after the snow event. This decrease was the largest signal in the respective time series. For $R_{eco}^{13}C$, the analysis of its potential meteorological drivers yielded different results for the time periods before and after the first snow. Additionally, according to Eddy Covariance measurements, the forest was a net $CO_2$ sink with negative diurnal net ecosystem exchange (NEE) before the 12. October (with only one exception), whereas it was a net $CO_2$ source with positive diurnal NEE after the snow event on 13. October (also with only one exception).

### 3.2.2 Potential drivers for $R_{\mathrm{eco}}^{13}\mathrm{C}$

Previous studies linked the temporal variability of the $^{13}C$ composition of ecosystem respiration $R_{\mathrm{eco}}^{13}\mathrm{C}$ partially to changes in the meteorological conditions during photosynthesis, namely relative humidity RH, Vapor pressure deficit VPD, photo-synthetically active radiation (PAR) and the ratio VPD/PAR (Ekblad and Högberg, 2001; Bowling et al., 2002; Knohl et al.,

2005). These links occurred with time lags that correspond to the time lag between assimilation and respiration, which is approximately four to five days for mature trees (Kuzyakov and Gavrichkova, 2010). The observed time lagged links between meteorological variables and $R_{\mathrm{eco}}^{13}\mathrm{C}$ were interpreted by the respective authors as an indication for a link between the isotopic composition of respiration $R_{\mathrm{eco}}^{13}\mathrm{C}$ and the isotopic composition of recent assimilates $\delta^{13}\mathrm{C}_{\mathrm{Ass}}$, which is controlled by photo-synthetic discrimination of the heavier $^{13}C$ according to the Farquhar Model (Farquhar et al., 1989). Thus, in accordance with

previous studies, we hypothesize that:

     **Hypothesis (a):** The variability of $R_{\mathrm{eco}}^{13}\mathrm{C}$ can be partly explained by the isotopic composition of recent assimilates $\delta^{13}\mathrm{C}_{\mathrm{Ass}}$, which is controlled by meteorological drivers during photosynthesis according to the Farquhar model. Thus, the variability of $R_{\mathrm{eco}}^{13}\mathrm{C}$ is linked to the variability of meteorological drivers of photosynthesis and photosynthetic discrimination with a time lag that is consistent with the time lag between respiration and assimilation.

To test this hypothesis, we calculated the Pearson correlation coefficient $r_{\mathrm{pear}}$ between $R_{\mathrm{eco}}^{13}\mathrm{C}$ and the n-day sum (with n from 1 to 6) of the meteorological quantities that we expect to control $^{13}C$ discrimination for different time shifts $\tau$. For the time period before the first snow (when the ecosystem was a $CO_2$ sink), the strongest correlation we found was a moderate negative correlation between $R_{\mathrm{eco}}^{13}\mathrm{C}$ and the two-day-sum of net radiation $R_n$ with a time shift $\tau$ of two days (Fig. 11). This correlation is significant with a Pearson correlation coefficient $r_{\mathrm{pear}}$ of approximately -0.56, which is clearly beyond the corresponding

critical value of approximately $\pm$ 0.38 for N=45 and $\alpha = 0.005$. The time lag of this correlation is in accordance with the expected time lag between assimilation and respiration of two to five days for mature trees (Kuzyakov and Gavrichkova, 2010). But the correlation itself cannot be directly explained by the Farquhar model of discrimination as radiation influences both, the $CO_2$ supply (by influencing stomatal conductance) and the $CO_2$ demand (by influencing assimilation) in the leaf (Farquhar and Sharkey, 1982). In particular we did not find a significant time lagged positive correlation between $R_{\mathrm{eco}}^{13}\mathrm{C}$ and

VPD, RH or the ratio VPD/PAR (Fig. 11), which could be directly associated with the Farquhar Model and have been found by the above mentioned studies. To test if it might be still reasonable to interpret the observed negative correlation of $R_{\mathrm{eco}}^{13}\mathrm{C}$ with $R_n$ as a time lagged link between $R_{\mathrm{eco}}^{13}\mathrm{C}$ and isotopic composition of recently assimilated material $\delta^{13}\mathrm{C}_{\mathrm{Ass}}$ on ecosystem scale, we performed a more complex calculation of $\delta^{13}\mathrm{C}_{\mathrm{Ass}}$ by using the multilayer model CANVEG, c.f. Sect. 2.9. The advantage of CANVEG is that it accounts for the non-linear interactions between air temperature, air humidity, radiation,

stomatal conductance and photosynthesis. Before the first snow event during our measurement period, the modeled $\delta^{13}\mathrm{C}_{\mathrm{Ass}}$ correlated significantly with the diurnal sum of net radiation $R_n$ with an $r_{\mathrm{pear}}$ of 0.89 (Fig. 12) - corresponding $r_{\mathrm{crit}} \approx 0.33$ for N=63. But in contrast to the time lagged correlation, which we found in our Keeling-Plot data, this correlation is positive (Fig. 12). As the multilayer model does not support the interpretation of the observed negative correlation between $R_n$ and $R_{\mathrm{eco}}^{13}\mathrm{C}$ through the variability of the isotopic composition of recent assimilates $\delta^{13}\mathrm{C}_{\mathrm{Ass}}$, it does not support hypothesis (a).

An alternative interpretation of the observed correlation between the isotopic composition of respiration $R_{\text{eco}}^{13}C$ and net radiation $R_n$ would be a link between $R_{\text{eco}}^{13}C$ and the amount of recent assimilates (alternatively to the isotopic composition of recent assimilates). Because soil respiration has been measured to account for around 80% of ecosystem respiration in an old beech forest in below 30 km distance to our field site (Knohl et al., 2008), we assume that soil respiration dominates ecosystem respiration and thus we further focus on soil respiration and discuss the following hypothesis:

**Hypothesis (b):** The observed time-lagged correlation between $R_{\text{eco}}^{13}C$ and net radiation $R_n$ is related to the temporal variability of the ratio of autotrophic to total soil respiration[3].

A link between photosynthesis and autotrophic soil respiration has been shown in many studies throughout different ecosystems, including a beech dominated forest in less than 30 km air-line distance to our field site in a managed beech forest (Moyano et al., 2008). In this study, the authors found that 73 % of the variability in rhizosphere respiration (the major part of autotrophic soil respiration) correlated with photosynthesis (GPP) and the ratio between autotrophic and total soil respiration was approximately 50 %. Additionally, evidences for a large temporal variability on diurnal and seasonal scale of the contribution of autotrophic to total soil respiration have been reported for a temperate hardwood forest (Savage et al., 2013) and for a mature temperate boreal forest (Carbone et al., 2016). In our field experiment, the observed correlation between $R_{\text{eco}}^{13}C$ and $R_n$ with an $r_{\text{pear}}$ of 0.56 (and thus ($r_{\text{pear}}^2 = 0.3$)) links 30 % of the variability of $R_{\text{eco}}^{13}C$ to $R_n$ with a time lag of 2-4 days. As the measured isotopic composition of ecosystem respiration $R_{\text{eco}}^{13}C$ spanned a range of 6‰, this corresponds to a range of 1.8 %. Hypothesis (b) would further imply, that this variability over a range of 1.8‰ corresponds to that proportion of the variability of autotrophic respiration that is linked to photosynthesis. If we estimate this proportion to represent 73 % of the total variability of autotrophic respiration (following Moyano et al., 2008), the corresponding total variability of autotrophic respiration would correspond to a range of approximately 2.5‰. If in autumn the ratio of autotrophic to total respiration would approximate 0 %, this value of 2.5‰ would be equal to the difference $\Delta_{\text{tot}-\text{aut}} = \delta_{\text{tot}} - \delta_{\text{aut}}$ between the isotopic composition of total respiration $\delta_{\text{tot}}$ and the isotopic composition of autotrophic respiration $\delta_{\text{aut}}$. In general, a value of $\Delta_{\text{tot}-\text{aut}} = +2.5‰$ is within the range of differences, that have been reviewed to be on average about +4‰ (Bowling et al., 2008) for different ecosystems. A positive value of $\Delta_{\text{tot}-\text{aut}}$ with a lighter $\delta^{13}C$ composition of autotrophic respiration would be consistent to Hypothesis (b). As a note of caution, however, none of the studies that analyze autotrophic soil respiration in the above mentioned review, was performed in a forest ecosystem. For C3 woody species, including forests, more enriched $\delta^{13}C$ values of autotrophic soil respiration, and thus negative values for $\Delta_{\text{tot}-\text{aut}}$, have been reported (Ghashghaie and Badeck, 2014). In a beech forest in Southern Germany, the sign of some involved fractionation effects varied temporally (Paya et al., 2016). Thus, the comparison with literature data about the temporal variability of the ratio between autotrophic and total soil respiration and the respective isotopic compositions gives the possibility that hypothesis (b) is true, but we can, however, not prove it without additional independent measurements. To test this hypothesis, we would need to measure the amount and the isotopic composition of autotrophic respiration, total soil respiration and ecosystem respiration (e.g. by a trenching experiment) at our field site with an appropriate time resolution to capture the day-to-day variability during the field campaign. Lab measurements

---

[3]The term 'autotrophic' is not consistently defined among different authors. Here we use this term equivalent to 'root derived respiration', including respiration from the living root tissue, from micro-organisms in the rhizosphere and mycorrrhizal symbiotic funghi.

using incubations could also give an idea of the isotopic composition of autotrophic and total soil respiration, but would not fully reflect field site conditions.

### 3.2.3 Characteristics of $R_{\text{eco}}^{18}O$ and $\delta^{18}O$

The seasonal variability of $\delta^{18}O$ and $R_{\text{eco}}^{18}O$ (shown in Fig. 9 and Fig. 10) are influenced by oxygen exchange when $CO_2$ gets
dissolved in different water pools (e.g. leaf and soil water) with variable isotopic compositions. These isotopic compositions in turn, are controlled by multiple physical and biological factors such as temperature, precipitation, vapor pressure deficit (VPD) or the activity of the enzyme carbonic anhydrase, that accelerates the oxygen exchange between water and $CO_2$, (Miller et al., 1999; Farquhar et al., 1993; Gillon and Yakir, 2000; Bowling et al., 2003a; Wingate et al., 2009). The strongest feature of the measured time series of $R_{\text{eco}}^{18}O$ is an approximately 30‰ large decrease within ten days from approximately -18‰ on 7.
October to approximately -46‰ on 18. October (Fig. 10). During the same time period, the $\delta^{18}O$ value of nighttime ambient $CO_2$ in 45 m height decreased from approximately -1‰ down to -3.5‰ at nighttime and down to -6‰ during daytime (Fig. 9). As for $R_{\text{eco}}^{18}O$, this decrease is the strongest signal in the measured time series of $\delta^{18}O$. The time of these decreases in $R_{\text{eco}}^{18}O$ and $\delta^{18}O$ coincided with the time of the first snow and frost event in autumn 2015. This indicates that the snow event has a noticeable effect on both $\delta^{18}O$ and $R_{\text{eco}}^{18}O$, but as the change in (nighttime) $R_{\text{eco}}^{18}O$ is more than ten times larger than
the corresponding change in $\delta^{18}O$ of nighttime $CO_2$ this effect is particularly enhanced for $R_{\text{eco}}^{18}O$. For comparison, similar strong peaks in $R_{\text{eco}}^{18}O$ have been observed in a semi-arid woodland after precipitation in New Mexico (Shim et al., 2013), but this study refers to a monsoon dominated ecosystem with comparably large variability in the $^{18}O$ and does not focus on the difference of these pulses of snow and rain events.

Possible explanations for the observed large decreases in both $\delta^{18}O$ and $R_{\text{eco}}^{18}O$ after the snow would involve the $^{18}O$
exchange of $CO_2$ with water pools that are fed by the recent snow event and the response to changes in multiple of the above mentioned physical and biological factors that influence the oxygen exchange between $CO_2$ and water. One of the factors that can cause a depletion in $^{18}O$ due to the exchange of oxygen between $CO_2$ with snow-fed water pools is the fact that snow has in general a lighter $^{18}O$-composition than rain. The isotopic composition of rain can often be related to Rayleigh fractionation processes (Gat, 1996) and thus is related to isotopic exchange between the raindrops and air masses in clouds
when rain is falling (Gat, 1996, citing Bolin, 1959 and Friedman et al., 1962). As a result of the continuous isotopic exchange with air masses in the cloud, raindrops do not carry the very depleted isotopic composition within the cloud whereas for snow, the isotopic exchange between the falling snowflakes and the air masses in the cloud does not take place, resulting in a more depleted precipitation (Gat, 1996). As example, Orlowski et al. (2016) reported a maximal difference of approximately 15.5‰ between the $\delta^{18}O$ values of rain and snow over a two-year measurement period at a field site in approximately 160 km
air-line distance from our field site. A smaller maximal difference of approximately 9‰ between the $\delta^{18}O$ of snow and the corresponding monthly means for rain was reported by Wenninger et al. (2011), based on two years of measurements at two catchments in German black forest 414 km air-line distance from our field site. Thus, the $^{18}O$-depleted isotopic composition of snow compared to rain may explain some of the observed 30‰ decrease in $R_{\text{eco}}^{18}O$. One possible additional effect could be the fact that soil respired $CO_2$ is typically in equilibrium not with rain, but with soil water in the top soil layers (0 to 20 cm) (Miller

et al., 1999; Wingate et al., 2009). Evaporative effects can shift the isotopic composition in the upper soil layers towards more $^{18}$O enriched values (Miller et al., 1999; Wingate et al., 2009) potentially increasing the $\delta^{18}$O difference before and after the snow event.

We tested the correlation between $R^{18}_{\mathrm{eco}}$O and different meteorological variables that potentially control the isotopic com-
position of different water pools within the ecosystem over the whole measurement period as well as the sub-periods before and after the first snow (Table 9). As the underlying multiple interaction processes between oxygen in $CO_2$ and different water pools and the respective isotopic compositions of these pools are complex, this analysis was not performed to causally link the measured $R^{18}_{\mathrm{eco}}$O to a single meteorological driver but rather to look for changes of these correlations that could be interpreted as changes in the processes that drive $R^{18}_{\mathrm{eco}}$O before and after the snow event. For the whole measurement period, the strongest of the analyzed correlations was a correlation between $R^{18}_{\mathrm{eco}}$O and soil moisture at a depth of $8\,\mathrm{cm}$ with an $R^2$ of 0.49 and p$<10^{-9}$. As this correlation becomes insignificant when it is calculated for the periods before and after the snow separately, it can be related to the strong decrease in $R^{18}_{\mathrm{eco}}$O after the snow event that correlates to a rise in soil moisture when the snow melts (Fig. 10). This would be consistent to a heavier $^{18}$O composition in the top soil layers (due to evaporation) before the snow, yielding also higher $\delta^{18}$O values of $R^{18}_{\mathrm{eco}}$O. Also other variables that correlated significantly with $R^{18}_{\mathrm{eco}}$O during the whole measurement period such as soil and air temperatures or shortwave radiation (Table 9) are related to soil evaporation. For the sub-periods before and after the first snow, we found multiple significant correlations with meteorological drivers such as soil and air temperatures, pressure or actual vapor pressure (Table 9). The significant correlations before the first snow become insignificant (or less significant) after the snow and vice versa. This behavior indicates a difference in the processes that drive the $^{18}$O isotopic composition of nighttime net ecosystem $CO_2$ exchange $R^{18}_{\mathrm{eco}}$O before and after the snow event.

## 4  Conclusions

Field-applicable instruments to analyze the isotopic composition of $CO_2$ have a large potential to be useful for long term measurement setups on meteorological towers and networks such as ICOS https://www.icos-ri.eu/ or NEON http://www.neonscience.org/ to deliver new insights into the carbon cycle. The new Isotope Ratio Infrared Spectrometer (IRIS) Delta Ray used in this study provides an opportunity to measure the $CO_2$ concentration $c$ and its isotopic compositions $\delta^{13}$C and $\delta^{18}$O with limited maintenance effort at remote sites. Here, we evaluate the instrument internal calibration and demonstrate the field applicability of the Delta Ray IRIS, which we used to measure $c$, $\delta^{13}$C and $\delta^{18}$O in a managed beech forest for three months in autumn 2015. The Delta Ray IRIS implemented here with the instrument's internal calibration, showed adequate precision, accuracy and repeatability to perform robust measurements of $c$, $\delta^{13}$C and $\delta^{18}$O in air in our continuous setup. For measurements of $\delta$ values at concentrations that deviate from the 'referencing' concentration, the uncertainty is dominated by the instrument internal correction of concentration dependency and improvements of the accuracy could potentially be achieved by more detailed analysis of this concentration dependency. The easy operation of the automatically calibrated Delta Ray IRIS allowed us to measure seasonal variability of the isotopic composition of nighttime $CO_2$ exchange based on Keeling-Plots. The

strong effect of the first frost and snow event on both the $\delta^{13}$C and $\delta^{18}$O of nighttime $CO_2$ exchange indicates that singular events, even if short, may strongly influence the isotopic imprint of terrestrial ecosystems on atmospheric $CO_2$.

## 5 Code availability

An earlier version of the multilayer model CANVEG can be found here:

https://nature.berkeley.edu/biometlab/BiometWeb/canoak_V2.c

## 6 Data availability

All data used for the figures presented here is provided in the supplementary material.

## 7 Appendices

    – Appendix A: Measures to improve data quality

*Author contributions.* The authors contributed to this paper in the following ways: The experimental setup in the field and data processing were carried out, discussed and interpreted by Jelka Braden-Behrens and Alexander Knohl. The validation of and simulations with the multilayer model CANVEG (Sect. 2.9) were performed, discussed and interpreted by Yuan Yan, Jelka Braden-Behrens and Alexander Knohl. All authors proof-read and commented on the paper.

## 8 Competing interests

We declare that we have no conflict of interest. However, we borrowed the instrument at no cost from *Thermo Scientific*, Waltham, USA, to perform instrument tests and to use the instrument under field conditions. Additionally we could do lab measurements to evaluate the calibration strategy at the facility of Thermo Scientific in Bremen at no cost. Jelka Braden-Behrens and Alexander Knohl published a short (not peer reviewed) technical paper (Braden-Behrens et al., 2017) together with HJ Jost and Magda Mandic, who were both working for *Thermo Scientific*, when we borrowed the instrument. This not peer reviewed technical paper also briefly describes our experimental setup
and discusses instrument performance.

    *Acknowledgements.* This project was partly funded by the Dorothea-Schlözer-Fellowship and by the German Research Foundation (DFG, project ISOFLUXES KN 582/7-1). The work was partially supported by the European Research Council under the European Union's Horizon 2020 research and innovation programme (grant agreement n. 682512 – OXYFLUX). We thank *Thermo Scientific*, Waltham, USA, to borrow us the instrument for free and made it possible for us to do additional measurements in the lab. We particularly thank HJ Jost, Magda Mandic
and Danijela Smajgl for their advice and support especially concerning how to set up, calibrate and operate the Delta Ray analyzer. The technicians of the bioclimatology group of the University of Goettingen, especially Dietmar Fellert Frank Tiedemann and Edgar Tunsch

as well as the student assistant Elke Schäpermeier helped substantially with the experimental setup and maintenance. We thank Yakov Kuzyakov, Lydia Gentsch and Mattia Bonazza for their remarks on data interpretation. Additionally we thank the forest manager Ulrich Breitenstein for allowing the experimental setup at this site.

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

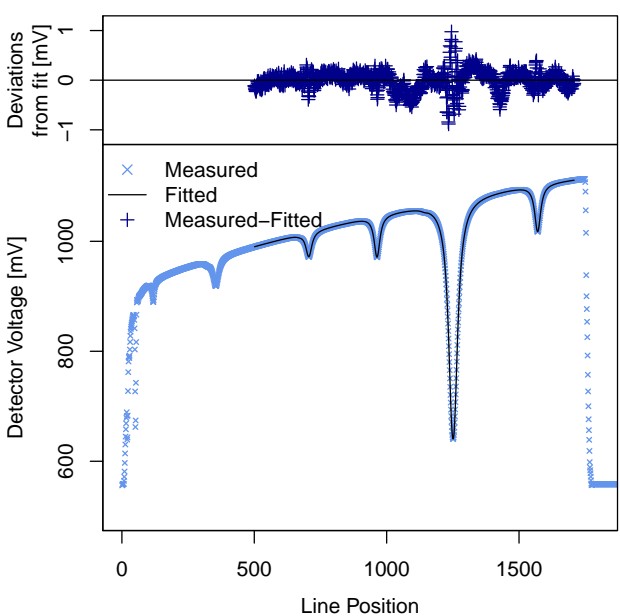

**Figure 1.** Measured and fitted spectrum, as exported from the instrument's operational software QTEGRA.

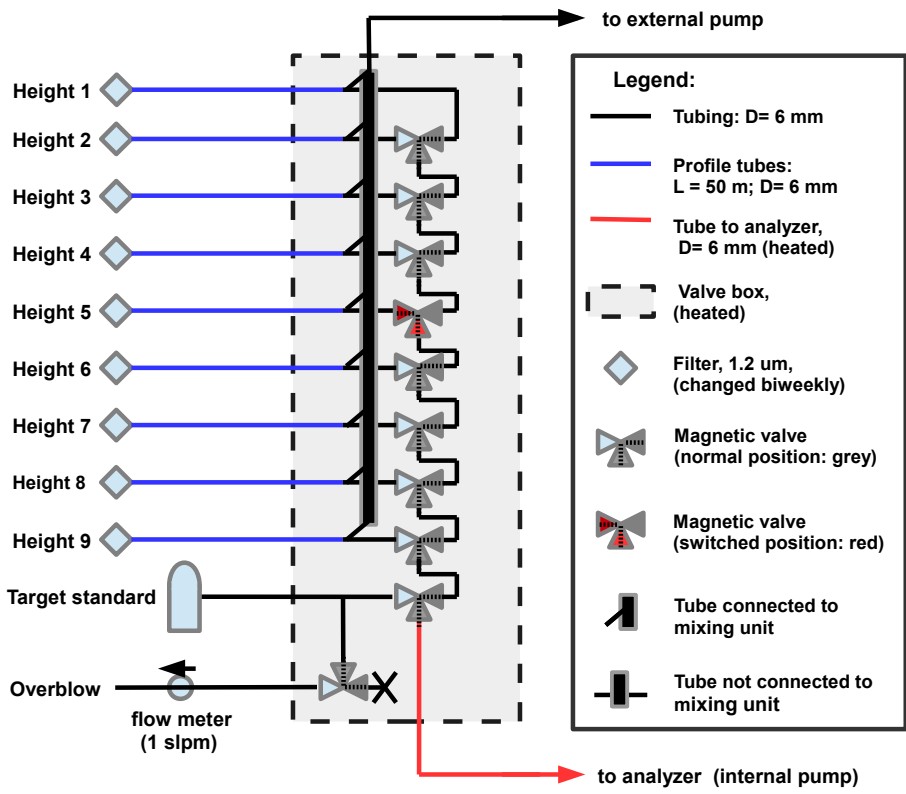

**Figure 2.** Plumbing scheme for the measurements of nine heights and a target standards, the example shows the valve positions when height 5 is sampled.

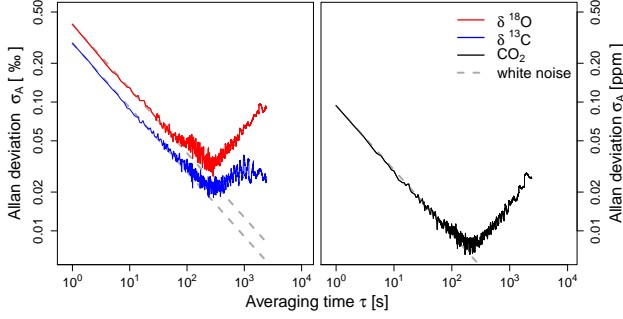

**Figure 3.** Allan deviation $\sigma_A$ in ‰ VPDB for $^{13}$C; in ‰ VPDB-CO$_2$ for $^{18}$O and in ppm for CO$_2$ concentration, solid lines show the calculated Allan deviation and dashed lines show the typical white frequency noise error scaling.

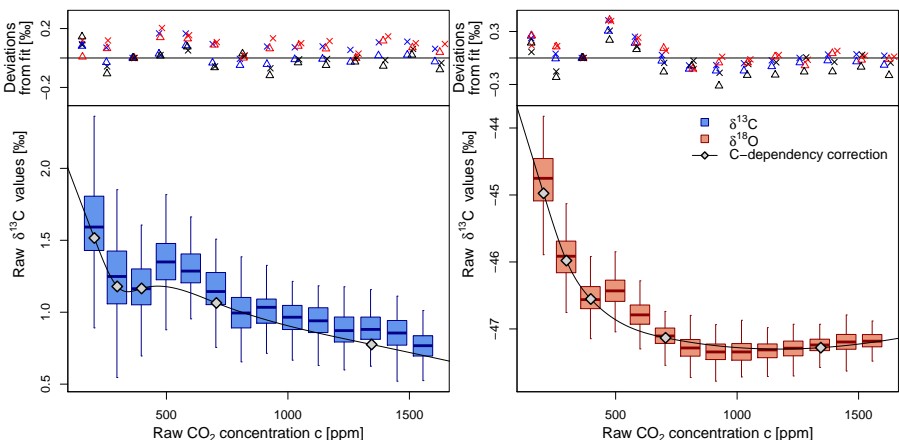

**Figure 4.** Box whiskers plots showing the nonlinear concentration-dependency ($c$-dependency) of raw $\delta$ values for $^{13}$C and $^{18}$O respectively, here as an example for the $CO_2$ tank 'ambient'. The measured $c$-dependency is compared to the respective $c$-dependency correction (black line, with grey symbols marking the data points used during the corresponding calibration measurement). The $c$-dependency correction is Offset-corrected to match the raw $\delta$ values at 400 ppm and the mean deviation from the offset-corrected fit is shown in the top panel for two measurements (different symbols) with three different gas tanks ('ambient' in blue, 'bio' in black and 'bio2' in red).

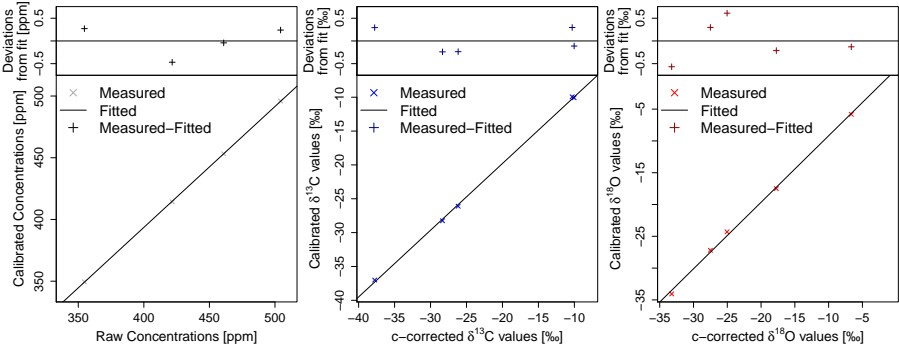

**Figure 5.** Linear calibrations for concentration (left panel) and concentration corrected $\delta^{13}$C respectively $\delta^{18}$O (middle and right panel).

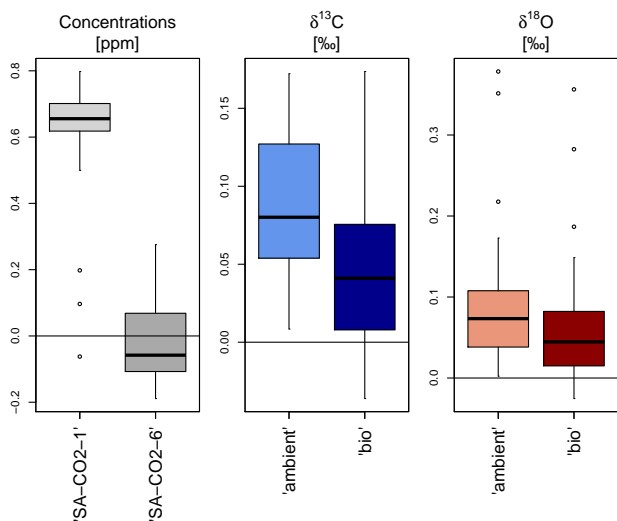

**Figure 6.** Box whiskers plots for the deviations of calibrated concentrations and $\delta$ values from laboratory measurements (at MPI in Jena) for repeated measurements of different calibration tanks (c.f. Table 3 for $c$ and $\delta$ values of the gas tanks) over a period of 9 days (N=36). Delta values were measured at 400 ppm and 'referencing' was done approximately very 30 minutes at 380 ppm to simulate conditions during a measurement campaign.

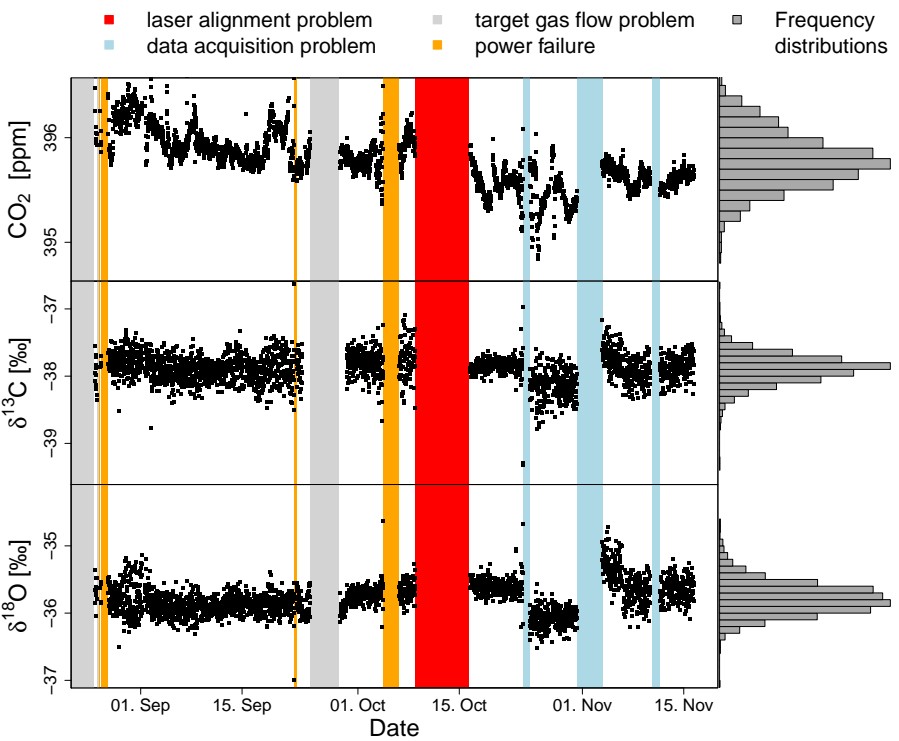

**Figure 7.** Time series and frequency distributions of half-hourly measurements of the concentration (top panel) and $\delta$ values (middle and bottom panel) for target gas 'SA-$CO_2$-5' (c.f. Table 3) for the whole measurement period excluding periods that show problems with target gas flow, calibration and a laser alignment problem. Major reasons for data gaps are marked with different colors.

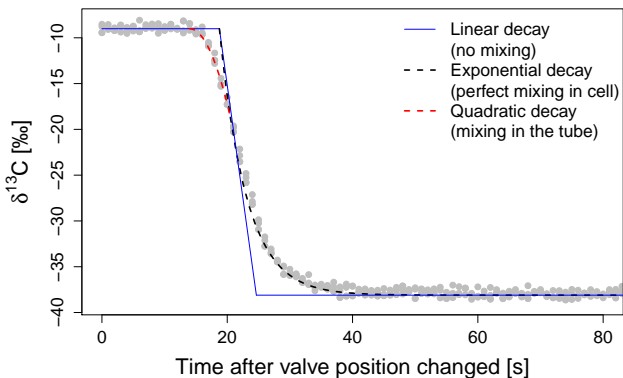

**Figure 8.** The response time of our experimental setup can be divided into three phases with different dominant mechanisms: Directly after switching it took approximately 14 s to flush the tubing, the adjacent 4 s were dominated by the mixing processes in the tubes before the gas entered the measuring cell (quadratic fit) and finally we observed a response behavior that is dominated by mixing processes within the measuring cell (exponential fit) with a characteristic decay time of $\tau_{10\%} = 10$ s for $\delta^{13}$C. These response times were similar for $\delta^{18}$O (not shown). The linear fit shown here describes a first order approximation of the theoretical cell response for the (unrealistic) assumption that there is no mixing in the measurement cell. From this assumption, it can be derived that the $\delta$ values would show a dominantly linear decay with the slope $m = (\delta_{\text{new}} - \delta_{\text{old}})/\tau_{\text{theoretical}}$ with the theoretical instrument cell response time $\tau_{\text{theoretical}} = p * V/\Phi$, with pressure $p$, Volume $V$ and flow rate $\Phi$. In our case $\delta_{\text{new}} - \delta_{\text{old}}$= -29 ‰ and thus $\tau_{\text{theoretical}}$= 5.9 s.

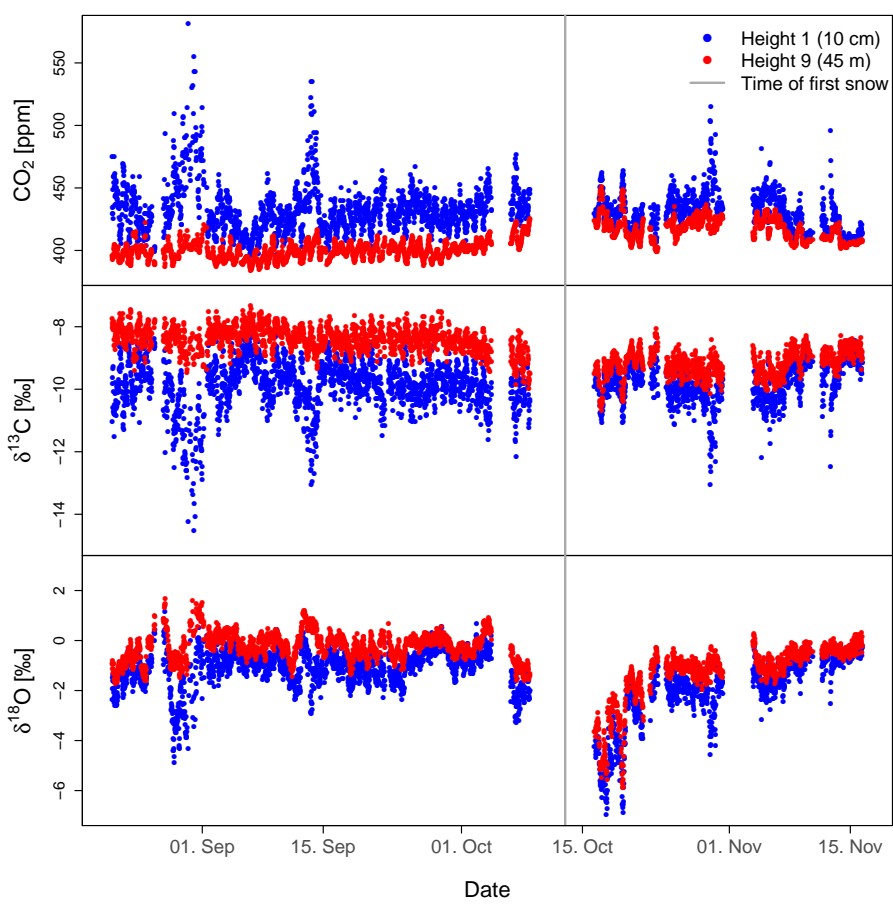

**Figure 9.** Time series of all measured concentrations $c$ and both $\delta$ values at the lowest (blue points) and highest (red points) inlet in 0.1 and respectively 45 m height.

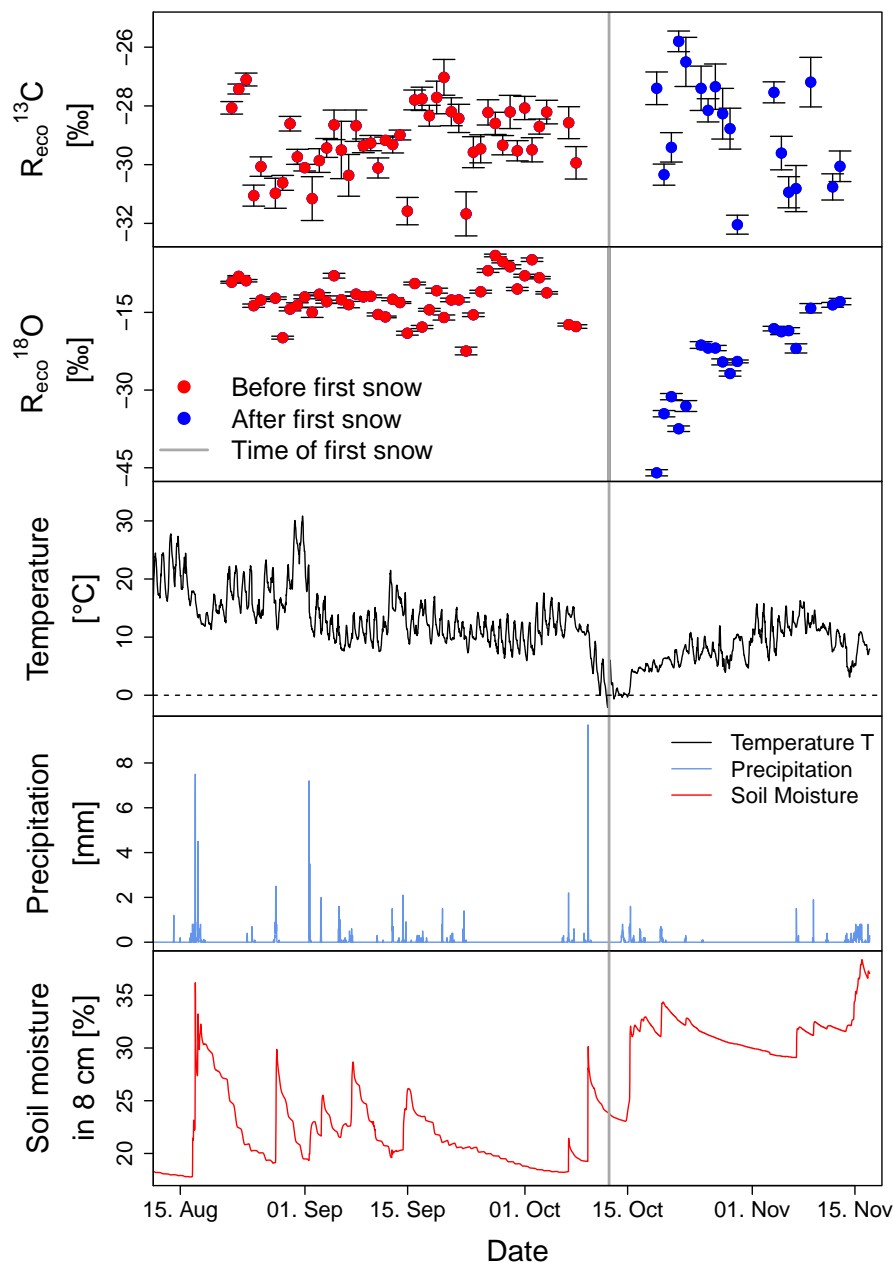

**Figure 10.** Time series of the measured isotopic composition of nighttime $CO_2$ exchange (respiration) $R_{eco}^{13}C$ and $R_{eco}^{18}O$ based on Keeling-Plot intercepts in combination with temperature, precipitation and soil moisture in 8 cm depth. Error bars denote the standard error of the Keeling-Plot intercept (based on the linear regression of $\delta$ vs. $1/c$). A particular feature of this time series is a first snow and frost event on 13. October 2015, marked in gray.

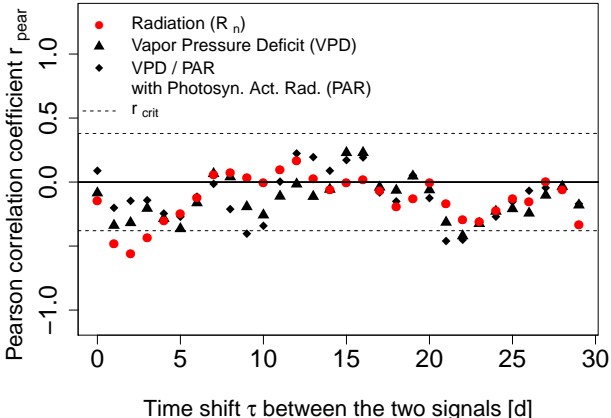

**Figure 11.** Pearson correlation coefficient of the isotopic composition of ecosystem respiration $R_{\mathrm{eco}}^{13}C$ and the two-day-sum of different meteorological variables (shifted by different times $\tau$) before the first snow event in autumn 2015.

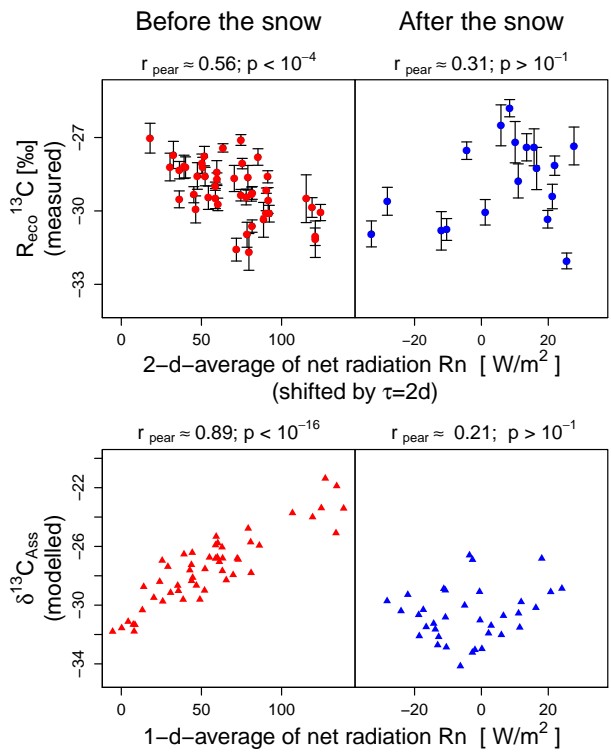

**Figure 12.** Observed relationships between net radiation $R_n$ and the measured isotopic composition of ecosystem respiration $R_{\mathrm{eco}}^{13}C$ (top panels) and the modeled $^{13}C$ composition of assimilated material $\delta^{13}C_{\mathrm{Ass}}$ (bottom panels). Significant correlations were observed before the first snow (left) but became insignificant after the snow (right). $r_{\mathrm{pear}}$ and p values are derived from the respective linear regressions.

**Table 1.** Nomenclature and abbreviations used in this publication, numbers for reference standards $R_{\mathrm{std}}$ from International Atomic Energy Agency (1995)

| Stable isotope specific nomenclature | |
| --- | --- |
| $R_{\mathrm{std}}$ | Isotope ratio $C_{\mathrm{heavy}}/C_{\mathrm{light}}$ of an (arbitrary) reference standard |
| $\delta$ value | Relative deviation of the measured isotope ratio from $R_{\mathrm{std}}$ |
| VPDB | Vienna Pee Dee Belemnite - standard for $^{13}$C ($R_{\mathrm{VPDB}} \approx 0.01124$) |
| VPDB-CO$_2$ | Vienna Pee Dee Belemnite - standard for $^{18}$O ($R_{\mathrm{VPDB-CO2}} \approx 0.0020883$) |

| Nomenclature | |
| --- | --- |
| $\sigma_A$ | Allan deviation |
| $R_n$ | Net radiation |
| RH | Relative humidity |
| VPD | Vapor pressure deficit |
| $\delta_{\mathrm{KP}}$ | Keeling-Plot intercept |
| $R_{\mathrm{eco}}$ | Isotopic composition of nighttime CO$_2$ exchange (respiration) integrated over the ecosystem |

| Technical abbreviations | |
| --- | --- |
| IRIS | Isotope Ratio Infrared Spectrometer |
| IRMS | Isotope Ratio Mass Spectrometer |
| OA-ICOS | Off-axis Integrated Cavity Output Spectroscopy |
| CRDS | Cavity Ring Down Spectroscopy |

**Table 2.** Examples for different optical instruments that measure the isotopic composition of $CO_2$ and reported values for minimal Allan deviations $\sigma_A$ and the corresponding averaging times $\tau_{\min}$ (if available), see also Table 2 of the review of Griffis (2013).

| | |
|---|---|
| **Broadband light source based instruments** | |
| Instrument: | Fourier Transform Infrared Spectrometer: Spectronus analyzer, *Ecotech Pty Ltd., Australia* |
| Minimal Allan deviation for $\delta^{13}C$ : | $\sigma_A(\tau_{\min}\approx6000\,\text{s})=0.01\,\text{‰}$ (Griffith et al., 2012) |
| Minimal Allan deviation for $\delta^{18}O$ : | $\sigma_A(\tau_{\min}\approx7200\,\text{s})=0.1\,\text{‰}$ (Vardag et al., 2015) |
| Instrument: | Fourier Transform Infrared Spectrometer: Nicolet Avatar, *Thermo Electron, USA* |
| Minimal Allan deviation for $\delta^{13}C$ : | $\sigma_A(\tau_{\min}\approx960\,\text{s})=0.15\,\text{‰}$ (Mohn et al., 2007) |
| **Laser based direct absorption spectrometers in mid infrared** | |
| Instrument: | Quantum cascade laser absorption spectrometer: QCLAS *Aerodyne Research Inc., USA* |
| Minimal Allan deviation for $\delta^{13}C$ : | $\sigma_A(\tau_{\min}\approx100\,\text{s})=0.01\,\text{‰}$ (Wehr et al., 2013); $\sigma_A(\tau_{\min}\approx100\,\text{s})=0.06\,\text{‰}$ (Sturm et al., 2012) |
| Minimal Allan deviation for $\delta^{18}O$ : | $\sigma_A(\tau_{\min}\approx100\,\text{s})=0.06\,\text{‰}$ (Sturm et al., 2012) |
| Instrument: | Lead-salt tunable diode laser absorption spectrometer: TGA100A/200, *Campbell Scientific Inc., USA* |
| Minimal Allan deviation: | no data for uncalibrated minimal $\sigma_A$, ideal averaging time $\tau_{\min}\approx30\,\text{s}$ (Bowling et al., 2003c) |
| High frequency Allan deviation: | $\sigma_A(\tau=0.1\,\text{s})=1.5\,\text{‰}$ for $\delta^{13}C$ and $2.2\,\text{‰}$ for $\delta^{18}O$ (Bowling et al., 2003c) |
| Instrument: | Isotope Ratio Infrared Spectrometer: Delta Ray, *Thermo Scientific Inc., USA* |
| Minimal Allan deviation for $\delta^{13}C$ : | $\sigma_A(\tau_{\min}\approx500\,\text{s})=0.04\,\text{‰}$ (Geldern et al., 2014); $\sigma_A(\tau_{\min}\approx300\,s)=0.02\,\text{‰}$ (this study, table 5) |
| Minimal Allan deviation for $\delta^{18}O$ : | $\sigma_A(\tau_{\min}\approx300\,\text{s})=0.04\,\text{‰}$ (this study, table 5) |
| **Laser based path length enhanced absorption spectrometers in near infrared** | |
| Instrument: | Cavity Ringdown Spectrometer: G1101-i+, *Picarro Inc., USA* |
| Minimal Allan deviation for $\delta^{13}C$ : | $\sigma_A(\tau_{\min}\approx3600\,\text{s}) \leq 0.1\,\text{‰}$ (Vogel et al., 2013) |
| Instrument: | Off-Axis Integrated Cavity Output Spectrometer: CCIA DLT-100, *Los Gatos Research Inc., USA* |
| Minimal Allan deviation for $\delta^{13}C$ : | $\sigma_A(\tau_{\min}\approx200\,\text{s})=0.04\,\text{‰}$ (at approximately $20\,000\,\text{ppm}\,CO_2$) (Guillon et al., 2012); $\sigma_A(\tau_{\min}\approx200\,\text{s})=0.6\,\text{‰}$ (at approximately $2\,000\,\text{ppm}\,CO_2$) (Guillon et al., 2012) |
| **Laser based path length enhanced absorption spectrometers in mid infrared** | |
| Instrument: | Quantum cascade laser absorption spectrometer: CCIA-48, *Los Gatos Research Inc., USA* |
| Minimal Allan deviation for $\delta^{13}C$ : | $\sigma_A(\tau_{\min}\approx300\,\text{s})=0.06\,\text{‰}$ (Oikawa et al., 2017) |
| Minimal Allan deviation for $\delta^{18}O$ : | $\sigma_A(\tau_{\min}\approx300\,\text{s})=0.04\,\text{‰}$ (Oikawa et al., 2017) |

**Table 3.** Known $CO_2$-concentrations $c$ and $\delta$ values for gas tanks used for calibration and instrument performance measurements. All measured concentrations and $\delta$ values refer to measurements that were done at Max-Planck Institute for Biogeochemistry in Jena and the $\delta^{13}C$-values of the two pure $CO_2$ tanks. The pure $CO_2$ tanks 'bio' and 'ambient' were additionally measured with IRMS at Geoscience Center in Göttingen (Isotope Geology Division, Göttingen University) for their $^{13}C$ composition. Abbreviations for the purpose of the tanks: cCAL=concentration calibration; dCAL=$\delta$-calibration; REF=referencing; EC=evaluating calibration ; pcCAL=post concentration calibration; REP=repeatability measurement

| Gas tank | Used for | $c$ [ppm] | $\delta^{13}C$ [‰VPDB] | $\delta^{18}O$ [‰VPDB-$CO_2$] |
|---|---|---|---|---|
| Pure $CO_2$ 'ambient' | dCAL, REF, EC | - | $-9.94 \pm 0.01$ | $-17.5 \pm 0.3$ |
| Pure $CO_2$ 'bio' | dCAL, EC | - | $-28.25 \pm 0.01$ | $-27.2 \pm 0.3$ |
| Pure $CO_2$ 'bio-2' | EC | - | $-26.1 \pm 0.3$ | $-24.3 \pm 0.3$ |
| Pressurized air 'PA-tank' | pcCAL, EC | $413.7 \pm 0.2$ | $-9.7 \pm 0.2$ | $-5.3 \pm 0.4$ |
| Syn. air with $CO_2$ 'SA-$CO_2$-1' | cCAL, pcCAL, EC | $349.5 \pm 0.1$ | $-37.01 \pm 0.02$ | $-34.1 \pm 0.4$ |
| Syn. air with $CO_2$ 'SA-$CO_2$-2' | cCAL, pcCAL | $453.9 \pm 0.1$ | $-36.98 \pm 0.02$ | $-34.2 \pm 0.6$ |
| Syn. air with $CO_2$ 'SA-$CO_2$-3' | pcCAL | $349.6 \pm 0.1$ | $-37.02 \pm 0.01$ | $-34.3 \pm 0.4$ |
| Syn. air with $CO_2$ 'SA-$CO_2$-4' | pcCAL, EC | $453.2 \pm 0.1$ | $-37.02 \pm 0.02$ | $-34.8 \pm 0.4$ |
| Syn. air with $CO_2$ 'SA-$CO_2$-5' | pcCAL, REP | $396.5 \pm 0.1$ | $-37.02 \pm 0.02$ | $-34.7 \pm 0.2$ |
| Syn. air with $CO_2$ 'SA-$CO_2$-6' | EC | $496.0 \pm 0.1$ | $-37.02 \pm 0.02$ | $-34.8 \pm 0.1$ |

**Table 4.** Validation of the multilayer model CANVEG using Eddy Covariance measurements of gross primary productivity GPP, net ecosystem exchange NEE, latent and sensible heat flux LE and H. Slopes, $R^2$ values and normalized standard error estimates NSEE of linear regressions between modeled and measured values are comparable to the numbers given by (Knohl and Baldocchi, 2008).

| | SLOPE | $R^2$ | NSEE |
|---|---|---|---|
| GPP | 0.92 | 0.90 | 0.26 |
| NEE | 0.97 | 0.92 | 0.28 |
| LE | 1.03 | 0.78 | 0.16 |
| H | 0.96 | 0.87 | 0.37 |

**Table 5.** Allan deviation $\sigma_A$ for different averaging times $\tau$, with the minimum Allan deviation for $\tau_{min} \approx 290\,s$ for both $\delta$-values and $170\,s$ for $CO_2$ concentration $c$

| $\tau$ [s] | $\delta^{13}C$ [‰] | $\delta^{18}O$ [‰] | $c$ [ppm] |
|---|---|---|---|
| 1 | 0.29 | 0.40 | 0.09 |
| 20 | 0.06 | 0.09 | 0.02 |
| 80 | 0.03 | 0.05 | 0.02 |
| $\tau_{min}$ | 0.02 | 0.03 | 0.007 |
| 500 | 0.03 | 0.04 | 0.01 |
| 1800 | 0.03 | 0.08 | 0.01 |

**Table 6.** Uncertainties related to the different calibration steps and their repeatability defined as $1\sigma$ standard deviation of the respective calibration step.

| Calibration | $\delta^{13}C$ [‰] | $\delta^{18}O$ [‰] | $c$ [ppm] |
|---|---|---|---|
| Linear calibrations | 0.24 | 0.3 | 0.45 |
| corresponding repeatability | 0.05 | 0.1 | 0.2 |
| Correction of $c$-dependency | 0.2 | 0.4 | - |
| corresponding repeatability | 0.15 | 0.15 | - |

**Table 7.** Standard deviations $\sigma$ of the measured (calibrated) $\delta$ values over a large concentration range based on 6 hourly lab measurements over a period of nine days.

| Concentration ppm | $\sigma(\delta_{\mathrm{meas}} - \delta_{\mathrm{tank}})$ ‰ | | | |
|---|---|---|---|---|
| | tank: 'ambient' | | tank: 'bio' | |
| | $^{13}$C | $^{18}$O | $^{13}$C | $^{18}$O |
| 202 | 0.07 | 0.14 | 0.09 | 0.13 |
| 396 | 0.04 | 0.05 | 0.08 | 0.08 |
| 600 | 0.09 | 0.08 | 0.12 | 0.12 |
| 807 | 0.08 | 0.08 | 0.11 | 0.11 |
| 1018 | 0.10 | 0.08 | 0.13 | 0.11 |
| 1232 | 0.12 | 0.09 | 0.13 | 0.11 |
| 1450 | 0.14 | 0.11 | 0.15 | 0.12 |
| 1664 | 0.14 | 0.11 | 0.14 | 0.12 |
| 3145 | 0.17 | 0.15 | 0.17 | 0.15 |

**Table 8.** Percentage of total measurement time for major data gaps. The latter two data gaps concerned only target gas measurements.

| Reason for data gap | Percentage |
|---|---|
| Data acquisition problems | 6.0 % |
| Laser alignment problem | 8.6 % |
| Calibration | 1.5 % |
| Power failures | 3.3 % |
| Additional measurements | 1.6 % |
| Plumbing issues (only target) | 9.5 % |
| Switching unit failure (only target) | 0.7 % |

**Table 9.** $R^2$ values for correlations between the $^{18}O$ composition of nighttime $CO_2$ exchange $R^{18}_{eco}O$ and different meteorological variables. Significance thresholds are given by *** for $p<10^{-4}$; ** for $p<10^{-3}$ and * for $p<10^{-2}$. For some parameters the height above the ground (with negative values indicating the depth below the ground) is given in brackets, the parameters without such indication are measured 42 m above the ground.

| | All periods | Before the snow | After the snow |
|---|---|---|---|
| Soil moisture (-8 cm) | 0.49 *** | 0.04 | 0.00 |
| Upwards shortwave radiation | 0.40 *** | 0.28 * | 0.04 |
| VPD | 0.18 ** | 0.09 | 0.22 |
| Soil temperature (-8 to -64 cm) | 0.36 *** | 0.06 | 0.70 *** |
| Air Temperature | 0.22 ** | 0.02 | 0.61 ** |
| Air Temperature (2 m) | 0.21 ** | 0.05 | 0.60 ** |
| Upwards longwave radiation | 0.20 ** | 0.02 | 0.61 ** |
| Incoming longwave radiation | 0.05 | 0.49 *** | 0.03 |
| Ambient pressure | 0.05 | 0.39 *** | 0.36 * |
| Incoming shortwave radiation | 0.39 *** | 0.23 ** | 0.13 |
| Dewpoint temperature | 0.02 | 0.38 *** | 0.14 |
| Specific humidity | 0.02 | 0.34 *** | 0.17 |
| $H_2O$ concentration | 0.02 | 0.34 *** | 0.17 |
| Actual vapor pressure | 0.02 | 0.33 *** | 0.18 |
| Relative humidity | 0.28 *** | 0.31 *** | 0.15 |
| Rain | 0.01 | 0.28 ** | 0.05 |

## Appendix A: Measures to improve data quality

To reduce the uncertainty of the calculated isotopic composition of nighttime $CO_2$ exchange (respiration) $R_{eco}^{13}C$ and $R_{eco}^{18}O$, we used the following approaches concerning setup and post-processing.

– **Minimizing the sampling time**

One of the key assumptions of the Keeling-Plot approach Eq. (1) is the mixing of a constant background with one (integrated) source. This assumption is justified if there is no significant change in the background concentration $c_{bg}$, its isotopic composition $\delta_{bg}$, and the isotopic composition of the (integrated) source $\delta_s$ for all data points that are taken into account for a single Keeling-Plot. For the case of an integrated source, a constant $\delta_s$ can be ensured when the isotopic composition of the individual source components $\delta_{s,i}$ as well as the relative contribution of the individual source components $\alpha_i$ in Eq. (2) are constant. As in general all these quantities ($\delta_{s,i}$, $\alpha_i$, $c_{bg}$ and $\delta_{bg}$) can vary with time, this assumption tends to be violated stronger for longer measurement times. Thus, the uncertainty of calculated Keeling-Plot intercepts can be reduced by minimizing the measurement time, as discussed e.g. by Bowling et al. (2003b), who recommend to use only measurements that took less than five hours for analyzing Keeling-Plot intercepts for $\delta^{18}O$. As our setup measures all the nine heights within 30 minutes, we were able to calculate Keeling-Plots for shorter periods. During data analysis we calculated Keeling-Plots on timescales between 30 min and 5 h.

– **Increasing the $CO_2$ concentration range**

The linear regression that underlies the Keeling-Plot, can be improved significantly by increasing the $CO_2$ concentration (Zobitz et al., 2006). In our setup, we increase the $CO_2$ concentration range by using data from all nine inlet heights within one Keeling-Plot, but this, on the other hand, could violate the assumption of constant relative contributions of the individual source components $\alpha_i$ in Eq. (2) to the integrated source. To analyze if there is any bias (which may have several contributions) due to the inclusion of the different inlet positions, we evaluated the Keeling-Plots for the lower inlets (heights 1-4) and for all all inlets (heights 1-9) separately. The difference $\Delta$ between the these Keeling-Plot intercepts based on different data sets showed a symmetric frequency distribution around 0 (Fig. S1 in the supplementary material). By including all heights into the data analysis, we reduced the error of the intercept $\sigma$ from a mean value of $\overline{\sigma_{low}} \approx 1.5‰$ to $\overline{\sigma_{all}} \approx 0.8‰$ for both isotopic species. These numbers refer to Keeling-Plots that include data from three consecutive measurement cycles, yielding a temporal resolution of 90 min. Reasons for the choice of this time resolution are given below.

– **Performing an ordinary Model I regression instead of a Model II regression**

We used an ordinary Model I regression instead of a Model II regression. According to Zobitz et al. (2006), this approach takes into account that the error of the measured $\delta$-values dominates over the error of the measured concentrations and yields unbiased estimates of the Keeling-Plot intercept. In our setup, the application of a Model I regression can be justified by the fact that the relative precision of $\delta$ measurements is more than an order of magnitude larger than the relative precision of the $CO_2$ concentration measurements: To estimate the relative precision of the three measured quantities, we calculated the ratio of Allan Deviation at our measurement time of 20 s over the typical range of $c$, $\delta^{13}C$ and $\delta^{18}O$. The typical range, we further define as the median of the ranges that were obtained during the 30 minutes measurement cycles. Thus, with the Allan deviations in Table 5 and with typical ranges of 26 ppm, 1.5‰ and 1.1‰ for $c$, $\delta^{13}C$ and $\delta^{18}O$, the relative precision for the obtained variability in $CO_2$, $\delta^{13}C$ and $\delta^{18}O$ is in the order of $10^{-3}$, $10^{-2}$ and $10^{-1}$, respectively. Thus, the relative precision of the concentration measurement is at least an order of magnitude better than the relative precision of $\delta$ measurements.

- **Filtering data to get only high quality linear regressions**

    Data filtering to remove bad quality and biased (Model II) linear regressions has been often done by excluding data with a to low $CO_2$ concentration range (Pataki et al., 2003; Bowling et al., 2005). Whereas Pataki et al. (2003) recommend to exclude all data from the analysis that spans a $CO_2$ range below 75 ppm, Bowling et al. (2005) choose this threshold to be 40 ppm. This data filtering approach,

based on $CO_2$ concentration range, does not seem necessary when applying a Model I regression: Zobitz et al. (2006) analyzed consequences of small $CO_2$ concentration ranges numerically as well as analytically and conclude that for Keeling-Plot intercepts based on Model I regressions 1) a bias at low $CO_2$ concentration ranges is not expected at current analytical error levels and 2) that errors in the intercept can be small, even for small $CO_2$ concentration ranges if the $\delta$-values are measured accurately enough. Figure S2 in the supplementary material shows the relationship between $CO_2$ concentration range and the standard error of the intercept $\sigma$

for a measurement period of 30 minutes. This figure also shows two comparable approaches for data filtering that both accept 85 % of the data: One approach would be to directly remove data with large intercept errors, and the other approach, as mentioned above, is to remove data with to low $CO_2$-range. As visible in Fig. S2 in the supplementary material, this approach would remove considerable amounts of data with a very small standard error of the intercept $\sigma$, which might be good quality data. For this reason (and as we do not expect a bias occurring for small $CO_2$ concentration ranges for our Model I type regression), we decide for a direct filtering based

on a $\sigma$-threshold and used the 85 % data points with the smallest standard error $\sigma$. The filtered nighttime Keeling-Plot intercepts based on 90 minutes of data acquisition had $R^2$ values with a median of 0.87 and 0.81 for $^{13}C$ and $^{18}O$ with mean values of 0.85 and 0.77 and standard deviations $\sigma$ of 0.1 and 0.16 respectively. Example Keeling-Plots with $R^2$ values spanning the range of mean $\pm 1\sigma$ are provided in the supplementary material (Figure S5 in the supplementary material).

- **Removing outliers**

Our set-up, based on the measurement of $\delta^{13}C$, $\delta^{18}O$ and $CO_2$ concentration $c$, enabled us to calculate individual Keeling-Plots based on all inlet heights (heights 1-9) with a temporal resolution of 30 min. We calculated Keeling-Plots on different timescales ranging from 30 min to 5 h by using one to ten measurement cycles and evaluated how the Keeling-Plot intercepts $\delta^{13}C_{KP}$ and the corresponding standard errors of the linear regression $\sigma$ changed (Fig. S3 in the supplementary material). Additionally, the calculation of Keeling-Plot intercepts based on longer timescales increased the number of Keeling-Plot intercepts within reasonable ranges. For

Keeling-Plots that were averaged over 2h (5h), a fraction of 97% (99%) of the Keeling Plot intercepts were between -33 and -25‰. Because the range of the Keeling-Plot intercepts should not depend on the chosen timescale, we considered the Keeling-Plot intercepts that were outside of this range as outliers and removed them from further analysis (also for Keeling-Plot Intercepts that were based on shorter timescales).

- **Choosing a time resolution for individual Keeling-Plots**

To decide for a suitable time resolution to analyze the temporal variability of the Keeling-Plot intercepts, we had to solve the trade-off between 1) more accurate data on longer timescales and 2) a larger number of data points that were available (after the above mentioned filtering procedures). We decided to fit the individual Keeling Plots on 90 min resolution, which yields a maximal number of $N_{filtered} \approx 2300$ accepted data points and a median of 0.76‰ for the standard errors $\sigma$ (Fig. S4 in the supplementary material).

- **Calculation of weighted means for nighttime data**

For analyzing variations in the nighttime $CO_2$ exchange (respiration) $R_{eco}$ on seasonal timescales we used the (filtered) individual Keeling-Plots, each based on 90 minutes of input data, and calculated the mean over all Keeling-Plots that were collected between 21h30 and 2h30 (using the weight $w$ based on with the standard error $\sigma$ of the Keeling-Plot intercept: $w = 1/\sigma^2$).