# Peer review of "A new instrument for stable isotope measurements of $^{13}C$ and $^{18}O$ in $CO_2$ - Instrument performance and ecological application of the Delta Ray IRIS analyzer"

_Atmospheric Measurement Techniques, 2017_

## Referee Comment (RC1) · Anonymous Referee #1 · 25 Jun 2017

The paper "A new instrument for stable isotope measurements of 13C and 18O in CO2 - Instrument per-formance and ecological application of the Delta Ray IRIS analyzer", by Braden-Behrens et al, reports on a recent commercial instrument for measurements of stable CO2 isotopes (d13C-CO2 and d18O-CO2) and its application in a field study. This work is relevant, however it is lacking many essential elements and it is not written carefully enough. The manuscript will thus need a major revision to be considered for publication in AMT.

[Figure]

Please also note the supplement to this comment:
http://www.atmos-meas-tech-discuss.net/amt-2017-120/amt-2017-120-RC1-supplement.pdf

———————————————————

[Figure]

**Supplement:**

Comments to manuscript AMT-2017-120

The paper "A new instrument for stable isotope measurements of $^{13}$C and $^{18}$O in $CO_2$ - Instrument performance and ecological application of the Delta Ray IRIS analyzer", by Braden-Behrens et al, reports on a recent commercial instrument for measurements of stable $CO_2$ isotopes ($\delta^{13}$C-$CO_2$ and $\delta^{18}$O-$CO_2$) and its application in a field study.
This work is relevant, however it is lacking many essential elements and it is not written carefully enough. The manuscript will thus need a major revision to be considered for publication in AMT.

This review focuses mainly on the performance evaluation of the instrument. It is important to also carefully review the hypothesis and the conclusions drawn from the field study.

Most remarks and suggestions are added to the original manuscript. Some mayor aspects are discussed below.

p3/22 The exact wavelength should be given together with a measured and fitted spectrum. The spectral range and the spectral resolution are important elements to judge the analytical performance, also in the context of gas matrix effects. One may assume that the frequencies are as in (Geldern 2014), but the latter does not show a measured spectrum.

Water vapor may significantly impact the retrieved $\delta$-values, either through spectral interference or through changes in absorption line characteristics (pressure broadening). If my understanding of the setup is correct, then humid samples were measured spectroscopically. Since this paper aims at validating a new spectrometer, it is vital to discuss and quantify the effect of changes in humidity.

p4/20 physically different samples: There is no indication that the instrument was used in a batch mode configuration. In continuous flow mode (as the text suggests), mixing in the cell (and to some extend in the tubing) corresponds to a low-pass filter, which is fundamentally different to "physically different samples".

p4/20 "temporal stability" is not standard terminology and only used once in this paper. I suggest using "repeatability", following the international vocabulary of metrology (VIM) throughout the text.

p6/20 Accuracy was tested by comparing with one (1) gas tank which was measured using an Aerodyne spectrometer. This part of the study is a key element and completely insufficient. The main challenge in laser spectroscopy is currently not (any more) precision but rather accuracy. There is no reason why anyone should trust another spectrometer (here Aerodyne) without a very detailed description of how the latter achieves traceability. Furthermore, accuracy will depend on at least two calibration scales, i.e. delta values and concentration. Therefore, the evaluation must (!) include measurements of traceable (likely IRMS) gases at different delta values and concentrations; otherwise it is an insufficient and somewhat random exercise. If this is not possible, then an alternative may be to use traceable standards and (!) field samples that are quantified in a traceable way. This is easily possible for $\delta^{13}$C-$CO_2$, but more difficult for $\delta^{18}$O-$CO_2$ because of the limited stability of the samples (see e.g. Tuzson 2007, DOI: 10.1007/s00340-008-3085-4).

Along the same line: p3/23 describes the DeltaRay having "an internal calibration procedure that automatically includes two point calibrations for concentration c and both $\delta$ values as well as corrections for the concentration dependency of the measured d-values". This concept is interesting and a key feature of the Delta Ray. However, since this publication evaluates a commercial instrument, it should clearly describe the way concentration dependency is corrected (and how large it is) and to validate the procedure (accuracy, see above). This has not been achieved or is not presented.

p6/26 Measurement of the Allan plot was done in the lab because of limited gas supply in the field. This is not sufficient, because the goal of this study is characterization under field conditions. The argument of limited gas in the field is not convincing because at 80 ml/min, it would easily be possible to have many corresponding measurements of about 10 - 30 minutes, which, given an Allan Minimum at around 100 s, would be sufficient. A minimal approach would be to evaluate the 80 s target gas measurements. Alternatively, or in addition, one may use ambient conditions that are sufficiently stable (e.g. well mixed, afternoon, highest sampling port) to obtain at least a conservative estimate for the

precision in the field. Finally, data from the PA tank measurement also give an indication of precision in the field.

p7/30 Referencing was done at the concentration of the highest sampling port. Discuss the uncertainty resulting from the fact that some height had other concentrations, taking into account the "linearity calibration" (which does not test linearity but dependence of the retrieved $\delta$ values on c; a terminology that should be improved).

p8/1 This whole chapter is badly written and should be revised with respect to language. In addition, the arguments are not convincing. The concentration range of HS and LS is not any larger than the standards used in the first calibration (300 and 430 ppm). Choosing two out of five standards, that were meant to evaluate accuracy for calibration, leads to only three remaining standards that are perfectly bracketed. The mean and uncertainty at N=3 becomes then statistically very weak. Furthermore, the results for c also illustrate why using just one tank to assess the accuracy of the $\delta$ values is not sufficient and somewhat arbitrary (see p6/20 above).

p9/15: The authors state that they chose an averaging time of 20 s as compromise between number of measurements and precision. This is misleading or not clear enough. If there are no measurements of standards between 20 s intervals, then the precision does not mean much because the next mean value for 20s may have an excellent precision (given as SD) but may have drifted significantly, thus the two values with good precision cannot be compared at the level of their individual precision (it then becomes an issue of repeatability or accuracy, depending on the context).

p9/20 "the mean deviation of N=300 field measurements of a tank with pressurized air" is a suitable way to quantify repeatability and should be compared (or moved) to the results found in the corresponding chapter 3.1.3. Unfortunately, the values are only given graphically in Fig. 3. However, looking at the difference of one (!) sample, one cannot determine accuracy of the spectrometer. Especially not for an analytical technique which is known to be strongly dependent on concentration and gas matrix. The test is thus not suited for its aim. This chapter and the next can be combined to determine repeatability (preferred terminology), and which - at least in the title - may be called long-term stability. However, it is critical to find a way to reliably determine accuracy.

p9/25 "sum of uncertainties". What the authors likely mean is the combined uncertainty or an uncertainty budget. However, this is not achieved by simple addition of the uncertainties, as suggested in the text. It is necessary to know what the authors consider for the individual uncertainty contributions (and why), what distribution they assume and – if the contributions are independent – how they calculate the combined uncertainty, and at what level of confidence they then express this combined uncertainty.

P10/7 The standard deviations of repeated measurements (0.2 ppm for $CO_2$ concentration and below 0.3‰ delta values) should be compared to literature values. For example, (Sturm 2013, amt-6-1659-2013) found repeated measurements of the same gas tank with a standard deviation which is a factor 4-7 better than the results shown here.

Fig 5 What is the slope of the linear decay, and what process does it represent?

[revised manuscript text omitted]

---

## Referee Comment (RC2) · Anonymous Referee #2 · 27 Jun 2017

This paper describes the testing and use of the new Delta Ray IRIS CO2 isotope spectrometer during a three-month field campaign. I have serious concerns about the analytical details, as well as the conclusions regarding interpretations of the field measurements. Overall, I am not convinced that this instrument has been put through the necessary rigorous tests.

The authors conclude in the abstract that "1) the new Delta Ray IRIS with its internal calibration procedure provides an opportunity to precisely and accurately measure c, $\delta$13C and $\delta$18O at field sites" I am concerned with this statement, because the internal

calibration procedure in the IRIS is never actually described. How are the absorption spectra used to calculate isotope ratios, and how are these modified based on the calibration? This point appears critical for understanding whether the internal procedure is adequate and/or necessary, or for understanding what other post-hoc calibrations may be needed. This is a critical gap in the paper. Once cannot simply assume that the manufacturers of the instrument have worked out the details here. There are instruments that are sold that do not necessarily function as advertised, thus it is necessary to validate every step of the way. I would like to see plots and regressions of raw vs. known values for both $\delta$13C and [CO2] for a number of different standards spanning a broad range of delta values and mole fractions of CO2.

The authors mention that they used a post-hoc CO2 concentration calibration, but it is unclear how often the additional standards used for this were measured (once? Half-hourly?) in relation to their check standard. Note that quadratic relationships may give a better fit—as employed elsewhere for other absorption-based CO2 instruments. Given that this is a methods paper, it would have been very useful to see tests using a broader range of CO2 mole fraction and isotope compositions in the range of standards, and to see more standards tested. Without this, we cannot validate the linearity of the instrument both in concentration and isotope space. This is a critical deficit of the paper.

Why was the need for a post-hoc $\delta$13C and $\delta$18O calibration not tested or described? Note that many of the other laser-based isotope instruments achieve much higher precision with frequent (e.g. 20 minute) isotope calibrations in the field. This need appears especially critical here given the large ($\sim$1 per mil) jumps in $\delta$13C values observed in the check standards shown in Figure 4. This suggests that there are some serious stability problems that need to be addressed with more frequent isotope calibration. With respect to the second major conclusion of the abstract, "2) even short snow or frost events could have strong effects on the isotopic composition of CO2 exchange at ecosystem scale" this finding is not new, but also not very well supported by the data

(e.g. Figs 7 and 8. There are now several multi-year datasets of canopy CO2 and $\delta$13C profiles in temperate ecosystems that have shown similar patterns.

With respect to the Keeling plot intercepts, no data is shown to actually validate the
approach (e.g. plots of $\delta$13C and 1/CO2 space), nor summary statistics presented for these regressions. This is another serious deficit given the key methodological issues the authors point out in the Appendix, but do not quantify in the text. I don't think the authors present enough information here to rigorously test the hypotheses proposed in the Results/Discussion section. The value of the CANVEG modeling exercise for the overall study was not terribly apparent to me, nor were the questions that it sought to address.

More specific comments: Introduction: there is much excessive detail here that repeats recent reviews, such as the Griffis 2013 paper. Please condense. P1 18: the main constraint is low temporal resolution P4 13: how are these "physically different" air samples if the pump is flowing continuously? P8 5: "A possible reason for this resulting deviation is the range of the gas tanks we used for the instrument-internal concentration calibration, that was approximately 300 to 430 ppm" this logic doesn't make sense to me—this is similar to your other standards P8 6: I am having trouble understanding how your "target standard" could be stable without posthoc calibration yet your five other standards were so variable. P8 9: "Secondly we set the IRIS analyzer's internal referencing procedure (described in Sect. 2.7) to 1800 s which corresponds to an Allan variance of 0.03 ‰ for both $\delta$ values and 0.01 ppm for CO2 concentration." This is unclear to me—are you measuring the standards every 1800 s? For how long? Where are these new Allan variance values coming from? Figure 4: There are apparently large (1 per mil) jumps in measured "target gas" isotope values at several points—these are disconcerting. Are the data shown in this figure the raw values or the calibrated values? If they are the calibrated values, this suggests that the two-point calibration employed here is inadequate

---

## Referee Comment (RC3) · Anonymous Referee #3 · 13 Jul 2017

Interesting paper but needs some major revisions. Please find below some listed points that should be changed or at least answered.

1a) Page 2 lines 13ff: text passage about IRMS: Pls cite Schnyder et al. there (citation below) 1b) in the same text passage: I think "sample preparation effort and cost" might be a minus for IRMS techniques. But here the main disadvantages should be mentioned like (storage) problems with vials (see Gemery et al., 1996 and Knohl et al., 2004) and the advantage of quasi-continuous measurements relative to the "discontinuous" measurement by IRMS. 2) Page 2 lines 22ff. text passage about different

spectrometer types: should be shortened as this manuscript is not a review on optical methods for measurement of isotope ratios 3) Page 3 lines 25ff: "to characterize the Delta Ray IRIS and its performance under field conditions": I think measurement of the "internal cell turnover" and "Allen deviation" is not sufficient to fulfill this topic here. The reference gas box from the Delta Ray is said to offer possibilities to adjust $CO_2$ conc of the "reference" gas to the measured $[CO_2]$ to cancel out a possible concentration effects on the measured d-values. The authors need to go more in detail here by showing data (!) from multiple $CO_2$-in air-standards with different $[CO_2]$ and different d13C- and d18O values measured with IRMS (preferred) in comparison to measurement with Delta Ray or a comparison with different optical measurement devices (more problematic). I suppose you have measured the data, so show them here please. 4) Please give more info (citation if available) on the kind of measurements performed at the MPI in Jena (isotopes and concentration). 5) The link to VPDP was done with the gas tank measured in Jena? Please extend the info on how this is done. Fig. 3 describes your quality control standard? Is there a way to compare measured values (+ stdev.) with a target value (+stdev.)? 6) Page 3 line 26 "b)" please add one or two sentences why R13Ceco and R18Oeco is interesting. 7) Page 11 line 21 "lighter" here means only 13C-depleted or also 18O-depleted ? Please specify (also in whole manuscript) 8) Page 13 line 26: more "enriched" in what? Please check that also in whole manuscript, depleted in 13C, enriched in 18O . . . (page 14 line 21 . . .) 9) I'm not totally happy to read a manuscript with 2 hypotheses where one hypothesis can be discarded but the 2nd one cannot be proven. The authors should find a way around this, at least the additional measurements for finally testing should be mentioned and discussed here 10) the unit "‰´ is not conform to the SI unit system, what about using "mUr"? It might be more a editorial decision . . .

Gemery et al. (1996): Oxygen isotope exchange between carbon dioxide and water following atmospheric sampling using glass flasks. J Geophys Res 101, D9, 14514-14420. Knohl et al. (2004): Kel-FTM discs improve storage time of canopy air samples in 10-mL vials for CO2-d13C analysis. Rapid Comm Mass Spectrom. 18, 1663-1665.

Schnyder et al. (2004): Mobile, outdoor continuous-flow isotope-ratio mass spectrometer system for automated high-frequency 13C- and 18O-CO2 analysis for Keeling plot applications. Rapid Comm Mass Spectrom. 18, 3068-3074.
* * *

---

## Author Comment (AC1) · 17 Aug 2017

**Author's reply to the referees comments to manuscript AMT-2017-120 - Anonymous referee 1**

The original referee's comments are written in black and the author's reply and changes to the manuscript are colored in blue/green respectively. References to page and line numbers as well as figures refer to the original manuscript, but references to sections refer to the corrected manuscript. In cases where we insert figures, tables and equations into this document, they are referenced with R1, R2, R3 …

The paper "A new instrument for stable isotope measurements of $^{13}$C and $^{18}$O in CO₂ - Instrument performance and ecological application of the Delta Ray IRIS analyzer", by Braden-Behrens et al, reports on a recent commercial instrument for measurements of stable CO₂ isotopes ($\delta^{13}$C -CO₂ and $\delta^{18}$O -CO₂) and its application in a field study. This work is relevant, however it is lacking many essential elements and it is not written carefully enough. The manuscript will thus need a major revision to be considered for publication in AMT.

This review focuses mainly on the performance evaluation of the instrument. It is important to also carefully review the hypothesis and the conclusions drawn from the field study. Most remarks and suggestions are added to the original manuscript. Some mayor aspects are discussed below.

**Author's response:** We thank the anonymous referee for the detailed feedback and suggestions to our manuscript, below we answer the referee's comments, starting with the major aspects mentioned in the review and followed by the additional comments in the supplement.

p3/22 The exact wavelength should be given together with a measured and fitted spectrum. The spectral range and the spectral resolution are important elements to judge the analytical performance, also in the context of gas matrix effects. One may assume that the frequencies are as in (Geldern 2014), but the latter does not show a measured spectrum.

**Author's response:** Thanks for pointing this out. We added the information about the spectral region and the used absorption lines to the manuscript. We also added more details about the drying of the air sample and the spectral fit to the description of the instrument in the introduction.
**Changes to the manuscript:** We added the following to the introduction.
"The instrument scans a spectral region from 4.3293 μm to 4.3275 μm (Geldern, 2014), containing four CO₂ absorption lines: at 4.3277 μm and 4.3280 μm (both for $^{16}$O$^{12}$C$^{16}$O), 4.3283 μm (for $^{16}$O$^{13}$C$^{16}$O), and 4.3286 μm (for $^{16}$O$^{12}$C$^{18}$O). A measured and a fitted spectrum is shown in Figure R1. The fitting procedure is based on a Voigt-Profile fit, that relates the isotopologue-specific absorption lines to their respective concentrations (information from the manufacturer, Thermo Fisher Scientific)"
We added a chapter about the spectrometer setup to the methods:
"Spectrometer setup
We set up the spectrometer to use the absorption lines at 4.3277 μm (for $^{16}$O$^{12}$C$^{16}$O), 4.3283 μm (for $^{16}$O$^{13}$C$^{16}$O), and 4.3286 μm (for $^{18}$O$^{12}$C$^{16}$O). Thus, only three of the four absorption lines in the instrument's measured spectra (Figure R1), were used for the spectral fit. In particular, for $^{16}$O$^{12}$C$^{16}$O, we did not use the strong absorption line at 4.3280 μm. The corresponding mode of operation is called "high concentration mode" in the instrument's operational software QTEGRA. Additionally, the sample was dried before it entered the measurement cell with the (instrument's internal) Nafion drier."

[Figure]

*Figure R1 Measured and fitted spectrum, as exported from the instrument's operational software QTEGRA.*

Water vapor may significantly impact the retrieved δ-values, either through spectral interference or through changes in absorption line characteristics (pressure broadening). If my understanding of the setup is correct, then humid samples were measured spectroscopically. Since this paper aims at validating a new spectrometer, it is vital to discuss and quantify the effect of changes in humidity.

**Author's response/ Changes to the manuscript:** The sample was dried with a Nafion drier before it was measured. This information was added to the manuscript, c.f. comment to p3/22.

(Along the same line) p3/23 describes the Delta Ray having "an internal calibration procedure that automatically includes two point calibrations for concentration c and both δ values as well as corrections for the concentration dependency of the measured d-values". This concept is interesting and a key feature of the Delta Ray. However, since this publication evaluates a commercial instrument, it should clearly describe the way concentration dependency is corrected (and how large it is) and to validate the procedure (accuracy, see above). This has not been achieved or is not presented.

**Author's response/changes to the manuscript:** We addressed this question by adding the chapter 'Evaluation of the calibration strategy', c.f. our comment to page p6/20, in especially Figures R2 and R3 below. Additionally, we changed the chapter about the calibration procedure to provide more detailed information. (We changed the order or your comments here, because we refer to this chapter later.)

2.6 Instrument internal calibration

The Delta Ray analyzer is equipped with three different internal calibration routines (Thermo Fisher Scientific, 2014). We performed these routines at the field site (in situ) each time the analyzer had to be restarted e.g. after power supply failures, instrument issues or when we manually turned off the analyzer for other reasons. All three instrument internal calibration procedures were usually done one day after restarting the analyzer, thus the instrument was in thermal equilibrium during calibration. The three different instrument internal calibration procedures are described below:

- **Correction of concentration dependency (called 'linearity calibration' in the instrument's documentation and operational software)**
  This calibration routine evaluates the concentration dependency of δ value measurements (Thermo Fisher Scientific, 2014). Mathematically, an experimentally derived correction factor $f_{correct}$ ($c_{raw}$) is multiplied with the raw isotopic ratio R (information from the manufacturer, Thermo Fisher Scientific)

$$R_{c-corrected} = f_{correct}(C_{raw}) \times R_{raw}$$

  (Equation R1)

  This factor as a function of concentration is determined via a natural spline fit of measurements of a gas tank with constant δ value at different concentrations (information from the manufacturer, Thermo Fisher Scientific). This is implemented by mixing pure $CO_2$ with $CO_2$.-free air, yielding concentrations between 200 to 3500 ppm. In our setup we used the pure $CO_2$ with near to ambient δ values (tank 'ambient $CO_2$ ', c.f. Table 3) and synthetic air for this calibration.

  […]

The instrument's internal calibration procedure is based on the measurement of these calibration curves after the instrument is started in combination with repeated measurements of a known gas, so called 'referencing' (see below). As the different calibrations are only performed once after the instrument is restarted, the accuracy and repeatability of measurements is further based on the assumption that, these relationships remain sufficiently constant, and temporal changes are corrected by 'referencing'.

- *Referencing*
  This procedure applies an offset correction of the calibrated δ values using a gas with known δ values that is measured at a freely selectable concentration in regular intervals (information from the manufacturer, Thermo Fisher Scientific). In our experimental setup, referencing is carried out every 30 minutes for 80 s after the tubes have been purged for 60 s using the pure $CO_2$ standard ('ambient $CO_2$', c.f. Table 3) diluted with synthetic air. We chose the reference concentration to be the same as in the highest inlet in the adjacent cycle, because most of the measurement inlets had concentrations close to those at the highest inlet and the temporal variability of the measured concentrations generally decreased with height. Thus, we performed the 'Referencing' as close as possible to as many height measurements as possible by these settings."

Thus, the calibration procedure for δ values can be expressed with the following formula with the correction factor $f_{correct}$ ($c_{raw}$) as determined from the concentration dependency correction, and the slope $m_{\delta scale}$ derived from the δ scale calibration (information from the manufacturer, Thermo Fisher Scientific).

$$\delta_{calibrated}(R_{raw}; C_{raw}; t) = m_{\delta scale} \times \underbrace{\left( \frac{f_{correct}(C_{raw})R_{raw}}{R_{std}} - 1 \right)}_{\delta_{c-corrected}} + \delta_{Offset}(t)$$

  (Equation R2)

p4/12 physically different samples: There is no indication that the instrument was used in a batch mode configuration. In continuous flow mode (as the text suggests), mixing in the cell (and to some

extend in the tubing) corresponds to a low-pass filter, which is fundamentally different to "physically different samples".

**Author's response:** We wanted to make sure, that we do not measure air samples that are majorly composed of the air masses in the previous measurement, thus we chose an averaging time that is larger than tau$_{5\%}$, yielding a situation in which less than 5% of the previous sample is mixed into the new sample (as $\tau_{5\%} = \tau_{10\%}$ ln(0.05)/ln(0.1) ≈14s, c.f. section 3.1.4). We agree that the formulation is misleading and changed the sentence to:
**Changes to the manuscript:** We changed this to "consisting of four measurements each averaged for 20 s - thus the averaging time is longer than the instrument internal cell response time $\tau_{10\%}$ c.f. section 3.1.4"

p4/1220 "temporal stability" is not standard terminology and only used once in this paper. I suggest using "repeatability", following the international vocabulary of metrology (VIM) throughout the text.

**Author's response/changes to the manuscript:** We changed this terminology and use 'repeatability' throughout the text.

p6/20 Accuracy was tested by comparing with one (1) gas tank which was measured using an Aerodyne spectrometer. This part of the study is a key element and completely insufficient. The main challenge in laser spectroscopy is currently not (any more) precision but rather accuracy. There is no reason why anyone should trust another spectrometer (here Aerodyne) without a very detailed description of how the latter achieves traceability. Furthermore, accuracy will depend on at least two calibration scales, i.e. δ values and concentration. Therefore, the evaluation must (!) include measurements of traceable (likely IRMS) gases at different δ values and concentrations; otherwise it is an insufficient and somewhat random exercise. If this is not possible, then an alternative may be to use traceable standards and (!) field samples that are quantified in a traceable way. This is easily possible for $\delta^{13}C$ -$CO_2$, but more difficult for $\delta^{18}O$ -$CO_2$ because of the limited stability of the samples (see e.g. Tuzson 2007, DOI: 10.1007/s00340-008-3085-4).

**Author's response:**
Concerning the general concerns about our accuracy measurement with N=1: We changed the chapter about accuracy in the manuscript and included measurements with gas tanks at different concentrations (N=4) and δ values (N=4 for $^{13}$C and N=5 for $^{18}$O), see description below. Here, we also evaluate 'potential accuracy' as defined by (Tuzson 2007, DOI: 10.1007/s00340-008-3085-4).
Concerning the comparison to the Aerodyne instrument: We agree that it is problematic to use another laser spectrometer for comparison here. We additionally analyzed this tank with a Picarro (for $CO_2$ concentration) and IRMS (for δ values) at Max Planck Institute for Biogeochemistry in Jena. In the revised manuscript, we use only gas tanks that were measured at MPI in Jena, both: for calibration as well as measuring potential accuracy (c.f. Table 3).
**Changes to the manuscript:** We rewrote chapter 2.5 to include additional measurements:
2.6 *Instrument characterization measurements*
We carried out additional measurement in the field and in the lab to quantify precision, evaluate the calibration strategy and quantify the instrument's response time and repeatability. These measurements involved changes in the analyzers plumbing. For all measurements that required connecting different gas tanks to the analyzer, they were either connected directly to the analyzer's internal ports ('CRef1' and 'CRef2') or the plumbing was equivalent to the plumbing of the target gas (Fig.1).

1) Lab measurements to quantify precision and evaluate the calibration strategy
   - We measured the Allan deviation by connecting pressurized air at atmospheric $\delta$ values to the analyzer and took measurements at the analyzer's maximum data acquisition rate of 1 Hz for two hours.
   - We diluted pure $CO_2$ with synthetic air over a $CO_2$ concentration range of 200 to 1500 ppm to measure the concentration dependency of the measured (raw) $\delta$ values. This dilution experiment was carried out for three different tanks with pure $CO_2$ at different $\delta$ values. Each gas tank was measured twice. (Used gas tanks: "ambient", "bio1" and "bio2", c.f. Table 3.)
   - We measured the concentration c and the isotopic compositions $\delta^{13}C$ and $\delta^{18}O$ of gases with concentrations ranging from (350 to 450 ppm) and isotopic compositions ranging from -37 to -9.7 ‰ for $\delta^{13}C$ and from -35 to -5 ‰ for $\delta^{18}O$. Each of these measurements was performed three times. (Used gas tanks: "ambient", "bio1","bio2", "PA-tank", $SACO_2$ -350, $SACO_2$ -450, $SACO_2$ -500, c.f. Table 3.)
   - We performed measurements of two pure $CO_2$ gas tanks at different $\delta$ values (diluted to different concentrations between 200 and 3000 ppm) as well as measurements of two gas tanks at different $CO_2$ concentrations (350 and 500 ppm). These measurements were repeated every six hours for a period of one week. (Used gas tanks: "ambient", "bio", ('SA-$CO_2$-350' and 'SA-$CO_2$-500', c.f. table 3.)"

2) Field measurements to quantify the setup's response time and repeatability
   - The response time of the tubing and the analyzer was measured by using the automatic switching unit (Figure 1) to switch from ambient air (height 1) to the target standard. We superimposed the measurements of four switching events to observe the adjacent turnover processes.
   - The analyzer's repeatability under field conditions was quantified by the half hourly target measurements described in Sect. 2.5.

We removed the chapter "Accuracy" and replaced it by the following:
3.1.2 Evaluation of the calibration strategy

The instrument's internal calibration strategy (described in section 2.7.1) is based on:

- A nonlinear relationship between raw $\delta$ values and concentrations (Figure R2).
- A linear relationship between calibrated $\delta$ value (measured with IRMS) and the concentration-corrected $\delta$ value - $\delta_{c-corrected}$ in Equation R1 (Fig. R3, left panel).
- A linear relationship between measured (raw) and real concentrations (Figure R3, middle and right panel).
- The repeatability of the calibration curves – for $\delta$ values modulo the Offset correction, that is applied by the instrument's internal 'Referencing' (Figure R4 and Table R1).

Raw $\delta$ values show a nonlinear dependency from raw concentrations (Fig. R2). This nonlinear relationship deviates from the concentration-dependency correction applied by the instrument ($\delta_{c-corrected}$ in equation R2). In Fig. R2, this function is shown for the used gas tank 'ambient' after an Offset correction at a concentration at 400 ppm, which is similar to the instrument's internal 'referencing'. Thus, the deviations of the measured $\delta$ values from the concentration-dependency correction (top panel of Fig. R2) give an estimate of the uncertainty of measurements that is related to the deviation from the reference concentration. For referencing at 400 ppm, these deviations were below 0.2 ‰ for $^{13}C$ and 0.4 ‰ for $^{18}O$.

[Figure]

*Figure R2 Box whiskers plots showing the nonlinear concentration-dependency of raw δ values for $^{13}$C and $^{18}$O respectively, here as an example for the CO₂ tank 'ambient'. This measured c-dependency is compared to the respective concentration-dependency correction (black line, with grey symbols marking the data points used during the respective calibration measurement). The c-dependency correction is Offset-corrected to match the raw δ values at 400ppm and the mean deviation from the fit is shown in the top panel for two measurements (different symbols) with three different gas tanks ('ambient' in blue, 'bio' in black and 'bio2' in red).*

The measured linear relationships for concentration and δ scale calibration (Fig. R3) have R^2 values of above 0.9999 for concentration, above 0.999 for δ $^{13}$C, and above 0.998 for δ $^{18}$O. The linearity and potential accuracy, as defined by (Tuzson et. al., 2008) can be quantified as the 1σ standard deviation from the linear fits. The so defined potential accuracy of the instrument internal calibration is 0.45 ppm for CO₂ concentration; 0.24 ‰ for δ$^{13}$C and 0.3 ‰ for δ$^{18}$O. For both δ values, this is comparable to the uncertainty related to the nonlinear concentration calibration that varies with δ and c as discussed above.

[Figure]

*Figure R3 Linear calibrations for concentration (left panel), $\delta^{13}$C (middle panel) and $\delta^{18}$O (right panel).*

The repeatability of the calibration curves is discussed here based on measurements of the nonlinear concentration dependency (Figure R2), and repeated measurements of gas tanks with two different c

and δ values to evaluate temporal changes in the respective linear relationships (Figure R3). These measurements were taken every six hours for a period of nine days. The standard deviation of the different measurement is below 0.2 ppm for concentrations and below 0.05 and 0.1 ‰ for [13]C and [18]O respectively. Thus the uncertainty related to the repeatability of the linear calibrations is smaller than the potential accuracy discussed above. For δ values, these values are comparable to the repeatability reported by several authors measured with other laser spectrometers (e.g. Sturm et al 2011; 2013; Vogel et al 2013). For concentrations on the other hand, Sturm et al 2013 reported a much smaller value of 0.03 ppm, based on more frequent calibration. In our setup, the concentration calibration is only performed once after the instrument is restarted, thus there might be a potential for better repeatability in concentration measurements by more frequent concentration calibration. For δ values, the repeatability that is related to deviations from the reference concentration depends on concentration (Table R1). Repeated measurements of these deviations have standard deviations of below 0.15 ‰ for concentrations between 200 and 1600 ppm.

[Figure]

*Figure R4 Box whiskers plots for the deviations of calibrated concentrations and δ values from laboratory measurements (at MPI in Jena) for repeated measurements of different calibration tanks (c.f. Table 3 for c and δ values of the gas tanks) over a period of 9 days (N=36). Delta values were measured at 400 ppm and 'referencing' was done app. Every 30 minutes at 380ppm to simulate conditions during a measurement campaign.*

| | tank ,ambient' | | tank ,bio' | |
| --- | --- | --- | --- | --- |
| Concentrations | $\sigma\,(\delta^{13}C - \delta^{13}C_{tank})$ | $\sigma\,(\delta^{18}O - \delta^{13}C_{tank})$ | $\sigma\,(\delta^{13}C - \delta^{13}C_{tank})$ | $\sigma\,(\delta^{18}O - \delta^{18}O_{tank})$ |
| 202 | 0,07 | 0,14 | 0,09 | 0,13 |
| 396 | 0,04 | 0,05 | 0,08 | 0,08 |
| 600 | 0,09 | 0,08 | 0,12 | 0,12 |
| 807 | 0,08 | 0,08 | 0,11 | 0,11 |
| 1018 | 0,10 | 0,08 | 0,13 | 0,11 |
| 1232 | 0,12 | 0,09 | 0,13 | 0,11 |
| 1450 | 0,14 | 0,11 | 0,15 | 0,12 |
| 1664 | 0,14 | 0,11 | 0,14 | 0,12 |
| 3145 | 0,17 | 0,15 | 0,17 | 0,15 |

*Table R1 Standard deviations σ of the differences between the calibrated δ values and the known values of used tanks 'ambient' and 'bio' over a large concentration range.*

p6/26 Measurement of the Allan plot was done in the lab because of limited gas supply in the field. This is not sufficient, because the goal of this study is characterization under field conditions. The argument of limited gas in the field is not convincing because at 80 ml/min, it would easily be possible to have many corresponding measurements of about 10 - 30 minutes, which, given an Allan Minimum at around 100 s, would be sufficient. A minimal approach would be to evaluate the 80 s target gas measurements. Alternatively, or in addition, one may use ambient conditions that are sufficiently stable (e.g. well mixed, afternoon, highest sampling port) to obtain at least a conservative estimate for the precision in the field. Finally, data from the PA tank measurement also give an indication of precision in the field.

**Author's response:** Thanks for these suggestions. We used the field measurements with the PA-tank to calculate Allan Deviations under field conditions at an averaging time of $\tau = 1$ s, yielding comparable values, c.f. table R2.

| $\sigma_A$ $^{13}$C [ppm] | | $\sigma_A$ $^{18}$O [ppm] | | $\sigma_A$ c [ppm] | |
|---------|---------|---------|---------|---------|---------|
| Lab | Field | Lab | Field | Lab | Field |
| 0.29 | 0.34 | 0.40 | 0.44 | 0.09 | 0.09 |

*Table R2: Comparison of Allan Deviations at 1 s averaging time based on field and lab measurements.*

However, based on your questions about the calibration strategy (see comment to p6/20 above) and your comment about instrument characteristics under field conditions (see your comment to p1/4 below), we decided to add more lab measurements to this manuscript (e.g. measurements to evaluate the calibration scheme). Thus in the revised manuscript, we generally focus more on lab measurements to characterize the instrument.
**Changes to the manuscript:** We removed "under field conditions" in the abstract, and rewrote chapter '2.5 Instrument characterization measurements' see our answer to your comment to p6/20.

p7/30 Referencing was done at the concentration of the highest sampling port. Discuss the uncertainty resulting from the fact that some height had other concentrations, taking into account the "linearity calibration" (which does not test linearity but dependence of the retrieved δ values on c; a terminology that should be improved).

**Author's response:**
Concerning the terminology: We agree that the term "linearity calibration" is not very clear. We used this term because this is the name of the corresponding calibration procedure that can be found in the Delta Ray's manual as well as in the operational software. Thus, we think we should keep this terminology, to be consistent with the manual. To avoid misunderstandings, we replaced "linearity calibration" by "Correction of c-dependency (called 'linearity calibration' in the instrument's documentation and operational software)" at first occurrence and by "Correction of c-dependency ('linearity calibration')" for the following occurrences.
Concerning the uncertainties related to the referencing: We addressed this question by adding a chapter 'Evaluation of the calibration strategy', c.f. our comment to page p6/20, in particular Table R1 and Figure R2.

p8/1 This whole chapter is badly written and should be revised with respect to language. In addition, the arguments are not convincing. The concentration range of HS and LS is not any larger than the standards used in the first calibration (300 and 430 ppm). Choosing two out of five standards, that were meant to evaluate accuracy for calibration, leads to only three remaining standards that are perfectly bracketed. The mean and uncertainty at N=3 becomes then statistically very weak. Furthermore, the results for c also illustrate why using just one tank to assess the accuracy of the δ values is not sufficient and somewhat arbitrary (see p6/20 above).

**Author's response:** We originally introduced this post calibration because we found a large jump in the concentration measured with the target standard. No such jump occurred in δ values. The jump in the target concentration could be removed replacing the instrument internal calibration with the applied post calibration. We agree with you, that it is not convincing that this is related to the concentration range of the instrument internal calibration. We think that during this period there was a problem with the instrument internal concentration calibration. The reason is not very clear to us; it might be that we have a problem with target gas flow during this particular concentration calibration. After replacing this particular concentration calibration by the linear post calibration, the corresponding jump in the target standard disappeared.

**Changes to the manuscript:** We rewrote the chapter about the post calibration and applied it only for a time period in which we observed a jump in the target concentration. "For the time period from the 15th of October to 15th of November, we replaced the instrument's internal concentration calibration by a manual linear calibration, based on manual measurements with five different gas tanks in the field. This was necessary, because measurements with five different gas tanks (including the target standard) showed a consistent linear relationship between raw and known concentrations, that deviated from the linear relationship that was used in the instrument's internal calibration. Thus, we conclude that during this period there was a problem with the instrument's internal concentration calibration which might be related to gas flow or a leak during this particular concentration calibration."

p9/15: The authors state that they chose an averaging time of 20 s as compromise between number of measurements and precision. This is misleading or not clear enough. If there are no measurements of standards between 20 s intervals, then the precision does not mean much because the next mean value for 20s may have an excellent precision (given as SD) but may have drifted significantly, thus the two values with good precision cannot be compared at the level of their individual precision (it then becomes an issue of repeatability or accuracy, depending on the context).

**Author's response:** We removed this misleading description.

p9/20 "the mean deviation of N=300 field measurements of a tank with pressurized air" is a suitable way to quantify repeatability and should be compared (or moved) to the results found in the corresponding chapter 3.1.3. Unfortunately, the values are only given graphically in Fig. 3. However, looking at the difference of one (!) sample, one cannot determine accuracy of the spectrometer. Especially not for an analytical technique which is known to be strongly dependent on concentration and gas matrix. The test is thus not suited for its aim. This chapter and the next can be combined to determine repeatability (preferred terminology), and which - at least in the title - may be called long-term stability. However, it is critical to find a way to reliably determine accuracy.

**Author's response/changes to the manuscript:** See comment to p6/20 about the accuracy measurement.

p9/25 "sum of uncertainties". What the authors likely mean is the combined uncertainty or an uncertainty budget. However, this is not achieved by simple addition of the uncertainties, as suggested in the text. It is necessary to know what the authors consider for the individual uncertainty contributions (and why), what distribution they assume and – if the contributions are independent – how they calculate the combined uncertainty, and at what level of confidence they then express this combined uncertainty.

**Author's response/changes to the manuscript:** This section was removed, instead we discuss 'potential uncertainty' as defined by Tuzson 2008**.**

P10/7 The standard deviations of repeated measurements (0.2 ppm for $CO_2$ concentration and below 0.3‰ δ values) should be compared to literature values. For example, (Sturm 2013, amt-6-1659-2013) found repeated measurements of the same gas tank with a standard deviation which is a factor 4-7 better than the results shown here.

**Author's response/changes to the manuscript:** We added more lab measurements and discuss the repeatability in chapter 'Evaluation of the calibration strategy', including the comparison to literature data, please see our comment to your question p6/20. Here we added the following to the discussion – please note that these values slightly changed, because we removed two periods with known instrument problems (c.f additional footnote and new figure 4) and recalculated the post-calibration only for the time period in which we observed a problem with our target measurements.

**"Repeatability during the field campaign**

For concentration, the measured repeatability of 0.3 ppm is slightly larger than the repeatability of the concentration calibration discussed above but still below the potential accuracy discussed in section 3.1.2. In the case of δ values, the obtained repeatability of app. 0.2‰ for $^{13}$C and 0.25 ‰ for $^{18}$O is larger than the repeatability of the linear calibration parameters obtained during lab measurements (0.05 ‰ for $^{13}$C and 0.1 ‰ for $^{18}$O). The measured repeatability during the field campaign also exceeds the repeatability of the measurements of the concentration dependency (below 0.15 ‰ for both δ values over a large concentration range) c.f. section 3.1.2. This could be related to the fact, that the δ values of our target standard were out of the calibration range, leading to an enhancement of fluctuations in the calibration parameters."

We added the following footnote*: "In the case of $^{13}$C, we excluded the target measurements between 23$^{rd}$ of September till 29$^{th}$ of September, because we obtained a problem with the $^{13}$C calibration that lead to a large jump in the delta $^{13}$C value of the (very depleted) target standard, but did not occur in the height measurements, probably because they were much closer to the reference delta value.

Fig 5 What is the slope of the linear decay, and what process does it represent?

**Author's response:** We derived this linear relationship from a first order approximation for the theoretical (and unrealistic) assumption that no mixing occurs in the measurement cell. We added the missing relevant information to the manuscript.
**Changes to the manuscript:** We added the following paragraph to the discussion.
"The linear fit shown here describes a first order approximation of the theoretical instrument response for the (unrealistic) assumption that there is no mixing in the measurement cell. From this assumption, it can be derived that the δ values would show a dominantly linear decay with the slope $m = (\delta_{new} - \delta_{old})/\tau_{theoretical}$ with the theoretical instrument cell response time $\tau_{theoretical} = p*V/\Phi$, with pressure p, Volume V and flow rate Φ. In our case $\delta^{13}C_{new} - \delta^{13}C_{old}$ = -29 ‰ , $\delta^{18}O_{new} - \delta^{18}O_{old}$ = -36.7 ‰ and $\tau_{theoretical}$ = 5.9 s."

**ADDITIONAL COMMENTS IN THE SUPPLEMENT OF RC1 (except typos and grammar mistakes, for which we directly include the referee's correction into the revised manuscript)**

P1/9 "field conditions" This is ok, but only if sufficiently exhaustive to replace lab characterization.
**Author's response/changes to the manuscript:** We changed this and included in general more lab measurements, c.f our comment to p6/20

P1/9 "accuracy of 0.1 ‰ for δ $^{13}$C" how can this be smaller than repeated measurements?
**Author's response/changes to the manuscript:** We changed the way how we quantified accuracy, c.f. our comment to p6/20. We changed the abstract to: "The potential accuracy (defined as the 1σ deviation from the respective linear regression that was used for calibration) was approximately 0.45ppm for c, 0.24‰ for $^{13}$C and 0.3‰ for $^{18}$O."

P1/14 "became insignificant" Explanation needed (or explicit statement that no explanation found).
**Author's response:** We are not sure which explanation is needed here a) the correlation itself or b) the change in the correlation coefficient from significant to insignificant? However, in the abstract, we just summarize the observed correlation and shifted the explanation into the discussion, because the discussion of both, a) and b) is a bit long.
**Changes to the manuscript:** "This correlation became insignificant (p>0.1) for the period after the first snow, indicating a decoupling of δ$^{13}$C of respiration from recent assimilates."

p2/25 and 25: "isotopologues" isotopocules, or isotopologues and isotomers, or remove bracket.
**Author's response:** We think it might be a bit confusing for the reader to use the term 'isotopocule' or add 'isotopomer' here, because the latter is irrelevant for CO$_2$- The Hitran database and many other authors use the the term isotopologue in this context (e.g. Kerstel and Giafriani 2008, Barbour 2011, Ellekoj 2013, Wehr2013 ,Oikawa 2017 , Mohn 2007 ,Affek and Yakir 2014, Vardag 2014). However, we added the information about isotopomers in the footnote and tried to give a clearer definition of isotopoloues.
**Changes to the manuscript:** "These rotational and vibrational transitions are characteristically different for isotopologues* (defined e.g. by Coplen 2007 as 'molecular species that differ only in isotopic composition'), see e.g. (Varadag 2014, Esler 2000, Kerstel and Giafrani 2008)."
We added this footnote*: "In general this is also true for isotopemers (defined by Coplen 2007 as 'Molecular species having the same number of each isotopic atom […] but differing in their positions.', (e.g. Mohn et al, 2008)".

P2/32 Since this is already cited, check the references and cite them directly.
**Author's response/changes to the manuscript:** We agree, but this part was removed anyway to shorten the introduction.

P3/9 The classification does not work for this instrument because it combines mid-IR with enhanced effective optical path length.
**Author's response:** Thanks for pointing that out. We added this information to the manuscript by adding a third category of instruments that combine the approaches of category a) and b)
**Changes to the manuscript:** We changed this classification in the introduction:
"A slightly modified categorization can be made that differs three classes of laser spectrometers (a) laser based direct absorption spectrometers in the mid infrared where strong absorption features are available (b) laser absorption spectrometers in the near infrared that compensate the weaker absorption in the near infrared by a strongly enhanced effective optical path length and (c) path length enhanced- absorption spectrometers in mid infrared. […] (Guillon et al… ). An example for an instrument of class category 2c) is the CCIA-48 (Los Gatos Research. Inc, *San Jose*, USA) that combines a mid-infrared quantum cascade laser with off-axis integrated cavity output spectroscopy (Oikawa et al., 2017)."

P3/20 "direct laser absorption spectrometer" direct absorption, not direct laser.
**Author's response/changes to the manuscript:** We changed this to "laser based direct absorption spectrometer" throughout the text to be clearer.

P5/33 "purging pump to avoid condensation in the tubes" purging is ok, but why should it avoid condensation (except because of pressure drop).
**Author's response/changes to the manuscript:** This was misleading, we changed this to: "We purged the main tube to reduce the time the air masses spend in the tubing. To avoid condensation, we heated the valve box (at which we expect a pressure drop) and the adjacent tubing."
P6/5 "the tubes with this small flow rate and" Please explain why condensation is linked to flow rate.
**Author's response/changes to the manuscript:** We removed this sentence.

P7/13 „linearity calibration" It's not really about linearity but about concentration dependence of the retrieved d values.
**Author's response/changes to the manuscript:** We changed the terminology, see our answer to your comment to p7/30.

P9/14 Put in relation. Is this better/worse/comparable?
**Author's response/changes to the manuscript:** We changed this to: "we measured a comparable (slightly better) Allan Deviation below 0.03‰ (c.f. Table 5)."

Multiple comments to chapter 3.1.2
- there is no such thing as "expected uncertainty".
- "measured uncertainty" unsuited terminology
- in the context of the evaluation of a new analyzer you have to make sure that this is not a coincidence. (N=1).
**Author's response/changes to the manuscript:** We replaced this chapter by a chapter about the evaluation of the calibration strategy. Please see our answer to your comment about p6/20

P10/10 "instrument drift" It would be very interesting to know what the instrument drift is. However, the data shown here is the drift of the retrieved data after all (drift) corrections.
**Author's response/changes to the manuscript:** Thanks for pointing this out. As the remaining drift after all drift corrections does not seem a meaningful quantity, we removed this part of the data evaluation.

P10/13 "Turnover time" This implies that it is the "turnover" of a perfectly mixed reactor (cell). However, what you then determine are several elements; I suggest calling this "response time".
**Author's response/changes to the manuscript:** We changed "turnover time" to "response time".

P10/20 „to mixing of gas" please state whether the gas flow is turbulent under the given conditions.
**Author's response/changes to the manuscript:** The gas flow in all tubes is laminar with Reynolds numbers below 100 for all tubing (6mm and 1/16'). We added this information to chapter 2.5.

P13/3 "As soil respiration has been measured to account for around 80% of total respiration in an old beech forest in below 30 km distance to our field site (Knohl et al., 2008), we further focus on soil respiration and discuss the following hypothesis:" Please check if this is really the line of thought that you want to communicate.
**Author's response:** We changed this to "Because soil respiration has been measured to account for around 80% of ecosystem respiration in an old beech forest in below 30 km distance to our field site (Knohl et al., 2008), we assume that soil respiration dominates ecosystem respiration and thus we further focus on soil respiration and discuss the following hypothesis:"

P30/table 5 "not necessary; Figure with 1 s and minimum values is sufficient."
**Author's response/changes to the manuscript:** We would like to keep this table for the readers convenience.

P30/table 6 „not necessary; can be described in one sentence."
**Author's response/changes to the manuscript:** We removed this table and added the numbers directly into the text: "The analyzer's power consumption of approximately 220W was slightly smaller than the power consumption of the basic infrastructure of the setup that included the pump to purge the 9 inlet tubes and the heated valve box (330W)."

P30/table 7 "not necessary; text and Fig. 4 are sufficient."
**Author's response/changes to the manuscript:** We would like to keep this table for the readers convenience.

P33/21 "review language of this paragraph"
**Author's response/changes to the manuscript:** We rewrote this paragraph: "Additionally, the calculation of Keeling-Plot intercepts based on longer timescales increased the number of Keeling-Plot intercepts within reasonable ranges. For Keeling-Plots that were averaged over 2h (5h), a fraction of 97% (99%) of the Keeling Plot intercepts were between -33 and -25‰. Because the range of the Keeling-Plot intercepts should not depend on the chosen timescale, we considered the Keeling-Plot intercepts that were outside of this range as outliers and removed them (also for Keeling-Plot Intercepts that were based on shorter timescales)."

---

## Author Comment (AC2) · 17 Aug 2017

**Author's reply to the referees comments to manuscript AMT-2017-120 - Anonymous referee 2**

The original referee's comments are written in black and the author's reply and changes to the manuscript are colored in blue/green respectively. References to page and line numbers as well as figures refer to the original manuscript, but references to sections refer to the corrected manuscript. In cases where we insert figures, tables and equations into this document, they are referenced with R1, R2, R3 …

This paper describes the testing and use of the new Delta Ray IRIS $CO_2$ isotope spectrometer during a three-month field campaign. I have serious concerns about the analytical details, as well as the conclusions regarding interpretations of the field measurements. Overall, I am not convinced that this instrument has been put through the necessary rigorous tests.

**Author's response:** We thank the anonymous referee for his comments and suggestions. In this response we show additional data of instrument test and more detailed data of the field measurements to answer the referee's questions.

1) The authors conclude in the abstract that "1) the new Delta Ray IRIS with its internal calibration procedure provides an opportunity to precisely and accurately measure c, $\delta^{13}C$ and $\delta^{18}O$ at field sites" I am concerned with this statement, because the internal calibration procedure in the IRIS is never actually described. How are the absorption spectra used to calculate isotope ratios, and how are these modified based on the calibration? This point appears critical for understanding whether the internal procedure is adequate and/or necessary, or for understanding what other post-hoc calibrations may be needed. This is a critical gap in the paper. Once cannot simply assume that the manufacturers of the instrument have worked out the details here. There are instruments that are sold that do not necessarily function as advertised, thus it is necessary to validate every step of the way. I would like to see plots and regressions of raw vs. known values for both $\delta^{13}C$ and $[CO_2]$ for a number of different standards spanning a broad range of δ values and mole fractions of $CO_2$ .

**Author's response:** We added this missing information about the spectral fit, the calibration procedure and about the evaluation of the calibration procedure (including the suggested plots) to the manuscript.

**Changes to the manuscript:**
We added the following to the introduction.
"The instrument scans a spectral region from 4.3293 μm to 4.3275 μm (Geldern, 2014), containing four $CO_2$ absorption lines: at 4.3277 μm and 4.3280 μm (both for $^{16}O^{12}C^{16}O$), 4.3283 μm (for $^{16}O^{13}C^{16}O$), and 4.3286 μm (for $^{16}O^{12}C^{18}O$). 
[revised manuscript text omitted]

The measured linear relationships for concentration and δ scale calibration (Fig. R3) have R^2 values of above 0.9999 for concentration, above 0.999 for δ $^{13}$C, and above 0.998 for δ $^{18}$O. The linearity and potential accuracy, as defined by (Tuzson et. al., 2008) can be quantified as the 1σ standard deviation from the linear fits. The so defined potential accuracy of the instrument internal calibration is 0.45 ppm for CO₂ concentration; 0.24 ‰ for δ$^{13}$C and 0.3 ‰ for δ$^{18}$O. For both δ values, this is comparable to the uncertainty related to the nonlinear concentration calibration that varies with δ and c as discussed above.

[Figure]

*Figure R3 Linear calibrations for concentration (left panel), δ$^{13}$C (middle panel) and δ$^{18}$O (right panel).*

The repeatability of the calibration curves is discussed here based on measurements of the nonlinear concentration dependency (Figure R2), and repeated measurements of gas tanks with two different c

and δ values to evaluate temporal changes in the respective linear relationships (Figure R3). These measurements were taken every six hours for a period of nine days. The standard deviation of the different measurement is below 0.2 ppm for concentrations and below 0.05 and 0.1 ‰ for [13]C and [18]O respectively. Thus the uncertainty related to the repeatability of the linear calibrations is smaller than the potential accuracy discussed above.  For δ values, these values are comparable to the repeatability reported by several authors measured with other laser spectrometers (e.g. Sturm et al 2011; 2013; Vogel et al 2013). For concentrations on the other hand, Sturm et al 2013 reported a much smaller value of 0.03 ppm, based on more frequent calibration. In our setup, the concentration calibration is only performed once after the instrument is restarted, thus there might be a potential for better repeatability in concentration measurements by more frequent concentration calibration. For δ values, the repeatability that is related to deviations from the reference concentration depends on concentration (Table R1). Repeated measurements of these deviations have standard deviations of below 0.15 ‰ for concentrations between 200 and 1600 ppm.

[Figure]

*Figure R4 Box whiskers plots for the deviations of calibrated concentrations and δ values from laboratory measurements (at MPI in Jena) for repeated measurements of different calibration tanks (c.f. Table 3 for c and δ values of the gas tanks) over a period of 9 days (N=36). Delta values were measured at 400 ppm and 'referencing' was done app. Every 30 minutes at 380ppm to simulate conditions during a measurement campaign.*

| | tank 'ambient' | | tank 'bio' | |
|---|---|---|---|---|
| Concentrations | $\sigma\ (\delta^{13}C - \delta^{13}C_{tank})$ | $\sigma\ (\delta^{18}O - \delta^{13}C_{tank})$ | $\sigma\ (\delta^{13}C - \delta^{13}C_{tank})$ | $\sigma\ (\delta^{18}O - \delta^{18}O_{tank})$ |
| 202 | 0,07 | 0,14 | 0,09 | 0,13 |
| 396 | 0,04 | 0,05 | 0,08 | 0,08 |
| 600 | 0,09 | 0,08 | 0,12 | 0,12 |
| 807 | 0,08 | 0,08 | 0,11 | 0,11 |
| 1018 | 0,10 | 0,08 | 0,13 | 0,11 |
| 1232 | 0,12 | 0,09 | 0,13 | 0,11 |
| 1450 | 0,14 | 0,11 | 0,15 | 0,12 |
| 1664 | 0,14 | 0,11 | 0,14 | 0,12 |
| 3145 | 0,17 | 0,15 | 0,17 | 0,15 |

*Table R1 Standard deviations σ of the differences between the calibrated δ values and the known values of used tanks 'ambient' and 'bio' over a large concentration range.*

2) The authors mention that they used a post-hoc $CO_2$ concentration calibration, but it is unclear how often the additional standards used for this were measured (once? Halfhourly?) in relation to their check standard. Note that quadratic relationships may give a better fit as employed elsewhere for other absorption-based $CO_2$ instruments.

**Author's response:** We originally introduced this post calibration because we found a large jump in the concentration measured with the target standard. No such jump occurred in δ values. The jump in the target concentration could be removed replacing the instrument internal calibration with the applied post calibration. We agree with you, that it is not convincing that this is related to the concentration range of the instrument internal calibration. We think that during this period there was a problem with the instrument internal concentration calibration. The reason is not very clear to us; it might be that we have a problem with target gas flow during this particular concentration calibration. After replacing this particular concentration calibration by the linear post calibration, the corresponding jump in the target standard disappeared.

**Changes to the manuscript:** Concerning the potential nonlinearity, Figure RC2 above evaluates the instrument's linearity and quantifies the deviations from the linear regression, please see also the chapter "Evaluating the calibration strategy" (above). We rewrote the chapter about the post calibration and applied it only for a time period in which we observed a jump in the target concentration:

"For the time period from the 15$^{th}$ of October to 15$^{th}$ of November, we replaced the instrument's internal concentration calibration by a manual linear calibration, based on manual measurements with five different gas tanks in the field. This was necessary, because measurements with five different gas tanks (including the target standard) showed a consistent linear relationship between raw and known concentrations, that deviated from the linear relationship that was used in the instrument's internal calibration. Thus, we conclude that during this period there was a problem with the instrument's internal concentration calibration which might be related to gas flow or a leak during this particular concentration calibration."

3) Given that this is a methods paper, it would have been very useful to see tests using a broader range of $CO_2$ mole fraction and isotope compositions in the range of standards, and to see more standards tested. Without this, we cannot validate the linearity of the instrument both in concentration and isotope space. This is a critical deficit of the paper. Why was the need for a post-hoc $\delta^{13}C$ and $\delta^{18}O$ calibration not tested or described?

**Author's response/changes to the manuscript:** We addressed this questions by adding an additional chapter about test measurements to evaluate the calibration strategy. This chapter evaluates the concentration dependency of the δ values over a range of 200 to 1500 ppm and includes measurements of different gas tanks with concentrations ranging from (350 to 450 ppm) and isotopic compositions ranging from -37 to -9.7 ‰ for $\delta^{13}C$ and from -35 to -5 ‰ for $\delta^{18}O$. Please see our response to question 1) above.

4) Note that many of the other laser-based isotope instruments achieve much higher precision with frequent (e.g. 20 minute) isotope calibrations in the field. This need appears especially critical here given the large (1 per mil) jumps in δ [13]C values observed in the check standards shown in Figure 4. This suggests that there are some serious stability problems that need to be addressed with more frequent isotope calibration.

**Author's response:**

Concerning the large jumps in observed δ values: Thanks for pointing this out. We showed Figure 4 mainly to show the repeatability of the instrument, but we agree with you, that the large (1 ‰) jumps in δ values need more discussion. For these two large jumps, we found explanations: The first of these large jumps appeared in [13]C after calibration on 23[th] of September and disappeared after calibration on 29[th] of September. This jump only occurs in δ [13]C of the target measurement. In particular, this jump was not visible in the δ [13]C in the measurements of the different heights (see figure R5 for the highest inlet as an example). Thus, we conclude that there was a problem with [13]C calibration. This problem might be enhanced for δ values that deviate from the 'reference' δ value, in particular for the very depleted target measurement, that was even out of the calibration range.

[Figure]

*Figure R5 Time series of $\delta^{13}C$ values for the time period that shows a large jump in δ [13]C for target measurements, but not for the height inlets, shown here as an example for the highest inlet.*

The second large jump in the time series of the isotopic composition of the target gas from the 9[th] until the 16[th] of October includes the period during which we had a laser alignment problem and the laser needed to be readjusted. After calibration on 16[th] of October, the measured target gas value jumped back to its value before the 9[th] of October. We originally wanted to show all data points for completeness, but as we can relate them to a) a problem with one specific δ [13]C calibration that occurs particularly for the very depleted target gas and b) a general laser alignment problem, we think it is more appropriate to remove the corresponding data points from further analysis. In case of the laser alignment problem we also removed the corresponding time series of the height measurements.

Concerning precision, here we quantify precision by measuring the Allan deviation of the uncalibrated δ values, like many other authors (e.g. Sturm et al 2013, van Geldern et al 2014). For this comparison, Table 2 gives an overview about the precision of the δ value measurements of different laser-based and broadband light source-based instruments. In case you refer to what we called 'long-term-stability' in the original manuscript, but call 'repeatability' in the revised manuscript, this is discussed in the chapter 'Evaluation of the calibration strategy' (See our comment

above).

**Changes to the manuscript:** We changed Figure 4 and the respective description to "Figure 4 Time series of target gas measurements excluding periods that show problems with target gas flow, calibration and a laser alignment problem."
We added the following chapter: "Repeatability during the field campaign
For concentration, the measured repeatability of 0.3 ppm is slightly larger than the repeatability of the concentration calibration discussed above but still below the potential accuracy discussed in section 3.1.2. In the case of δ values, the obtained repeatability of app. 0.2‰ for $^{13}$C and 0.25 ‰ for $^{18}$O is larger than the repeatability of the linear calibration parameters obtained during lab measurements (0.05 ‰ for $^{13}$C and 0.1 ‰ for $^{18}$O). The measured repeatability during the field campaign also exceeds the repeatability of the measurements of the concentration dependency (below 0.15 ‰ for both δ values over a large concentration range) c.f. section 3.1.2. This could be related to the fact, that the δ values of our target standard were out of the calibration range, leading to an enhancement of fluctuations in the calibration parameters."

We added the following footnote*: "In the case of $^{13}$C, we excluded the target measurements between 23$^{rd}$ of September till 29$^{th}$ of September, because we obtained a problem with the $^{13}$C calibration that lead to a large jump in the delta $^{13}$C value of the (very depleted) target standard, but did not occur in the height measurements, probably because they were much closer to the reference delta value.

5) With respect to the second major conclusion of the abstract, "2) even short snow or frost events could have strong effects on the isotopic composition of $CO_2$ exchange at ecosystem scale" this finding is not new, but also not very well supported by the data (e.g. Figs 7 and 8. There are now several multi-year datasets of canopy $CO_2$ and δ $^{13}$C profiles in temperate ecosystems that have shown similar patterns.

**Author's response:**
Concerning conclusion 2) Here we summarize the results concerning $^{13}$C as well as $^{18}$O that are discussed in detail in the results section. This statement does not only refer to Figures 7 and 8. The parts of the manuscript that support this conclusion are in particular figure 9 (top panel) for $^{13}$C and Table 8 for $^{18}$O (in addition to figure 7). As we discuss in section 3.2.2, for $^{13}$C we do not observe a change in the δ $^{13}$C values, but we find indications, that the processes controlling the $^{13}$C of $CO_2$ exchange shifted. For brevity in the abstract, we tried to stay general, but specified this in the revised manuscript.
Concerning the mentioned multi-year records: We are well aware that there are now several multi-year records of $^{13}$C and $^{18}$O in $CO_2$ profiles (e.g. Bowling et al 2002b, Wehr et al 2016, Bowling et al 2003, Shim et al 2013). However, we are not so sure if the 'similar' pattern that you talk about show the same change in the time lagged (and negative) correlation between Reco$^{13}$C and 2-d averaged radiation (not VPD), particularly in the combination with frost events. It would be very interesting for us to see which species ($^{13}$C or $^{18}$O) and which datasets you are referring to in particular and we are happy to include the respective citation
**Changes to the manuscript:** We specified the abstract "2) even short snow or frost events could have strong effects on the isotopic composition (in particular $^{18}$O) of $CO_2$ exchange at ecosystem scale."
We added a more comprehensive list of citations, focussing on multi-year record to the introduction (page to page 2 line 12) "The temporal variability of the isotopic composition of respiration for example has been studied on timescales ranging from sub-diurnal (Barbour et al., 2011) to seasonal (Ekblad and Högberg, 2001; Bowling et al., 2002; Knohl et al., 2005). Further, the isotopic

composition in $CO_2$ profiles has been studied on several sites over multiple years for $^{13}C$ (e.g. Bowling et al 2002b, Wehr 2016 ) as well as for $^{18}O$ (e.g. Bowling 2003, Shim et al 2013)."
We added a sentence referring to the observed peaks $R_{eco}^{18}O$ in the monsoon-dominated woodland observed by Shim et al to the discussion: "Similarly strong peaks in $R_{eco}^{18}O$ have been observed in a semi-arid woodland after precipitation in New Mexico (Shim et al 2013), but this study refers to a monsoon dominated ecosystem with comparably large variability in the $^{18}O$ and does not focus on the difference of these pulses of snow and rain events."

6) With respect to the Keeling plot intercepts, no data is shown to actually validate the approach (e.g. plots of $\delta^{13}C$ and $1/CO_2$ space), nor summary statistics presented for these regressions. This is another serious deficit given the key methodological issues the authors point out in the Appendix, but do not quantify in the text. I don't think the authors present enough information here to rigorously test the hypotheses proposed in the Results/Discussion section.

**Author's response:** Below we show the histogram of $R^2$.

[Figure]

*Figure R6 Histograms showing the $R^2$ values of accepted Keeling Plots based on data that was measured within 90 minutes during nighttime (between 20pm and 4 am).*

**Changes to the manuscript:** We provide summary statistics about the regressions to the text:
"The filtered nighttime Keeling-Plot intercepts based on 30 minutes of data acquisition had $R^2$ values with a median of 0.87 and 0.81 for $^{13}C$ and $^{18}O$ with mean values of 0.85 and 0.77 and standard deviations of 0.1 and 0.16 respectively."
We added the following Example Keeling-Plots to the supplementary material. We chose Keeling-Plots with $R^2$ values spanning the range of the respective mean+- 1 standard deviation.

[Figure]

*Figure 7 Example nighttime Keeling-Plots with typical R^2-values (spanning the range of the mean +-1σ. Each Keeling-Plot is based on 90 min input data. Different colors represent different inlet heights.*

The value of the CANVEG modeling exercise for the overall study was not terribly apparent to me, nor were the questions that it sought to address.

**Author's response:** We included the modelling to test the Hypothesis (a) (page 12 lines 1-14) as discussed in particular in lines 22ff. We modified this explanation to make the reason for the inclusion of the model clearer.

**Changes to the manuscript:** We added an additional sentence to the explanation in section 3.2.2 "Hypothesis (a):The variability of $R^{13}C$ eco can be partly explained by the isotopic composition of recent assimilates $^{13}C$ Ass, which is controlled by meteorological drivers during photosynthesis according to the Farquhar model. Thus, the variability of $R^{13}C_{eco}$ is linked to the variability of meteorological drivers of photosynthesis and photosynthetic discrimination with a time lag that is consistent with the time lag between respiration and assimilation. […We observed a correlation between radiation $R_n$ and $R^{13}C_{eco}$,…] But the correlation itself cannot be directly explained by the Farquhar model of discrimination as radiation influences both, the $CO_2$ supply (by influencing stomatal conductance) and the $CO_2$ demand (by influencing assimilation) in the leaf (Farquhar and Sharkey, 1982). In particular, we did not find a significant time lagged positive correlation between $R^{13}C$ eco and VPD, RH or the ratio VPD/PAR (Fig. 8), which could be directly associated with the Farquhar Model and has been found by the above mentioned studies. [this refers to (Ekblad and Högberg, 2001; Bowling et al., 2002; Knohl et al., 2005)] To test if it might be still reasonable to interpret the observed negative correlation of $R^{13}C$ eco with Rn as a time lagged link between $R^{13}C$ eco and isotopic composition of recently assimilated material $^{13}C$ Ass on ecosystem scale, we performed a more complex calculation of $^{13}C$ Ass by using the multilayer model CANVEG. The advantage of CANVEG is that it accounts for the non-linear interactions between air temperature, air humidity, radiation, stomatal conductance and photosynthesis."

To explain this thought earlier, we added/moved the following to the beginning of chapter 2.8:
"To test if the measured variability of the isotopic composition of respiration can be partly explained by the variability of the isotopic composition of recent assimilates, we used the Multi-layer model CANVEG to simulate the isotopic composition of assimilated material during our measurement campaign. In particular, we analyzed the correlation of modeled $^{13}C$ Ass to net radiation $R_n$, a driver of photosynthesis and photosynthetic discrimination, during our measurement period in autumn 2015. We further compared the resulting relationship between Rn and $^{13}C$ Ass to the observed (time lagged) relationship between Rn and the $^{13}C$ composition of respiration $R_{eco}^{13}C$, derived from the measured

Keeling-Plots, c.f. section 3.2.2. This analysis was performed to test the hypotheses of a link between δ values in assimilated material and respiration."

7) More specific comments:

Introduction: there is much excessive detail here that repeats recent reviews, such as the Griffis 2013 paper. Please condense.

**Author's response/changes to the manuscript:** We shortened the introduction, in particular p2 line 22 ff.

P1 18: the main constraint is low temporal resolution

**Author's response/changes to the manuscript:** Thanks for pointing this out. We added this information to the manuscript.

P4 13: how are these "physically different" air samples if the pump is flowing continuously?

**Author's response/changes to the manuscript:** We removed this misleading description.

P8 5: "A possible reason for this resulting deviation is the range of the gas tanks we used for the instrument-internal concentration calibration, that was approximately 300 to 430 ppm" this logic doesn't make sense to me this is similar to your other standards
**Author's response:** We agree that this might not be the reason for the observed problems with concentration calibration, please see our answer to your comment 2).

P8 6: I am having trouble understanding how your "target standard" could be stable without posthoc calibration yet your five other standards were so variable.

**Author's response:** This was not the case. We found a need for post concentration calibration because the 'target' standard was not stable, please see our answer to your comment 2).

P8 9: "Secondly we set the IRIS analyzer's internal referencing procedure (described in Sect. 2.7) to 1800 s which corresponds to an Allan variance of 0.03 ‰ for both δ values and 0.01 ppm for $CO_2$ concentration." This is unclear to me are you measuring the standards every 1800 s? For how long?
**Author's response:** Yes, we measured the reference standard every 30 minutes. We measured it for 80s after the tubes were purged for 60s. We rewrote the chapter about the calibration procedure to be clearer (see our comment above).

Where are these new Allan variance values coming from?
**Author's response:** That was a typo. Thanks for finding it!

Figure 4: There are apparently large (1 per mil) jumps in measured "target gas" isotope values at several points. These are disconcerting. Are the data shown in this figure the raw values or the calibrated values? If they are the calibrated values, this suggests that the two-point calibration employed here is inadequate.

**Author's response:** The figure you are referring to (figure 4 in the original manuscript) shows calibrated values. Please see our answer to your comment number 4) above and the new chapter about the evaluation of the calibration strategy.

---

## Author Comment (AC3) · 17 Aug 2017

We thank the anonymous referee for the feedback and the suggestions to our manuscript, in the supplementary pdf we answer the referee's comments in detail.

Please also note the supplement to this comment:
https://www.atmos-meas-tech-discuss.net/amt-2017-120/amt-2017-120-AC3-supplement.pdf

---

## Author Response (AR1)

**Point-by-point reply to all referee comments to manuscript AMT-2017-120**

We thank all referees for their feedback and suggestions. Below we answer the reviews point-by-point followed by a marked up pdf that tracks changes in the manuscript (produced with latexdiff). The original referee's comments are written in black and the author's reply and changes to the manuscript are colored in blue/green respectively. References in green refer to the corrected manuscript and references in black (e.g. pagenumbers) refer to the original manuscript.

**Author's reply to anonymous referee 1**

p3/22 The exact wavelength should be given together with a measured and fitted spectrum. The spectral range and the spectral resolution are important elements to judge the analytical performance, also in the context of gas matrix effects. One may assume that the frequencies are as in (Geldern 2014), but the latter does not show a measured spectrum.

**Author's response:** Thanks for pointing this out. We added the information about the spectral region and the used absorption lines to the manuscript. We also added more details about the drying of the air sample and the spectral fit to the description of the instrument in the introduction.

**Changes to the manuscript:** We added this information to the introduction (page 3 line 12). We added chapter 2.3 about the spectrometer setup to the methods.

Water vapor may significantly impact the retrieved δ-values, either through spectral interference or through changes in absorption line characteristics (pressure broadening). If my understanding of the setup is correct, then humid samples were measured spectroscopically. Since this paper aims at validating a new spectrometer, it is vital to discuss and quantify the effect of changes in humidity.

**Author's response/ Changes to the manuscript:** The sample was dried with a Nafion drier before it was measured. This information was added to the manuscript, c.f. comment to p3/22.

(Along the same line) p3/23 describes the Delta Ray having "an internal calibration procedure that automatically includes two point calibrations for concentration c and both δ values as well as corrections for the concentration dependency of the measured d-values". This concept is interesting and a key feature of the Delta Ray. However, since this publication evaluates a commercial instrument, it should clearly describe the way concentration dependency is corrected (and how large it is) and to validate the procedure (accuracy, see above). This has not been achieved or is not presented.

**Author's response/changes to the manuscript:** We addressed this question by adding the chapter 'Evaluation of the calibration strategy', c.f. our comment to page p6/20, in especially Figures 4 and 5 in the revised manuscript. Additionally, we changed chapter 2.8.1 about the calibration procedure to provide more detailed information. (We changed the order or your comments here, because we refer to this chapter later.)

p4/12 physically different samples: There is no indication that the instrument was used in a batch mode configuration. In continuous flow mode (as the text suggests), mixing in the cell (and to some extend in the tubing) corresponds to a low-pass filter, which is fundamentally different to "physically different samples".

**Author's response:** We wanted to make sure, that we do not measure air samples that are majorly composed of the air masses in the previous measurement, thus we chose an averaging time that is larger than tau$_{5\%}$, yielding a situation in which less than 5% of the previous sample is mixed into the new sample (as $\tau_{5\%} = \tau_{10\%} \ln(0.05)/\ln(0.1) \approx 14s$, c.f. section 3.1.4). We agree that the formulation is

misleading and changed the sentence to:

**Changes to the manuscript:** We changed this to "consisting of four measurements each averaged for 20 s - thus the averaging time is longer than the instrument internal cell response time $\tau_{10\%}$ c.f. section 3.1.4"

p4/20 "temporal stability" is not standard terminology and only used once in this paper. I suggest using "repeatability", following the international vocabulary of metrology (VIM) throughout the text.

**Author's response/changes to the manuscript:** We changed this terminology and use 'repeatability' throughout the text.

p6/20 Accuracy was tested by comparing with one (1) gas tank which was measured using an Aerodyne spectrometer. This part of the study is a key element and completely insufficient. The main challenge in laser spectroscopy is currently not (any more) precision but rather accuracy. There is no reason why anyone should trust another spectrometer (here Aerodyne) without a very detailed description of how the latter achieves traceability. Furthermore, accuracy will depend on at least two calibration scales, i.e. $\delta$ values and concentration. Therefore, the evaluation must (!) include measurements of traceable (likely IRMS) gases at different $\delta$ values and concentrations; otherwise it is an insufficient and somewhat random exercise. If this is not possible, then an alternative may be to use traceable standards and (!) field samples that are quantified in a traceable way. This is easily possible for $\delta^{13}C$ -$CO_2$, but more difficult for $\delta^{18}O$ -$CO_2$ because of the limited stability of the samples (see e.g. Tuzson 2007, DOI: 10.1007/s00340-008-3085-4).

**Author's response:**
Concerning the general concerns about our accuracy measurement with N=1: We changed the chapter about accuracy in the manuscript and included measurements with gas tanks at different concentrations (N=4) and $\delta$ values (N=4 for $^{13}C$ and N=5 for $^{18}O$), see description below. Here, we also evaluate 'potential accuracy' as defined by (Tuzson 2007, DOI: 10.1007/s00340-008-3085-4).
Concerning the comparison to the Aerodyne instrument: We agree that it is problematic to use another laser spectrometer for comparison here. We additionally analyzed this tank with a Picarro (for $CO_2$ concentration) and IRMS (for $\delta$ values) at Max Planck Institute for Biogeochemistry in Jena. In the revised manuscript, we use only gas tanks that were measured at MPI in Jena, both: for calibration as well as measuring potential accuracy (c.f. Table 3).
**Changes to the manuscript:** We rewrote chapter 2.6 about Instrument characterization measurements to include additional measurements.
We removed the chapter "Accuracy" and replaced it by chapter 3.1.2 Evaluation of the calibration strategy.

p6/26 Measurement of the Allan plot was done in the lab because of limited gas supply in the field. This is not sufficient, because the goal of this study is characterization under field conditions. The argument of limited gas in the field is not convincing because at 80 ml/min, it would easily be possible to have many corresponding measurements of about 10 - 30 minutes, which, given an Allan Minimum at around 100 s, would be sufficient. A minimal approach would be to evaluate the 80 s target gas measurements. Alternatively, or in addition, one may use ambient conditions that are sufficiently stable (e.g. well mixed, afternoon, highest sampling port) to obtain at least a conservative estimate for the precision in the field. Finally, data from the PA tank measurement also give an indication of precision in the field.

**Author's response:** Thanks for these suggestions. We used the field measurements with the PA-tank to calculate Allan Deviations under field conditions at an averaging time of $\tau$ =1 s, yielding comparable values, c.f. table 4.

| σ$_A$ $^{13}$C [ppm] | | σ$_A$ $^{18}$O [ppm] | | σ$_A$ c [ppm] | |
|---|---|---|---|---|---|
| Lab | Field | Lab | Field | Lab | Field |
| 0.29 | 0.34 | 0.40 | 0.44 | 0.09 | 0.09 |

*Table 4: Comparison of Allan Deviations at 1 s averaging time based on field and lab measurements.*

However, based on your questions about the calibration strategy (see comment to p6/20 above) and your comment about instrument characteristics under field conditions (see your comment to p1/4 below), we decided to add more lab measurements to this manuscript (e.g. measurements to evaluate the calibration scheme). Thus in the revised manuscript, we generally focus more on lab measurements to characterize the instrument.

**Changes to the manuscript:** We removed "under field conditions" in the abstract, and rewrote chapter '2.5 Instrument characterization measurements' see our answer to your comment to p6/20.

p7/30 Referencing was done at the concentration of the highest sampling port. Discuss the uncertainty resulting from the fact that some height had other concentrations, taking into account the "linearity calibration" (which does not test linearity but dependence of the retrieved δ values on c; a terminology that should be improved).

**Author's response:**
Concerning the terminology: We agree that the term "linearity calibration" is not very clear. We used this term because this is the name of the corresponding calibration procedure that can be found in the Delta Ray's manual as well as in the operational software. Thus, we think we should keep this terminology, to be consistent with the manual. To avoid misunderstandings, we replaced "linearity calibration" by "Correction of c-dependency (called 'linearity calibration' in the instrument's documentation and operational software)" at first occurrence and by "Correction of c-dependency ('linearity calibration')" for the following occurrences.
Concerning the uncertainties related to the referencing: We addressed this question by adding a chapter 'Evaluation of the calibration strategy', c.f. our comment to page p6/20, in particular Table 7

 and Figure 4.

p8/1 This whole chapter is badly written and should be revised with respect to language. In addition, the arguments are not convincing. The concentration range of HS and LS is not any larger than the standards used in the first calibration (300 and 430 ppm). Choosing two out of five standards, that were meant to evaluate accuracy for calibration, leads to only three remaining standards that are perfectly bracketed. The mean and uncertainty at N=3 becomes then statistically very weak. Furthermore, the results for c also illustrate why using just one tank to assess the accuracy of the δ values is not sufficient and somewhat arbitrary (see p6/20 above).

**Author's response:** We originally introduced this post calibration because we found a large jump in the concentration measured with the target standard. No such jump occurred in δ values. The jump in the target concentration could be removed replacing the instrument internal calibration with the applied post calibration. We agree with you, that it is not convincing that this is related to the concentration range of the instrument internal calibration. We think that during this period there was a problem with the instrument internal concentration calibration. The reason is not very clear to us; it might be that we have a problem with target gas flow during this particular concentration calibration. After replacing this particular concentration calibration by the linear post calibration, the corresponding jump in the target standard disappeared.

**Changes to the manuscript:** We rewrote chapter 2.8.2 about the post calibration and applied it only for a time period in which we observed a jump in the target concentration.

p9/15: The authors state that they chose an averaging time of 20 s as compromise between number of measurements and precision. This is misleading or not clear enough. If there are no measurements of standards between 20 s intervals, then the precision does not mean much because the next mean value for 20s may have an excellent precision (given as SD) but may have drifted significantly, thus the two values with good precision cannot be compared at the level of their individual precision (it then becomes an issue of repeatability or accuracy, depending on the context).

**Author's response:** We removed this misleading description.

p9/20 "the mean deviation of N=300 field measurements of a tank with pressurized air" is a suitable way to quantify repeatability and should be compared (or moved) to the results found in the corresponding chapter 3.1.3. Unfortunately, the values are only given graphically in Fig. 3. However, looking at the difference of one (!) sample, one cannot determine accuracy of the spectrometer. Especially not for an analytical technique which is known to be strongly dependent on concentration and gas matrix. The test is thus not suited for its aim. This chapter and the next can be combined to determine repeatability (preferred terminology), and which - at least in the title - may be called long-term stability. However, it is critical to find a way to reliably determine accuracy.

**Author's response/changes to the manuscript:** See comment to p6/20 about the accuracy measurement.

p9/25 "sum of uncertainties". What the authors likely mean is the combined uncertainty or an uncertainty budget. However, this is not achieved by simple addition of the uncertainties, as suggested in the text. It is necessary to know what the authors consider for the individual uncertainty contributions (and why), what distribution they assume and – if the contributions are independent – how they calculate the combined uncertainty, and at what level of confidence they then express this combined uncertainty.

**Author's response/changes to the manuscript:** This section was removed, instead we discuss 'potential uncertainty' as defined by Tuzson 2008.

P10/7 The standard deviations of repeated measurements (0.2 ppm for $CO_2$ concentration and below 0.3‰ δ values) should be compared to literature values. For example, (Sturm 2013, amt-6-1659-2013) found repeated measurements of the same gas tank with a standard deviation which is a factor 4-7 better than the results shown here.

**Author's response/changes to the manuscript:** We added more lab measurements and discuss the repeatability in chapter 'Evaluation of the calibration strategy', including the comparison to literature data, please see our comment to your question p6/20. Here we added the following to the discussion – please note that these values slightly changed, because we removed two periods with known instrument problems (c.f additional footnote and new figure 7) and recalculated the post-calibration only for the time period in which we observed a problem with our target measurements. We changed the chapter 3.1.3 Repeatability during the field campaign. We added the following footnote2 on page 12.

Fig 5 What is the slope of the linear decay, and what process does it represent?

**Author's response:** We derived this linear relationship from a first order approximation for the theoretical (and unrealistic) assumption that no mixing occurs in the measurement cell. We added

the missing relevant information to the manuscript.
**Changes to the manuscript:** We added the missing information to the figure caption.

P1/9 "field conditions" This is ok, but only if sufficiently exhaustive to replace lab characterization.
**Author's response/changes to the manuscript:** We changed this and included in general more lab measurements, c.f our comment to p6/20

P1/9 "accuracy of 0.1 ‰ for $\delta$ $^{13}$C" how can this be smaller than repeated measurements?
**Author's response/changes to the manuscript:** We changed the way how we quantified accuracy, c.f. our comment to p6/20. We changed the abstract to: "The potential accuracy (defined as the 1σ deviation from the respective linear regression that was used for calibration) was approximately 0.45ppm for c, 0.24‰ for $^{13}$C and 0.3‰ for $^{18}$O."

P1/14 "became insignificant" Explanation needed (or explicit statement that no explanation found).
**Author's response:** We are not sure which explanation is needed here a) the correlation itself or b) the change in the correlation coefficient from significant to insignificant? However, in the abstract, we just summarize the observed correlation and shifted the explanation into the discussion, because the discussion of both, a) and b) is a bit long.
**Changes to the manuscript:** "This correlation became insignificant (p>0.1) for the period after the first snow, indicating a decoupling of $\delta^{13}$C of respiration from recent assimilates."

p2/25 and 25: "isotopologues" isotopocules, or isotopologues and isotomers, or remove bracket.
**Author's response:** We think it might be a bit confusing for the reader to use the term 'isotopocule' or add 'isotopomer' here, because the latter is irrelevant for $CO_2$- The Hitran database and many other authors use the the term isotopologue in this context (e.g. Kerstel and Giafriani 2008, Barbour 2011, Ellekoj 2013, Weh4013 ,Oikawa 2017 , Mohn 2007 ,Affek and Yakir 2014, Vardag 2014). However, we added the information about isotopomers in the footnote and tried to give a clearer definition of isotopoloues.

P2/32 Since this is already cited, check the references and cite them directly.
**Author's response/changes to the manuscript:** We agree, but this part was removed anyway to shorten the introduction.

P3/9 The classification does not work for this instrument because it combines mid-IR with enhanced effective optical path length.
**Author's response:** Thanks for pointing that out. We added another class of instruments in Table 2. However, this part of the manuscript was shortened a lot after we read the referees comments.

P3/20 "direct laser absorption spectrometer" direct absorption, not direct laser.
**Author's response/changes to the manuscript:** We changed this to "laser based direct absorption spectrometer" throughout the text to be clearer.

P5/33 "purging pump to avoid condensation in the tubes" purging is ok, but why should it avoid condensation (except because of pressure drop).
**Author's response/changes to the manuscript:** This was misleading, we changed this to: "We purged the main tube to reduce the time the air masses spend in the tubing. To avoid condensation, we heated the valve box (at which we expect a pressure drop) and the adjacent tubing."
P6/5 "the tubes with this small flow rate and" Please explain why condensation is linked to flow rate.
**Author's response/changes to the manuscript:** We removed this sentence.

P7/13 „linearity calibration" It's not really about linearity but about concentration dependence of the retrieved d values.

**Author's response/changes to the manuscript:** We changed the terminology, see our answer to your comment to p7/30.

P9/14 Put in relation. Is this better/worse/comparable?
**Author's response/changes to the manuscript:** We changed this to: "we measured a comparable (slightly better) Allan Deviation below 0.03‰ (c.f. Table 5)."

Multiple comments to chapter 3.1.2
- there is no such thing as "expected uncertainty".
- "measured uncertainty" unsuited terminology
- in the context of the evaluation of a new analyzer you have to make sure that this is not a coincidence. (N=1).
**Author's response/changes to the manuscript:** We replaced this chapter by a chapter about the evaluation of the calibration strategy. Please see our answer to your comment about p6/20

P10/10 "instrument drift" It would be very interesting to know what the instrument drift is. However, the data shown here is the drift of the retrieved data after all (drift) corrections.
**Author's response/changes to the manuscript:** Thanks for pointing this out. As the remaining drift after all drift corrections does not seem a meaningful quantity, we removed this part of the data evaluation.

P10/13 "Turnover time" This implies that it is the "turnover" of a perfectly mixed reactor (cell). However, what you then determine are several elements; I suggest calling this "response time".
**Author's response/changes to the manuscript:** We changed "turnover time" to "response time".

P10/20 „to mixing of gas" please state whether the gas flow is turbulent under the given conditions.
**Author's response/changes to the manuscript:** The gas flow in all tubes is laminar with Reynolds numbers below 100 for all tubing (6mm and 1/16'). We added this information to chapter 2.5.

P13/3 "As soil respiration has been measured to account for around 80% of total respiration in an old beech forest in below 30 km distance to our field site (Knohl et al., 2008), we further focus on soil respiration and discuss the following hypothesis:" Please check if this is really the line of thought that you want to communicate.
**Author's response:** We changed this to "Because soil respiration has been measured to account for around 80% of ecosystem respiration in an old beech forest in below 30 km distance to our field site (Knohl et al., 2008), we assume that soil respiration dominates ecosystem respiration and thus we further focus on soil respiration and discuss the following hypothesis:"

P30/table 5 "not necessary; Figure with 1 s and minimum values is sufficient."
**Author's response/changes to the manuscript:** We would like to keep this table for the readers convenience.

P30/table 6 „not necessary; can be described in one sentence."
**Author's response/changes to the manuscript:** We removed this table and added the numbers directly into the text: "The analyzer's power consumption of approximately 220W was slightly smaller than the power consumption of the basic infrastructure of the setup that included the pump to purge the 9 inlet tubes and the heated valve box (330W)."

P30/table 7 "not necessary; text and Fig. 4 are sufficient."
**Author's response/changes to the manuscript:** We would like to keep this table for the readers convenience.

P33/21 "review language of this paragraph"
**Author's response/changes to the manuscript:** We rewrote this paragraph.

**Author's reply to anonymous referee 2**

1) The authors conclude in the abstract that "1) the new Delta Ray IRIS with its internal calibration procedure provides an opportunity to precisely and accurately measure c, $\delta^{13}C$ and $\delta^{18}O$ at field sites" I am concerned with this statement, because the internal calibration procedure in the IRIS is never actually described. How are the absorption spectra used to calculate isotope ratios, and how are these modified based on the calibration? This point appears critical for understanding whether the internal procedure is adequate and/or necessary, or for understanding what other post-hoc calibrations may be needed. This is a critical gap in the paper. Once cannot simply assume that the manufacturers of the instrument have worked out the details here. There are instruments that are sold that do not necessarily function as advertised, thus it is necessary to validate every step of the way. I would like to see plots and regressions of raw vs. known values for both $\delta^{13}C$ and $[CO_2]$ for a number of different standards spanning a broad range of $\delta$ values and mole fractions of $CO_2$ .

**Author's response:** We added this missing information about the spectral fit, the calibration procedure and about the evaluation of the calibration procedure (including the suggested plots) to the manuscript.

**Changes to the manuscript:**
**Changes to the manuscript:** We added this information to the introduction (page 3 line 12). We added chapter 2.3 about the spectrometer setup to the methods. We changed chapter 2.8.1 about the calibration procedure to provide more detailed information and added chapter 2.6 that describes the additional measurements. We removed the chapter "Accuracy" and replaced it by chapter 3.1.2 Evaluation of the calibration strategy.

2) The authors mention that they used a post-hoc $CO_2$ concentration calibration, but it is unclear how often the additional standards used for this were measured (once? Halfhourly?) in relation to their check standard. Note that quadratic relationships may give a better fit as employed elsewhere for other absorption-based $CO_2$ instruments.

**Author's response:** We originally introduced this post calibration because we found a large jump in the concentration measured with the target standard. No such jump occurred in $\delta$ values. The jump in the target concentration could be removed replacing the instrument internal calibration with the applied post calibration. We agree with you, that it is not convincing that this is related to the concentration range of the instrument internal calibration. We think that during this period there was a problem with the instrument internal concentration calibration. The reason is not very clear to us; it might be that we have a problem with target gas flow during this particular concentration calibration. After replacing this particular concentration calibration by the linear post calibration, the corresponding jump in the target standard disappeared.
**Changes to the manuscript:** Concerning the potential nonlinearity, Figure 5 in the revised manuscript evaluates the instrument's linearity and quantifies the deviations from the linear regression, please see also the chapter "Evaluating the calibration strategy". We rewrote chapter 2.8.2 about the post calibration and applied it only for a time period in which we observed a jump in the target concentration.

3) Given that this is a methods paper, it would have been very useful to see tests using a broader range of $CO_2$ mole fraction and isotope compositions in the range of standards, and to see more standards tested. Without this, we cannot validate the linearity of the instrument both in

concentration and isotope space. This is a critical deficit of the paper. Why was the need for a post-hoc $\delta^{13}C$ and $\delta^{18}O$ calibration not tested or described?

**Author's response/changes to the manuscript:** We addressed this questions by adding an additional chapter about test measurements to evaluate the calibration strategy. This chapter evaluates the concentration dependency of the δ values over a range of 200 to 1500 ppm and includes measurements of different gas tanks with concentrations ranging from (350 to 450 ppm) and isotopic compositions ranging from -37 to -9.7 ‰ for $\delta^{13}C$ and from -35 to -5 ‰ for $\delta^{18}O$. Please see our response to question 1) above.

4) Note that many of the other laser-based isotope instruments achieve much higher precision with frequent (e.g. 20 minute) isotope calibrations in the field. This need appears especially critical here given the large (1 per mil) jumps in $\delta^{13}C$ values observed in the check standards shown in Figure 4. This suggests that there are some serious stability problems that need to be addressed with more frequent isotope calibration.

**Author's response:**

Concerning the large jumps in observed δ values: Thanks for pointing this out. We showed Figure 4 mainly to show the repeatability of the instrument, but we agree with you, that the large (1 ‰) jumps in δ values need more discussion. For these two large jumps, we found explanations: The first of these large jumps appeared in $^{13}C$ after calibration on 23th of September and disappeared after calibration on 29th of September. This jump only occurs in $\delta^{13}C$ of the target measurement. In particular, this jump was not visible in the $\delta^{13}C$ in the measurements of the different heights (see figure R1 for the highest inlet as an example). Thus, we conclude that there was a problem with $^{13}C$ calibration. This problem might be enhanced for δ values that deviate from the 'reference' δ value, in particular for the very depleted target measurement, that was even out of the calibration range.

[Figure]

*Figure R1 Time series of $\delta^{13}C$ values for the time period that shows a large jump in $\delta^{13}C$ for target measurements, but not for the height inlets, shown here as an example for the highest inlet.*

The second large jump in the time series of the isotopic composition of the target gas from the 9th until the 16th of October includes the period during which we had a laser alignment problem and the laser needed to be readjusted. After calibration on 16th of October, the measured target gas value jumped back to its value before the 9th of October. We originally wanted to show all data points for completeness, but as we can relate them to a) a problem with one specific $\delta^{13}C$ calibration that occurs particularly for the very depleted target gas and b) a general laser alignment problem, we

think it is more appropriate to remove the corresponding data points from further analysis. In case of the laser alignment problem we also removed the corresponding time series of the height measurements.

Concerning precision, here we quantify precision by measuring the Allan deviation of the uncalibrated δ values, like many other authors (e.g. Sturm et al 2013, van Geldern et al 2014). For this comparison, Table 2 gives an overview about the precision of the δ value measurements of different laser-based and broadband light source-based instruments. In case you refer to what we called 'long-term-stability' in the original manuscript, but call 'repeatability' in the revised manuscript, this is discussed in the chapter 'Evaluation of the calibration strategy' (See our comment above).

**Changes to the manuscript:** We changed Figure 4/7 and the respective description to "Figure 7 Time series and frequency distributions of half-hourly measurements of the […] target gas [..]for the whole measurement period excluding periods that show problems with target gas flow, calibration and a laser alignment problem. Major reasons for data gaps are marked with different colors."
We added the chapter 3.1.3 "Repeatability during the field campaign.
We added the following footnote*: "In the case of $^{13}$C, we excluded the target measurements between 23$^{rd}$ of September till 29$^{th}$ of September, because we obtained a problem with the $^{13}$C calibration that lead to a large jump in the delta $^{13}$C value of the (very depleted) target standard. This jump did not occur in the height measurements, probably because they were much closer to the reference delta value."

5) With respect to the second major conclusion of the abstract, "2) even short snow or frost events could have strong effects on the isotopic composition of $CO_2$ exchange at ecosystem scale" this finding is not new, but also not very well supported by the data (e.g. Figs 7 and 8. There are now several multi-year datasets of canopy $CO_2$ and δ $^{13}$C profiles in temperate ecosystems that have shown similar patterns.

**Author's response:**
Concerning conclusion 2) Here we summarize the results concerning $^{13}$C as well as $^{18}$O that are discussed in detail in the results section. This statement does not only refer to Figures 7 and 8. The parts of the manuscript that support this conclusion are in particular figure 9 (top panel) for $^{13}$C and Table 8 for $^{18}$O (in addition to figure 7). As we discuss in section 3.2.2, for $^{13}$C we do not observe a change in the δ $^{13}$C values, but we find indications, that the processes controlling the $^{13}$C of $CO_2$ exchange shifted. For brevity in the abstract, we tried to stay general, but specified this in the revised manuscript.
Concerning the mentioned multi-year records: We are well aware that there are now several multi-year records of $^{13}$C and $^{18}$O in $CO_2$ profiles (e.g. Bowling et al 2002b, Wehr et al 2016, Bowling et al 2003, Shim et al 2013). However, we are not so sure if the 'similar' pattern that you talk about show the same change in the time lagged (and negative) correlation between Reco$^{13}$C and 2-d averaged radiation (not VPD), particularly in the combination with frost events. It would be very interesting for us to see which species ($^{13}$C or $^{18}$O) and which datasets you are referring to in particular and we are happy to include the respective citation
**Changes to the manuscript:** We specified the abstract "2) even short snow or frost events might have strong effects on the isotopic composition (in particular $^{18}$O) of $CO_2$ exchange at ecosystem scale."
We added a more comprehensive list of citations, focussing on multi-year record to the introduction (page to page 2 line 12) "The temporal variability of the isotopic composition of respiration for example has been studied on timescales ranging from sub-diurnal (Barbour et al., 2011) to seasonal (Ekblad and Högberg, 2001; Bowling et al., 2002; Knohl et al., 2005). Further, the isotopic composition in $CO_2$ profiles has been studied on several sites over multiple years for $^{13}$C (e.g. Bowling et al 2002b, Wehr 2016 ) as well as for $^{18}$O (e.g. Bowling 2003, Shim et al 2013)."

We added a sentence referring to the observed peaks $R_{eco}^{18}O$ in the monsoon-dominated woodland observed by Shim et al to the discussion: "For comparison, similar strong peaks in $R_{eco}^{18}O$ have been observed in a semi-arid woodland after precipitation in New Mexico (Shim et al 2013), but this study refers to a monsoon dominated ecosystem with comparably large variability in the $^{18}O$ and does not focus on the difference of these pulses of snow and rain events."

6) With respect to the Keeling plot intercepts, no data is shown to actually validate the approach (e.g. plots of $\delta^{13}C$ and $1/CO_2$ space), nor summary statistics presented for these regressions. This is another serious deficit given the key methodological issues the authors point out in the Appendix, but do not quantify in the text. I don't think the authors present enough information here to rigorously test the hypotheses proposed in the Results/Discussion section.

**Author's response:** Below we show the histogram of $R^2$.

[Figure]

*Figure R2 Histograms showing the $R^2$ values of accepted Keeling Plots based on data that was measured within 90 minutes during nighttime (between 20pm and 4 am).*

**Changes to the manuscript:** We provide summary statistics about the regressions to the text:
"The filtered nighttime Keeling-Plot intercepts based on 90 minutes of data acquisition had $R^2$ values with a median of 0.87 and 0.81 for $^{13}C$ and $^{18}O$ with mean values of 0.85 and 0.77 and standard deviations of 0.1 and 0.16 respectively."
We added the following Example Keeling-Plots to the supplementary material. We chose Keeling-Plots with $R^2$ values spanning the range of the respective mean+- 1 standard deviation.

[Figure]

*Figure S05 Example nighttime Keeling-Plots with typical R^2-values (spanning the range of the mean +- 1σ. Each Keeling-Plot is based on 90 min input data. Different colors represent different inlet heights.*

The value of the CANVEG modeling exercise for the overall study was not terribly apparent to me, nor were the questions that it sought to address.

**Author's response:** We included the modelling to test the Hypothesis (a) (page 12 lines 1-14) as discussed in particular in lines 22ff. We modified this explanation to make the reason for the inclusion of the model clearer.

**Changes to the manuscript:** We added an additional sentence to the explanation in section 3.2.2 "Hypothesis (a):The variability of $R^{13}C$ eco can be partly explained by the isotopic composition of recent assimilates $^{13}C$ Ass, which is controlled by meteorological drivers during photosynthesis according to the Farquhar model. Thus, the variability of $R^{13}C_{eco}$ is linked to the variability of meteorological drivers of photosynthesis and photosynthetic discrimination with a time lag that is consistent with the time lag between respiration and assimilation. [...We observed a correlation between radiation $R_n$ and $R^{13}C_{eco}$,...] But the correlation itself cannot be directly explained by the Farquhar model of discrimination as radiation influences both, the $CO_2$ supply (by influencing stomatal conductance) and the $CO_2$ demand (by influencing assimilation) in the leaf (Farquhar and Sharkey, 1982). In particular, we did not find a significant time lagged positive correlation between $R^{13}C$ eco and VPD, RH or the ratio VPD/PAR (Fig. 8), which could be directly associated with the Farquhar Model and has been found by the above mentioned studies. [this refers to (Ekblad and Högberg, 2001; Bowling et al., 2002; Knohl et al., 2005)] To test if it might be still reasonable to interpret the observed negative correlation of $R^{13}C$ eco with Rn as a time lagged link between $R^{13}C$ eco and isotopic composition of recently assimilated material $^{13}C$ Ass on ecosystem scale, we performed a more complex calculation of $^{13}C$ Ass by using the multilayer model CANVEG. The advantage of CANVEG is that it accounts for the non-linear interactions between air temperature, air humidity, radiation, stomatal conductance and photosynthesis."

To explain this thought earlier, we added/moved the following to the beginning of chapter 2.9: "To test if the measured variability of the $_{13}C$ composition of respiration can be partly explained by the variability of the $_{13}C$ composition of recent assimilates, we used the multilayer model CANVEG to simulate the isotopic composition of assimilated material during our measurement campaign. In particular, we analyzed the correlation of modeled $_{13}C_{Ass}$ to net radiation $R_n$, a driver of photosynthesis and photosynthetic discrimination, during our measurement period in autumn 2015.We further compared the resulting relationship between $R_n$ and $_{13}C_{Ass}$ to the observed (time lagged) relationship between $R_n$ and the $_{13}C$ composition of ecosystem respiration $R_{13eco}C$, derived from the measured

7) More specific comments:

Introduction: there is much excessive detail here that repeats recent reviews, such as the Griffis 2013 paper. Please condense.

**Author's response/changes to the manuscript:** We shortened the introduction, in particular p2 line 22 ff.

P1 18: the main constraint is low temporal resolution

**Author's response/changes to the manuscript:** Thanks for pointing this out. We added this information to the manuscript.

P4 13: how are these "physically different" air samples if the pump is flowing continuously?

**Author's response/changes to the manuscript:** We removed this misleading description.

P8 5: "A possible reason for this resulting deviation is the range of the gas tanks we used for the instrument-internal concentration calibration, that was approximately 300 to 430 ppm" this logic doesn't make sense to me this is similar to your other standards
**Author's response:** We agree that this might not be the reason for the observed problems with concentration calibration, please see our answer to your comment 2).

P8 6: I am having trouble understanding how your "target standard" could be stable without posthoc calibration yet your five other standards were so variable.

**Author's response:** This was not the case. We found a need for post concentration calibration because the 'target' standard was not stable, please see our answer to your comment 2).

P8 9: "Secondly we set the IRIS analyzer's internal referencing procedure (described in Sect. 2.7) to 1800 s which corresponds to an Allan variance of 0.03 ‰ for both δ values and 0.01 ppm for $CO_2$ concentration." This is unclear to me are you measuring the standards every 1800 s? For how long?
**Author's response:** Yes, we measured the reference standard every 30 minutes. We measured it for 80s after the tubes were purged for 60s. We rewrote the chapter about the calibration procedure to be clearer (see our comment above).

Where are these new Allan variance values coming from?
**Author's response:** That was a typo. Thanks for finding it!

Figure 4: There are apparently large (1 per mil) jumps in measured "target gas" isotope values at several points. These are disconcerting. Are the data shown in this figure the raw values or the calibrated values? If they are the calibrated values, this suggests that the two-point calibration employed here is inadequate.

**Author's response:** The figure you are referring to (figure 4 in the original manuscript) shows calibrated values. Please see our answer to your comment number 4) above and the new chapter about the evaluation of the calibration strategy.

**Author's reply to anonymous referee 3**

1a) Page 2 lines 13ff: text passage about IRMS: Pls cite Schnyder et al. there (citation below)
1b) in the same text passage: I think "sample preparation effort and cost" might be a minus for IRMS techniques. But here the main disadvantages should be mentioned like (storage) problems with vials (see Gemery et al., 1996 and Knohl et al., 2004) and the advantage of quasi-continuous measurements relative to the "discontinuous" measurement by IRMS.

**Author's response/changes to the manuscript:** Thanks for these suggestions, we added this information to the respective paragraph: "IRMS has been widely used for isotope studies in environmental sciences, but shows limited applicability for *in situ* measurements (Griffis, 2013), but see also the field applicable continuous flow IRMS described by (Schnyder,2004). Disadvantages of flask-sampling based IRMS techniques include high sample preparation effort and costs (Griffis, 2013), low temporal resolution and discontinuous measurements. Additionally, there are potential problems during sample storage and transport, see (Knohl et al 2004) for minimizing such storage effects in case of $^{13}$C. For $^{18}$O storage effects can be related to oxygen exchange between water and $CO_2$ (Gemery 1996, Tuzson 2008)."

2) Page 2 lines 22ff. text passage about different spectrometer types: should be shortened as this manuscript is not a review on optical methods for measurement of isotope ratios

**Author's response/changes to the manuscript:** We shortened the introduction in especially page 2 lines 22ff.

3) Page 3 lines 25ff: "to characterize the Delta Ray IRIS and its performance under field conditions": I think measurement of the "internal cell turnover" and "Allan deviation" is not sufficient to fulfill this topic here. The reference gas box from the Delta Ray is said to offer possibilities to adjust $CO_2$ conc of the "reference" gas to the measured [$CO_2$] to cancel out a possible concentration effects on the measured d-values. The authors need to go more in detail here by showing data (!) from multiple $CO_2$-in air-standards with different [$CO_2$] and different $\delta^{13}$C-and d$^{18}$O values measured with IRMS (preferred) in comparison to measurement with Delta Ray or a comparison with different optical measurement devices (more problematic). I suppose you have measured the data, so show them here please.

**Author's response:** We show the measured concentration dependency as well as a comparison of multiple $CO_2$-in-air standards with IRMS measurements ($\delta$ values) and measurements with a Picarro (concentration) in figures 4,5, and 6.

We added chapter 2.6 '*Instrument characterization measurements'* to describe the additional measurements to the manuscript. We removed the chapter "Accuracy" and replaced it by chapter 3.1.2 Evaluation of the calibration strategy .

4) Please give more info (citation if available) on the kind of measurements performed at the MPIin Jena (isotopes and concentration).

**Author's response/changes to the manuscript:** We added this information to the manuscript: "All used $CO_2$ containing gas tanks were measured high precisely for their $CO_2$ concentration and isotopic composition in $^{13}$C and $^{18}$O at the Max Planck Institute for Biogeochemistry in Jena. There, the $CO_2$ concentrations were measured with a Picarro CRDS G1301 and the isotopic composition was measured with IRMS linked to VPDB (VPDP-$CO_2$) by using the multi point scale anchor JRA-S06 (Wendeberg et al 2013)."

5) The link to VPDP was done with the gas tank measured in Jena? Please extend the info on how this is done. Fig. 3 describes your quality control standard? Is there a way to compare measured values (+ stdev.) with a target value (+stdev.)?

**Author's response:**

Concerning the link to VPDB: We added a chapter that describes the calibration using the tanks that were measured in Jena.

Concerning Fig 3: We used the deviations between measured and target value with the respective

uncertainties to calculate accuracy in chapter 3.1.2. We changed this and used measurements with more than one tank to quantify 'potential accuracy' of the instrument. We added chapter 2.8.1 Instrument internal calibration to the manuscript.

6) Page 3 line 26 "b)" please add one or two sentences why $R_{eco}^{13}C$ and $R_{eco}^{18}O$ is interesting.
**Author's response/changes to the manuscript:** We added one more sentence to the first paragraph in the introduction. "The $^{13}C$ composition of ecosystem respiration $R_{eco}^{13}C$ on the one hand, has been used to assess the time lag between assimilation and respiration (e.g. Ekblad and Högberg, 2001; Bowling et al., 2002; Knohl et al., 2005) and to evaluate biosphere models on global scale (Ballantyne et al, 2011). The $^{18}O$ composition of ecosystem $CO_2$ exchange $R_{eco}^{18}O$ on the other hand is particularly interesting to study the coupled $CO_2$ and water cycle (see e.g. Yakir and Wang, 1996)."

7) Page 11 line 21 "lighter" here means only $^{13}C$-depleted or also $^{18}O$-depleted ? Please specify (also in whole manuscript)
**Author's response/changes to the manuscript:** We specified this terminology throughout the manuscript.

8) Page 13 line 26: more "enriched" in what? Please check that also in whole manuscript, depleted in $^{13}C$, enriched in $^{18}O$ (page 14 line 21...)
**Author's response/changes to the manuscript:** We specified this terminology throughout the manuscript.

9) I'm not totally happy to read a manuscript with 2 hypotheses where one hypothesis can be discarded but the 2nd one cannot be proven. The authors should find a way around this, at least the additional measurements for finally testing should be mentioned and discussed here.
**Author's response/changes to the manuscript:** We added a paragraph that describes which measurements would be needed to support this hypothesis. These measurements are however very laborious, carry high uncertainty by themselves, and beyond the scope of this study. "To test this hypothesis, we would need to measure the amount and the isotopic composition of autotrophic respiration, total soil respiration and ecosystem respiration (e.g. by a trenching experiment) at our field site with an appropriate time resolution to capture the day-to-day variability during the field campaign. Lab measurements using incubations could also give an idea of the isotopic composition of autotrophic and total soil respiration, but would not fully reflect field site conditions. "

10) the unit '‰' is not conform to the SI unit system, what about using "mUr"? It might be more an editorial decision
**Author's response:** As ‰ is so commonly used and also the literature we are citing uses ‰, we think it might be most convenient for the reader is we also use ‰, even if it is not a SI unit. We are however happy to follow the editor's suggestion.

We added the suggested references to the manuscript:

[revised manuscript text omitted]